



# 1 Overview: On the transport and transformation of pollutants
# 2 in the outflow of major population centres - observational
# 3 data from the EMeRGe European intensive operational
# 4 period in summer 2017

M. Dolores Andrés Hernández[1], Andreas Hilboll[2,†], Helmut Ziereis[3], Eric Förster[4], Ovid O.
Krüger[5], Katharina Kaiser[6,7], Johannes Schneider[7], Francesca Barnaba[8], Mihalis
Vrekoussis[2,18], Jörg Schmidt[9], Heidi Huntrieser[3], Anne-Marlene Blechschmidt[1], Midhun
George[1], Vladyslav Nenakhov[1,*], Theresa Klausner[3], Bruna A. Holanda[5], Jennifer Wolf[3], Lisa
Eirenschmalz[3], Marc Krebsbach[10], Mira L. Pöhlker[5], Anna B. Hedegaard[2], Linlu Mei[1], Klaus
Pfeilsticker[11], Yangzhuoran Liu[1], Ralf Koppmann[10], Hans Schlager[3], Birger Bohn[12], Ulrich
Schumann[3], Andreas Richter[1], Benjamin Schreiner[11], Daniel Sauer[3], Robert Baumann[3],
Mariano Mertens[3], Patrick Jöckel[3], Markus Kilian[3], Greta Stratmann[3,**], Christopher
Pöhlker[5], Monica Campanelli[8], Marco Pandolfi[13] Michael Sicard[14,15], José L. Gómez-Amo[16],
Manuel Pujadas[17], Katja Bigge[11], Flora Kluge[11], Anja Schwarz[9], Nikos Daskalakis[2], David
Walter[5], Andreas Zahn[4], Ulrich Pöschl[5], Harald Bönisch[4], Stephan Borrmann[6,7], Ulrich
Platt[11], and John Phillip Burrows[1].
[1]Institute of Environmental Physics, University of Bremen, Bremen, Germany
[2]Laboratory for Modeling and Observation of the Earth System, Institute of Environmental Physics, Bremen,
Germany.
[3]Deutsches Zentrum für Luft- und Raumfahrt (DLR), Institut für Physik der Atmosphäre, Oberpfaffenhofen,
Germany
[4]Karlsruhe Institute of Technology, Institute of Meteorology and Climate Research, Karlsruhe, Germany
[5]Multiphase Chemistry Department, Max Planck Institute for Chemistry, Mainz, Germany
[6]Institute for Atmospheric Physics, Johannes Gutenberg University, Mainz, Germany
[7]Particle Chemistry Department, Max Planck Institute for Chemistry, Mainz, Germany
[8]National Research Council of Italy, Institute of Atmospheric Sciences and Climate (CNR-ISAC), Roma, Italy
[9]Leipzig Institute for Meteorology, Leipzig University, Leipzig, Germany
[10] Institute for Atmospheric and Environmental Research, University of Wuppertal, Wuppertal Germany
[11]Institute for Environmental Physics, University of Heidelberg, Heidelberg, Germany
[12]Institute of Energy and Climate Research IEK-8, Forschungszentrum Jülich, Jülich, Germany
[13]Institute of Environmental Assessment and Water Research, Barcelona, Spain
[14]CommSensLab, Dept. of Signal Theory and Communications, Universitat Politècnica de Catalunya, Barcelona,
Spain
[15]Ciències i Tecnologies de l'Espai-Centre de Recerca de l'Aeronàutica i de l'Espai/Institut d'Estudis Espacials
de Catalunya ), Universitat Politècnica de Catalunya Barcelona, Spain
[16]Dept. Earth Physics and Thermodynamics, University of Valencia, Burjassot, Spain
[17]Centro de Investigaciones Energéticas, Medioambientales y Tecnológicas (Ciemat), Madrid, Spain
[18]Climate and Atmosphere Research Center (CARE-C), The Cyprus Institute, Nicosia, Cyprus
*now at Flight Experiments, Deutsches Zentrum für Luft- und Raumfahrt (DLR), Oberpfaffenhofen, Germany
**now at Deutsches Elektronen-Synchrotron DESY, Notkestr. 85, 22607 Hamburg, Germany
† deceased
*Correspondence to*: M.D.Andrés Hernández (lola@iup.physik.uni-bremen.de)
**Abstract.** EMeRGe (**E**ffect of **Me**gacities on the transport and transformation of pollutants on the **R**egional to
**G**lobal scal**e**s) is an international project focusing on atmospheric chemistry, dynamics and transport of local and
regional pollution originating in megacities and other major population centres (MPCs). Airborne measurements,
taking advantage of the long range capabilities of the HALO research platform (High Altitude and Long range
research aircraft, www.halo-spp.de), are a central part of the research project. In order to provide an adequate set
of measurements at different spatial scales, two field experiments were positioned in time and space to contrast



situations when the photochemical transformation of plumes emerging from MPCs is large. These experiments
were conducted in summer 2017 over Europe and in the inter-monsoon period over Asia in spring 2018. The
intensive observational periods (IOP) involved HALO airborne measurements of ozone and its precursors,
volatile organic compounds, aerosol particles and related species as well as coordinated ground-based ancillary
observations at different sites. Perfluorocarbon (PFC) tracer releases and model forecasts supported the flight
planning and the identification of pollution plumes.
This paper describes the experimental deployment of the IOP in Europe, which comprised 7 HALO research
flights with aircraft base in Oberpfaffenhofen (Germany) for a total of 53 flight hours. The MPC targets London
(Great Britain), Benelux/Ruhr area (Belgium, The Netherlands, Luxembourg and Germany), Paris (France),
Rome and Po Valley (Italy), Madrid and Barcelona (Spain) were investigated. An in-flight comparison of HALO
with the collaborating UK-airborne platform FAAM took place to assure accuracy and comparability of the
instrumentation on-board.
Generally, significant enhancement of trace gases and aerosol particles are attributed to emissions originating in
MPCs at distances of hundreds of kilometres from the sources. The proximity of different MPCs over Europe
favours the mixing of plumes of different origin and level of processing and hampers the unambiguous
attribution of the MPC sources. Similarly, urban plumes mix efficiently with natural sources as desert dust and
with biomass burning emissions from vegetation and forest fires. This confirms the importance of wildland fire
emissions in Europe and indicates an important but discontinuous contribution to the European emission budget
that might be of relevance in the design of efficient mitigation strategies.
The synergistic use and consistent interpretation of observational data sets of different spatial and temporal
resolution (e.g. from ground-based networks, airborne campaigns, and satellite measurements) supported by
modelling within EMeRGe, provides a unique insight to test the current understanding of MPC pollution
outflows. The present work provides an overview of the most salient results and scientific questions in the
European context, these being addressed in more detail within additional dedicated EMeRGe studies. The
deployment and results obtained in Asia will be the subject of separate publications.
**1 Introduction**
In recent decades, the number and size of major population centres (MPCs) have increased dramatically. The
term MPC describes a single metropolitan area or converging urban conurbations with a population exceeding 10
million inhabitants. In 1950, New York and Tokyo were the only two megacities in the world (Gardi, 2017)
whereas for 2018 the United Nations reported 33 megacities and 48 urban agglomerations of 5 to 10 million
inhabitants (UN, 2019). One cause of the recent growth of the number of MPCs is the rapid industrialisation of
some parts of the world, in particular East Asia.
The economic consequences of urbanisation, the spatial growth of MPCs, and, in particular, the environmental
and economical sustainability of megacities, have been a focus of recent discussion (ESPAS, 2018; Melchiorri et
al., 2018; Hoole et al., 2019; Odendahl et al., 2019). The MPC has occasionally been presented as a favourable
urban model, because the concentration of resources and services and the development of more effective
mitigation strategies make it potentially less harmful for the environment than other more dispersed population
distributions (Grimm, 2008; Dodman, 2009). However, the power required for transport, industrial and domestic



purposes, which is mostly generated from fossil fuel combustion, makes MPCs a growing and globally
significant emission source of trace gases and aerosol particles for the troposphere.
High levels of urbanisation are associated with severe air pollution events which lead to adverse effects on
human health (Lelieveld et al., 2015, 2020). Frequent exposure to poor air quality affects the respiratory,
cardiovascular and neurocognitive systems, and is associated with cancer and premature death. The World
Health Organisation has reviewed (WHO, 2013) the scientific evidence for the health risk from particulate
matter (PM), and trace gases such as ozone ($O_3$), carbon monoxide (CO), nitrogen dioxide ($NO_2$), sulphur
dioxide ($SO_2$), metals (e.g. arsenic, lead and mercury) and polycyclic aromatic hydrocarbons (PAH). The effects
of pollution originating from MPCs and the development of adequate control strategies are receiving growing
attention as the public concern about air quality and the interaction of pollution and climate on a warming planet
increases (e.g., Jacob and Winner, 2009). In that respect, the MPC emissions of environmental interest are
aerosol particles, which contain sulphate ($SO_4^{2-}$) and nitrate ($NO_3^-$), particulate organic matter (POM), black
carbon (BC), and ammonium ($NH_4^+$), and long-lived greenhouse gases (GHG) such as carbon dioxide ($CO_2$) and
methane ($CH_4$). Short-lived constituents of smog, such as nitrogen oxides ($NO_x$, i.e., NO and $NO_2$), volatile
organic compounds (VOC), and $SO_2$, react to produce $O_3$ and secondary aerosol particles and also have a climatic
effect (UNEP, 2011; Mar, 2021).
The impact of aerosol particles on climate change has been investigated in detail (e.g. Pöschl, 2005; IPCC report,
2014). The aerosol net radiative effect largely depends on the size and chemical composition of the aerosol
particles which determine their scattering and absorption capabilities (e.g., Haywood and Boucher, 2000).
Furthermore, aerosol particles act as cloud condensation nuclei (CCN) and modify the optical properties and
lifetime of clouds. Anthropogenic aerosol is known to increase the number of cloud droplets while decreasing
their sizes (e.g. Andreae and Rosenfeld, 2008; Campos Braga et al., 2017 and references therein). This results in
extended cloud lifetimes, suppressing precipitation (Rosenfeld et al., 2008). Consequently, an accurate
representation of mass and number concentration, size distribution and chemical composition of particles in
models is essential to assess climatic change (Reddington et al., 2013).
Primary MPC emissions are transported and transformed into secondary pollutants such as $O_3$ or secondary
organic aerosols (SOA) and lead to smog episodes downwind of the source. Modelling studies using artificial
aerosol tracers and estimations of deposition potentials, indicate that about 50% of MPC emitted particles with
diameter ≤2.5 □m ($PM_{2.5}$) deposit more than 1000 km from their source (Kunkel et al., 2012). Chemical and
physical processing of MPC emitted pollutants can in turn be affected by mixing with natural, biogenic and other
anthropogenic emissions from regional sources or long-range transported from other areas (Lawrence et al.,
2007, Monks et al., 2009, Lawrence and Lelieveld, 2010, and references therein).
The specific impact of the plumes from MPCs, therefore, depends not only on the type of emission sources (e.g.
industry, traffic, domestic heating, and generation of electricity) but also on the variability of trace constituent
emissions, the local meteorology and topography. The impact of MPC pollution on the atmospheric composition
has been summarised by Zhu et al., (2012). In spite of the growing number of measurements campaigns,
improved monitoring and modelling capabilities and the results achieved in the last decades, this review
identifies important unresolved issues which limit the assessment of the impact of megacities on air quality and
climate. Some examples are:





• the inaccurate modelling of the global effect of MPCs on anthropogenic emissions resulting from the
127       current inconsistent local and regional MPC emission inventories (Denier van der Gon et al.; 2011, Mayer
128       et al., 2000; Butler and Lawrence, 2009),

• the insufficient sub-grid parametrisation of MPCs in models,
• the inadequate characterisation of pollution transport patterns, and,
• the inaccurate prediction of cumulative pollution events observed in downwind regions of MPCs (Zhang et
132       al., 2007; Kunkel et al., 2012).

In addition, modelling studies indicate that the combined effect of near-surface wind speeds and convection
leads to significant latitudinal differences in regional to hemispheric dispersion characteristics (Lawrence et al.,
2007 and references therein; Cassiani et al., 2013). Plumes emitted at higher latitudes are probably subject to
faster transport than outflows from tropical or sub-tropical MPC, travel larger distances and for time scales
exceeding ten days. Transport and transformation of MPC outflows are affected by the general weather patterns
such as frontal passages and the frequency and duration of stagnation episodes, which are important for pollutant
ventilation. The predicted changes in these patterns indicate that future air quality in MPCs will generally be less
influenced by local emission sources than by the mixing of anthropogenic and natural emissions outside the
MPC (Butler et al., 2012).
In summary, the overall assessment and prediction of the impact of pollution emitted by MPCs on tropospheric
chemistry are challenging. Medium and long-term effects of anthropogenic emissions and their interaction with
natural and biogenic emissions in the local and regional surroundings of individual MPCs are poorly understood
and imprecisely quantified. In addition, controlling policies, changes in land cover and climate might
substantially modify the relation between anthropogenic emissions and both natural aerosol and trace gases, as
predicted by e.g., Butler et al., (2012), and recently reported for East Asia (Fu et al., 2016; Silver et al., 2018 and
references herein; Leung et al., 2018). Decoupling the pollutant input upwind from the MPC emissions remains
essential to establish accurate source-receptor relationships and effective control and mitigation policies. The
current knowledge on all these aspects is still insufficient.
**1.1 Overarching objective of EMeRGe and methodology**
The EMeRGe (**E**ffect of **Me**gacities on the transport and transformation of pollutants on the **R**egional to **G**lobal
scal**e**s) project began in 2016 and is part of the Priority research program of the German Research Foundation
(DFG: Deutsche Forschungsgemeinschaft, www.halo-spp.de) to exploit the High Altitude and Long range
research aircraft (HALO) for atmospheric science. EMeRGe has as an overarching objective the improvement of
the current understanding of photochemical and heterogeneous processing of MPC plumes along expected
transport pathways. This knowledge is required to assess the local and regional impacts of MPC outflows.
EMeRGe has a focus on airborne measurements and fostered cooperation with an international research
partnership (hereinafter referred to as EMeRGe international) to facilitate the delivery and comprehensive
analysis of a unique set of data from aircraft-, ground- and satellite-based sensors. The institutions currently
involved in EMeRGe and EMeRGe international are listed in the supplementary information (see S1 and S2).
Europe and Asia are regions of the world with a differing heritage of pollution control strategies and notable
differences in the number, size and proximity of MPCs as well as in the nature of emissions. For this reason, two
field experiments were designed in EMeRGe to investigate the transport and transformation processes of
pollution plumes originating from European and Asian MPCs. The first intensive observational period (IOP) was



carried out in Europe from 10 to 28 July 2017 with special focus on the study of active plume processing close to
emission sources. The second IOP aimed at the investigation of long-range transport (LRT) of MPC outflows
from the Asian continent to the Pacific during the spring inter-monsoon period and took place with HALO base
in Taiwan from 10 March to 9 April 2018.
EMeRGe aims to identify emission signatures and pollution hot spots by relating observations of pollutants to
simulations and air mass trajectories. Chemical processing of the MPC emissions during transport is evaluated
from the measurement of aerosol particles and trace gases. In particular $O_3$ and its precursors provide
information about the photochemical activity and the transformation of primary into secondary pollutants within
the MPC outflows. Furthermore, measurements at different altitudes downwind of selected MPCs are required
for the identification of plume transport. Mixing of MPC plumes with biomass burning (BB) and mineral dust
transport events and / or convection processes might have an impact in the processing of the MPC outflows.
Finally, the accuracy and suitability of atmospheric chemistry models is investigated by comparing EMeRGe
observations with dedicated simulations from state-of-the-art global and regional atmospheric chemistry models.
The present article describes the experimental design and specific objectives of the IOP of EMeRGe in Europe.
It highlights key research questions and some of the scientific results, which are further explored in forthcoming
papers.
**2 EMeRGe in Europe**
**2.1 MPC pollution in Europe**
The level of urbanisation in Europe is presently ~ 74% and is expected to further increase by 10% up to the
middle of this century (UN, 2019). Large conurbations are a more abundant European urban phenomena than
megacities, of which there are a few. According to the European Environment Agency (EEA), the emission of
air pollutants and precursors has decreased across Europe from the year 2000 to the present, partly as a result of
the EU air quality legislation. Emissions of CO, BC, $NO_x$ and non-methane VOCs have been reduced by around
30% and those of sulphur oxide ($SO_x$, primarily $SO_2$) up to 77%. Nevertheless, the daily and annual $O_3$ and PM
limit concentrations for protection of human health are often exceeded in several areas of the continent (EEA,
2019). Significant differences in pollution and photochemical episodes between Northern and Central Europe
and the Mediterranean region are regularly observed, in particular due to the differences in solar actinic radiation
(Kanakidou et al., 2011).
Europe air quality is frequently influenced by LRT of North American pollution as captured by airborne
measurements and investigated in several model studies (e.g. Stohl et al., 2003; Huntrieser and Schlager, 2004;
Huntrieser et al., 2005). Some evidence of LRT of Asian pollution to the Mediterranean has also been
documented (Lawrence and Lelieveld, 2010; Lelieveld et al., 2002). The chemical signatures of LRT of
pollutants vary depending on pollutant lifetime and mixing. Some recent modelling studies infer that the impact
of non-European pollution on the European surface $O_3$ annual average is larger than previously expected (Jonson
et al., 2018).
In recent years, large European projects such as MEGAPOLI (http://megapoli.dmi.dk) and CityZen (Megacity-
Zoom for the Environment; http://www.cityzen-project.eu), provided comprehensive theoretical and
experimental data about MPCs in Europe. MEGAPOLI was conducted in Paris in summer 2009 and winter 2010
(Beekmann et al., 2015) and investigated source apportionment and photochemical processing of emitted





gaseous and particulate substances using several ground-based stations and measurement vehicles (Crippa et al.,
2013; Freutel et al., 2013; von der Weiden-Reinmüller et al., 2014). Beekmann et al., (2015) estimated the
impact of the urban emissions from the Paris megacity to be relatively low in comparison to other external
industrial sources of pollution. Aircraft measurements were restricted to the near-field outflow (up to 200 km) in
the boundary layer below 700 m asl (Brands et al., 2011; Freney et al., 2014). In comparison, EMeRGe focuses
on the impact of different MPCs in middle and Southern Europe and investigates atmospheric pollution plumes
over much larger latitudinal and longitudinal scales.
CityZen (2008-2011) studied air pollution in and around selected megacities and emission hotspots by using in-
situ and satellite observations (Hilboll et al., 2013; Vrekoussis et al., 2013) as well as a series of different scale
models (Colette et al., 2011; Im et al., 2012). The project focused on selected MPCs such as the Eastern
Mediterranean, the Po Valley, the Benelux region, and the Pearl River Delta for intensive case studies but, in
contrast to EMeRGe, did not conduct measurements of the photochemical evolution in the outflow of the studied
regions.
The above studies focused on trace gases linked to air quality and provided relatively sparse information on
GHGs. Long-lived greenhouse gases such as $CH_4$ and $CO_2$ emitted from individual European urban areas have
been investigated in airborne and ground-based studies, e.g. for London (O'Shea et al., 2014; Helfter et al., 2016;
Pitt et al., 2019), Paris (Bréon et al., 2015; Lian et al., 2019), Cracow (Kuc et al., 2003; Zimnoch et al., 2019),
Berlin (Klausner et al., 2020) and Rome (Gioli et al., 2014). Collectively, they report on inconsistencies between
the current emission inventories and measurements. This indicates the need for further experimental
investigation of the GHG budget in Europe.
The capability of chemistry-transport models (CTMs) to reproduce the variability in air quality of major
anthropogenic emission hot spots in Europe has been evolving and investigated (e.g. Colette et al., 2011, 2012).
State-of-the-art models reasonably captured trends of primary species but the modelling of $O_3$ changes and
projected exposure to $O_3$ pollution in Europe is still challenging.
Overall, the proximity of most European MPCs results in the mixing of different pollution plumes during their
transport. This hampers the identification of the air mass origin. BB and mineral dust events have, moreover, a
variety of impact on the total European burden of atmospheric aerosol and trace gases. Particularly in Southern
Europe, BB and mineral dust plumes occur frequently and can significantly affect the chemical processing of
MPC pollution plumes. BB events from agriculture or wildland fires have a strong seasonal pattern in Europe
(Barnaba et al., 2011). Wildfires emit similar to MPC large amounts of pollutants, e.g. PM, $NO_x$, CO, VOC and
PAH (Andreae, 2019). The number and severity of wildfires are expected to increase in Europe under warmer
and drier conditions as a co-effect of climate change (Forzieri et al., 2017; Guerreiro et al., 2018; Turco et al.,
2018). Desert dust episodes of different intensity originating in North Africa frequently affect air mass
composition and atmospheric stratification over the Mediterranean (Kalivitis et al., 2007; Pey et al., 2013;
Pikridas et al.; 2018), in spring and in summer (Barnaba and Gobbi, 2004; Gkikas et al., 2013; Pey et al., 2013)..
**2.2 Specific scientific questions relevant to EMeRGe in Europe**
EMeRGe in Europe focuses on three primary scientific goals addressing a series of related specific questions:
I. Identification of emission signatures in MPC plumes over Europe
• Are there individual MPC emission signatures identifiable in pollution plumes measured over Europe?





• Is it possible to unambiguously identify MPC plumes after transport times of hours or days by tagging the air
masses in the source regions with passive tracers released at the surface and using airborne sensors
downwind?
• Can the effect of plumes from different emission sources (e.g., anthropogenic, BB, and/or a mixture of them)
on the oxidation potential of the atmosphere be inferred from changes in the $NO/NO_y$ and $NO/VOC$ ratios in
airborne measurements?
• Can airborne measurements detect signatures of urban and other emission sources of $CH_4$ in Europe
adequately?
• How abundant are organic acids in European MPC plumes relative to inorganic acids and what are their main
sources?
• Are satellite measurements of aerosol and trace gases capable of supporting the identification of MPC plumes
and dominant transport paths?
II. Investigation and assessment of chemical processing in MPC pollution outflows
• Is the photochemical activity of MPC plumes readily related to changes in concentrations of radicals and
their precursors measured by the HALO sensors?
• Is the photochemical ageing of MPC plumes well described by the chemical clocks inferred from the
airborne measurements of trace gases and aerosol particles?
• Can the $O_3$ production efficiency and $NO_x$-and VOC-sensitive regimes in MPC plumes be determined? How
do these change with respect to the plume age and mixing with background air?
• Can the importance of the role of formaldehyde (HCHO) as an intermediate product in the oxidation of
VOCs, and glyoxal ($C_2H_2O_2$) and methylglyoxal ($C_3H_4O_2$) in secondary aerosol formation be inferred from
their airborne measurement in MPC pollution plumes?
• Which processes control the heterogeneous formation of HONO in polluted air masses of MPC origin in the
BL and lower troposphere over Europe?
III. Assessment of the relative importance of MPCs as sources of pollution over Europe
• How important are BB and dust emissions to MPC plume photochemistry over Europe in the summer 2017?
• How do the regional $CH_4$ urban emission distributions in Europe compare with previous observations in the
same areas?
• Is it possible to assess the relative role of primary and secondary pollutants in the proximity and in the
outflow of MPCs?
• Are state-of-the-art chemical models capable of adequately simulating transport and transformation of
European MPC outflows?
**2.3 Selection of MPC targets and measurement strategy**
The dominant source of $NO_x$ and CO in the planetary boundary layer (PBL) in Europe is anthropogenic activity,
primarily fossil fuel combustion and biomass burning. Cloud free monthly average tropospheric composites of
$NO_2$ columns retrieved from GOME2-B and OMI instruments on-board the MetOp-B and Aura satellites were
used to identify the major MPCs in Europe during July in the EMeRGe study. Due to its short lifetime, $NO_2$ is a
good indicator of the origin of emission sources. The tropospheric $NO_2$ columns retrieved in July 2016 during
the campaign preparation showed enhanced $NO_2$ concentrations over the London, Moscow and Paris megacities,



over large urban agglomerations such as the Benelux/Ruhr metropolitan area in Central Europe and the Po
Valley in Northern Italy, and over the conurbations in Southern Europe such as Rome, Naples, Madrid and
Barcelona. The satellite observations during the EMeRGe IOP in 2017 confirmed the $NO_2$ hot spots identified
(Fig. 1). The differences observed are most likely related to the special weather situation in 2017, as described in
Sect. 3.1.

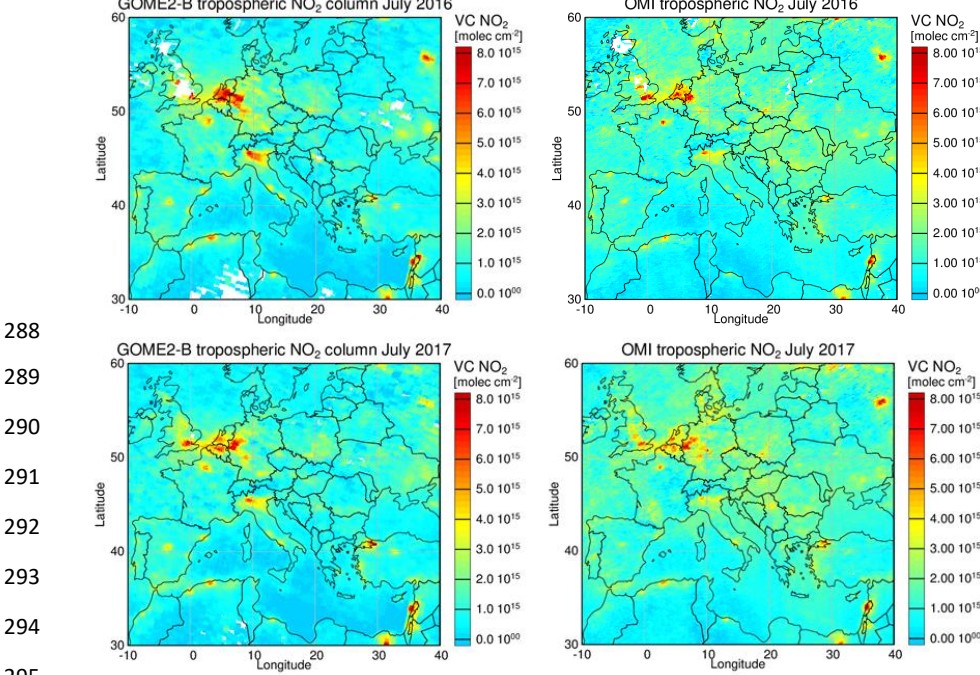









**Figure 1:** Satellite tropospheric $NO_2$ columns retrieved from GOME2-B (left panel, overpass at 9:30 h local time),
and OMI (right panel, overpass at 12:45 h local time) instruments for a) July 2016, a year before the EMeRGe IOP in
Europe (top), and b) the IOP period in July 2017 (bottom).
CO was used in dispersion calculations to identify anthropogenic pollution from combustion. CO is a suitable
tracer for transport pathways due to its relatively long atmospheric lifetime which is primarily loss by reaction
with the OH radical and varies between a few weeks and a few months. To address the EMeRGe scientific
objectives, the day-to-day flight planning focused on the identification of the location of the plumes from the
targeted MPC outflows during potential flights. For this, the following forecast tools were exploited:
i)   ECMWF (European Centre for Medium-Range Weather Forecasts, https://www.ecmwf.int/) and NCEP
(National Center for Environmental Prediction, https://www.ncep.noaa.gov/) weather forecasts,
ii)   NOAA (National Oceanic and Atmospheric Association) HYSPLIT (Hybrid Single Particle Lagrangian
Integrated Trajectories, https://www.arl.noaa.gov/hysplit/) model for forward dispersion calculations using
CO as a tracer of pollution. These forecasts, carried out by DLR (Deutsches Zentrum für Luft- und
Raumfahrt), assume MPCs to be continuous emission sources and provide snap shots as well as horizontal
and vertical cross sections of the selected outflows at certain times.



iii)  Tailor-made CO and stratospheric ozone tracer simulations provided by CAMS (Copernicus Atmosphere
Monitoring Service, http://atmosphere.copernicus.eu) through its field campaign support (see also
Flemming et al., 2019).
A list of model simulations and satellite observations used for flight planning is given in Tables 1a and 1b. These
are described in more detail in the supplement (see S3). The dedicated mission support tool (MSS, Mission
Support System; Rautenhaus et al., 2012) provided additional assistance in the flight planning.

**Table 1a:** Model simulations used for flight planning during EMeRGe in Europe

| Name | Type | Resolution of model output | Institution |
|------|------|---------------------------|-------------|
| CAMS-global (CIFS-TM5) | CTM | 0.4° x 0.4°; 60 vertical levels | ECMWF |
| CAMS-regional ensemble | Median of 7 regional CTMs | 0.1°x 0.1°; surface, 50, 250, 500, 1000, 2000, 3000, 5000 km | ECMWF |
| EMEP | regional CTM | 0.25° E x 0.125° N; 20 vertical levels | Norwegian Meteorological Institute |
| HYSPLIT | Lagrangian trajectory model | 0.1° x 0.1°; 20 vertical levels | NOAA/DLR |
| FLEXPART | Lagrangian trajectory model | 1min /10 days back ECMWF-ERA5; 0.25° horizontal | NILU |


**Table 1b:** Satellite observations used during EMeRGe in Europe

| Sensor name | Satellite | Equator crossing time | Footprint | Institution |
|-------------|-----------|----------------------|-----------|-------------|
| GOME-2 | MetOp-B | 10:30 LT | 40 x 80 km$^2$ | IUP Uni-Bremen |
| OMI | EOS-Aura | 13:30 LT | 13 x 24 km$^2$ | IUP Uni-Bremen |
| SEVIRI | MSG | Geostationary | 3 x 3 km$^2$ | ICARE |


The flight track and patterns available to HALO were constrained by a) flight restrictions from the air traffic
authorities and special military used airspaces (SUA), and b) the unstable meteorological conditions dominating
in Central Europe during the measurement period (see Sect. 3.1).
Flight tracks to investigate the plumes from the MPC targets, London (Great Britain), Benelux/Ruhr area
(Benelux countries and Germany, hereinafter referred to as BNL/Ruhr), Paris (France), Rome and Po Valley
(Italy), and Madrid and Barcelona (Spain) were selected. It was possible to fly these flight tracks under
favourable conditions typically more than once during the EMeRGe IOP, improving somewhat the
representativeness of the measurements.



The HYSPLIT dispersion forecast indicated that the MPC pollution plumes targeted by EMeRGe resided
predominantly below 3000 m. Consequently, the flights over Europe made use of the HALO long-endurance
capabilities to fly in the PBL and incorporated vertical shuttles. The flight pattern involve the descent or climb
between holding altitudes, coupled with long flight tracks at a given flight altitude. Typically, three flight levels
(FL), upwind or downwind of the target MPCs are part of the shuttle. Some of the MPC outflows were tagged by
a coordinated release of a perfluorocarbon (PFC) tracer at the ground (see Sect. 2.4.2).
All HALO flights started from the DLR base Oberpfaffenhofen (OP), located Southwest of Munich in Germany.
The flights are named E-EU-FN, where E stands for EMeRGe, EU for Europe and FN are the two digits of the
flight number. Details about flight tracks and flight routes are provided in Sect. 3.3.

**2.4 EMeRGe instrumentation**

The pollutant measurements made aboard HALO were enhanced during the EMeRGe IOP in Europe by
coordinated flights with other airborne sensors, complementary ground-based measurements and model
predictions. In this manner, the EMeRGe international cooperation provided additional aircraft-, satellite- and
ground-based observations and modelling studies during the preparation and execution phases of the EMeRGe
IOP in Europe, as described in the following sections.

**2.4.1 HALO payload**

A key element of the EMeRGe data are the airborne measurements made on-board HALO, a Gulfstream G550
business jet modified and specifically equipped for scientific research (see www.halo.dlr.de). The HALO
payload for EMeRGe comprises a set of state-of-the-art instrumentation for the measurement of trace gases and
aerosol particles. Table 2 summarises target species and parameters measured by the instruments installed on-
board HALO, which are complemented by the HALO ancillary measurements (BAHAMAS, see S4 in the
supplement) during the EMeRGe campaign in Europe.

**Table 2:** HALO instrumental payload for EMeRGe: PeRCA: Peroxy Radical Chemical Amplification; CRDS: Cavity Ring-
Down Spectroscopy; HVS: High Volume Sampler; GC-C-IRMS: Gas Chromatography Combustion Isotope Ratio Mass
Spectrometry; PTR-MS: Proton-Transfer-Reaction Mass Spectrometer; CI-ITMS: Chemical Ionisation Ion Trap Mass
Spectrometry; GC-MS: Gas chromatography-mass spectrometry analysis; PAN: Peroxyacetyl nitrate; $\delta^{13}$C(CH$_4$): Isotopic
signature of methane; PFC: Perfluorinated carbon chemicals; DOAS: Differential Optical Absorption Spectrometry; AT-BS:
Adsorption Tube and Bag air Sampler; TD-GC-MS: Thermal Desorption Gas Chromatography and Mass Spectrometry; ToF-
AMS: Time of Flight- Aerosol Mass Spectrometry; SP2: Single Particle Soot Photometry; CCNC: Cloud Condensation
Nucleus Counting; MI: Multi Impactor for aerosol off-line analysis; CPC: Condensation Particle Counting; DMA:
Differential Mobility Analysis; OPC: Optical Particle Counting; PSAP: Particle Soot Absorption Photometry. See details and
HALO ancillary measurements in the supplement. The instrument details are given in the quoted literature.





| Trace gas-in situ measurements | | | | |
|---|---|---|---|---|
| Species/parameters | Acronym | Institution | Technique/Instrument | Reference |
| $RO_2^* = HO_2 + \sum RO_2$ | PeRCEAS | Univ. Bremen | PeRCA + CRDS | George et al., 2020 |
| VOC/C isotope ratios | MIRAH | Univ. Wuppertal | HVS/GC-C-IRMS | Wintel et al., 2013 |
| OVOC | HKMS | KIT Karlsruhe | PTR-MS | Brito and Zahn, 2011 |
| $O_3$ | FAIRO | KIT Karlsruhe | UV-Photometry/ Chemiluminescence | Zahn et al., 2012 |
| $O_3$, CO | AMTEX | DLR-IPA | UV-Photometry/ VUV-Fluorimetry | Gerbig et al, 1996 |
| NO, $NO_y$ | AENEAS | DLR-IPA | Chemiluminiscence/ Gold converter | Ziereis et al., 2004 |
| $SO_2$, HCOOH | CI-ITMS | DLR-IPA | CI-ITMS | Speidel et al., 2007 |
| a) $CO_2$ and $CH_4$ | | | a) CRDS | Chen et al., 2010 |
| b) PAN | CATS | DLR-IPA | b) GC-MS | Volz-Thomas et al., 2001 |
| c) $\delta^{13}C(CH_4)$ | | | c) GC-IRMS | Fisher et al., 2006 |
| PFC tracer | PERTRAS | DLR-IPA | AT-BS/TD-GC-MS | Ren et al., 2015 |
| Trace gas- remote sensing measurements | | | | |
| Species/parameters | Acronym | Institution | Technique/Instrument | Reference |
| $NO_2$, HONO, BrO, $CH_2O$, $C_2H_2O_2$, $C_3H_4O_2$, $SO_2$, IO | mini-DOAS | Univ. Heidelberg | DOAS / UV-nIR: 2D optical spectrometer | Hüneke et al., 2017 |
| $NO_2$, $CH_2O$, $C_2H_2O_2$, $H_2O$, $SO_2$, BrO, $O_3$ | HAIDI | Univ. Heidelberg | DOAS / 3x2D-imaging spectrometers | General et al., 2014 |
| Aerosol measurements | | | | |
| Species/parameters | Acronym | Institution | Technique/Instrument | Reference |
| Particle composition | C-ToF-AMS | MPIC Mainz & Univ. Mainz | ToF-AMS | Schulz et al., 2018 |
| BC, CCN, microscopic properties | CCN-Rack | MPIC Mainz | SP2 CCNC, MI | Holanda et al., 2020 Wendisch et al., 2016 |
| Particle size distribution/number concentration | AMETYST | DLR-IPA | CPC, OPC, PSAP, DMA | Andreae et al., 2018 |
| Other parameters | | | | |
| Species/parameters | Acronym | Institution | Technique/Instrument | Reference |
| Spectral actinic flux density (up/down) Photolysis frequencies | HALO-SR | FZ Jülich | CCD spectro- radiometry | Bohn and Lohse, 2017 |
| Basic aircraft data | BAHAMAS | DLR -FX | various | Mallaun et al., 2015 |








### 2.4.2 Perfluorocarbon tracer experiments


Tracer experiments were performed during EMeRGe using perfluorocarbon compounds (PFC). PFCs are
suitable tracers as they are chemically inert, do not interact with aerosol and clouds, have very low background
in the atmosphere (~10 ppqv), and can be detected at mixing ratios as low as 1 ppqv. The tracer experiments
involved the release of a mixture of PFCs at a site close to the centre of an MPC. These experiments establish
Lagrangian connections between MPC centres and HALO measurements downwind. They support the studies on
the formation of secondary gases and aerosol particles from the primary emissions in the pollution plumes. In
addition, tracer experiments were used to test the dispersion parametrisations in transport models.
During the EMeRGe IOP in Europe, PMCH ($C_7F_{14}$, 350 amu) was the PFC used to tag polluted air masses at the
release sites. The tracer was sampled on sorption tubes on-board and subsequently analysed in the laboratory, as
described in Ren et al., (2013, 2015). The limit of detection (LOD) and limit of quantification (LOQ) of the PFC
analysis system are 0.7 ppqv and 2 ppqv, respectively, for sorption tube samples loaded for 3 min. The precision
and accuracy are 6% and 11%, respectively. Three tracer releases were performed two in the city centre of
London at the Imperial College on 17 and 26 July 2017 and one in the Ruhr region, at the University of
Wuppertal on 26 July 2017 in Germany. The HALO flights and pattern for the tracer sampling in the plumes
downstream were optimised with respect to the time of the tracer releases by using HYSPLIT tracer dispersion
forecasts. Post-campaign comparisons of the tracer measurements were performed with HYSPLIT and
FLEXPART. More details of the EMeRGe tracer experiments are described in Schlager et al. (2021, in
preparation).

### 2.4.3 Other airborne observations


The Facility for Airborne Atmospheric Measurements (FAAM, see www.faam.ac.uk) from the UK Natural
Environment Research Council (NERC) joined the EMeRGe IOP in Europe. It made a set of flights around
London in the Southeast of England in the UK.
To assure the accuracy and comparability of the instrumentation on-board, one research flight on 13 July 2017
was dedicated to common and simultaneous measurements of HALO and FAAM in a so-called blind
intercomparison exercise. The two research aircraft flew in close formation for 1.6 hours around noon in the
northern part of a restricted airspace. In total, 24 instruments were operated on the two aircraft and provided data
for the comparison. The data obtained were uploaded under blind conditions and evaluated by an external
referee. In addition, observational data were collected from the German Meteorological Service at the
observatory Hohenpeissenberg (47°48'N, 11°01'E) located downwind of the aircraft track, and model results
were generated from 6 models and interpolated along the common flight path. A summary of the measured and
modelled data available for direct comparisons is provided in the supplement (S5). Overall, about half of the data
pairs from the sets of measurements on the two aircraft differ less than their combined error estimates. In most
cases, the differences between the measurements are smaller than the deviations between the model results. For
some instruments, the comparison led to significant data analysis improvements. The root mean square
deviations between the measurements on FAAM and HALO were less than estimated errors for temperature,
relative and absolute humidity, $CO_2$, benzene, vertical and horizontal wind components, and methane. The





largest discrepancies were found for some VOCs, sulphate aerosol and black carbon mass and number
concentrations. The instrumental accuracy assessment from the comparison results in Schumann (2020).
The Italian Sky Arrow Environmental Research Aircraft (Gioli et al., 2009) from the National Research Council
of Italy (CNR) undertook additionally two research flights up to 2000 m over the city of Rome (Italy)
concurrently with the HALO overpass flight on 11 July 2017. The aircraft was equipped with instrumentation
targeting some aerosol parameters (total number and size distribution), gas concentrations ($CO_2$, $O_3$, $H_2O$) and
key meteorological data (temperature, pressure and wind).

### 2.4.4 Collocated ground-based observations

EMeRGe was supported by measurements from a variety of ground-based stations which complemented the
HALO observations. These measurements were also used for the planning of subsequent HALO flights and
occasionally for in-flight manoeuvres.
For example, the European Aerosol Research Lidar Network, EARLINET (Pappalardo et al., 2014), a key
component of the Aerosols, Clouds and Trace gases Research Infrastructure ACTRIS, joined as an EMeRGe
international partner and provided coordinated, ground-based lidar measurements. Additional support was
provided from other non-EARLINET lidar stations. Altogether, 19 stations supported the EMeRGe IOP in
Europe. The specifications and location of the operated lidars as well as the coordinated measurements for each
HALO flight are included in the supplement (S6).
In addition, measurements from several ceilometer networks contributed to EMeRGe, in particular the German
Ceilonet of DWD (Deutscher Wetterdienst), the Italian ALICEnet (Automated Lidar-Ceilometer network) and
the ceilometers of the Belgian RMI (Royal Meteorological Institute of Belgium). The RMI also provided ozone
soundings from Uccle three times per week. Additional ground-based and in-situ measurements were provided
from ACTRIS stations, and sun-photometer measurements from AERONET (Aerosol Robotic Network, Holben
et al., 1998)
Two ground-based field campaigns deploying both remote sensing and in-situ measurements concurred with the
EMeRGe IOP: ACTRIS-2 in the Po Valley, Italy (see http://actris-cimone.isac.cnr.it/), and HOUSE (High
Ozone, Ultrafine particles and Secondary aerosol Episodes in urban and regional backgrounds) in Northeast
Spain (see https://www.idaea.csic.es/egar/portfolio-items/house/). These data were made accessible for the
analysis in the framework of EMeRGe international.

### 2.4.5 Satellite observations

Near real-time tropospheric $NO_2$ columns from the GOME-2 instruments on MetOp-A (GOME2-A; 40 km x 40
km resolution) and MetOp-B (GOME2-B; 80 km x 40 km resolution) as well as OMI (13 km x 24 km resolution
at nadir) on NASA Aura were provided in July and August 2017 to support flight planning and quick-look
interpretation of the EMeRGe IOP observations. $NO_2$ columns are calculated using the method described in
Richter et al., (2005, 2011), and Hilboll et al., (2014). The retrievals use GOME-2 lv1 data provided by
EUMETSAT and OMI lv1 data provided by NASA. They are not official GOME-2 / OMI data products. The
plots were usually available 6 hours after measurement (https://www.iup.uni-bremen.de/doas/emerge.htm).
In addition, daily values of the aerosol optical thickness (AOT) at 0.55 μm were retrieved from the Spinning
Enhanced Visible and Infrared Imager (SEVIRI) on-board the Meteosat Second Generation (MSG) satellite. The
spatial and temporal resolutions for the SEVIRI AOT product are 3 km at nadir and 15 minutes, respectively.



The SEVIRI AOT product over land (SMAOL_AOT.v1.3.6) and ocean (SEV_AER-OC-L2.v1.04) (Thieuleux et
al., 2005; Bréon et al., 2011) are merged and post-processed by using the eXtensible Bremen Aerosol/cloud and
surfacE parameters Retrieval (XBAER) algorithm to minimise potential cloud contamination (Mei et al., 2017a,
2017b).
**3 Characteristics of the EMeRGe IOP in Europe and its conditions**
The EMeRGe IOP in Europe took place from 10 July 2017 to 28 July 2017. The results obtained are analysed
considering the prevailing meteorological conditions in Europe during this period and the characteristics of the
deployment in the different flight legs.
**3.1 Meteorological conditions**
The month of July was selected for the EMeRGe investigation because the summer period in Europe offers
frequent events of high temperature and high insolation, which result in active photochemical processing of the
air masses.
The monthly average weather conditions of July 2017 were evaluated by comparing 500 hPa geopotential height,
temperature, wind and precipitable water with a 30-year (1981-2010) reference climatology using NCEP
reanalysis data (Kalnay et al., 1996). As shown in Fig. 2, stagnation events, high temperatures and insolation
dominated Southern Europe similar to the average of the 30-year climatology. At the ground, the summer 2017
was characterised by heatwaves, which contributed to the propagation of frequent fire events especially on the
Iberian Peninsula (EEA, 2018). In contrast, an upper-level negative pressure and temperature anomaly was
located over Northern Europe. The polar front was positioned further southwards than is usual with anomalously
high upper-level wind speeds over Central Europe. These conditions favoured the passage of upper-level troughs
associated with mid-latitude cyclones and enhanced precipitation over Central Europe. A cut-off low located
over Great Britain during approximately the last ten days of the campaign affected the average weather
conditions. Thunderstorms frequently developed near the Alps over Southern Germany and Northern Italy.

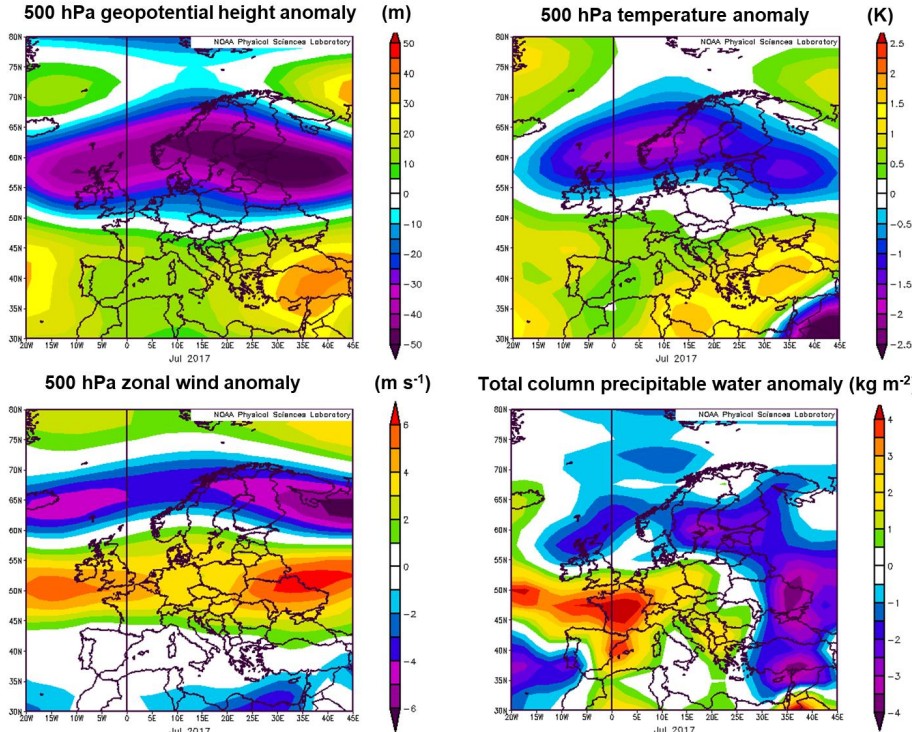

**Figure 2:** Mean anomalies of the 500 hPa geopotential height (top left panel), temperature (top right), zonal wind (bottom left) and total column precipitable water (bottom right) for July 2017 with respect to a 1981-2010 July climatology based on NCEP reanalysis data (Kalnay et al. 1996). Total column precipitable water is the amount of water potentially available in the atmosphere for precipitation from the surface to the upper edge of the troposphere. NCEP reanalysis data and images provided by the NOAA/ESRL Physical Sciences Laboratory, Boulder Colorado (http://psl.noaa.gov/).

### 3.2 Aerosol optical depth

The aerosol load in the target regions during the EMeRGe IOP in July 2017 was investigated. Monthly averages of aerosol optical depths (AODs) measured in July 2017 at 14 AERONET sun-sky photometer sites (AERONET, 2020), in all six EMeRGe target regions (see S7 in the supplement) were compared to the 10-year AOD July average between 2009 and 2019. Throughout this study, only version 3 level 2.0 data were considered (Giles et al., 2019). The measurements at 1020 nm presented here have the largest data coverage (139 data points). Data for other wavelengths (500 nm, 118 data points; and 675 nm, 132 data points) are shown in the supplement. Figure 3 displays the derived AODs. The AODs measured in July 2017 close to Paris and in Southern Great Britain are very similar in the period 2009 to 2019. The AODs are within the standard deviation of the 10-year average for the majority of the other stations with relative deviations ranging from 10% to 14%. In contrast, the AOD observed in the Rome region was 22% lower than the 10- year average.



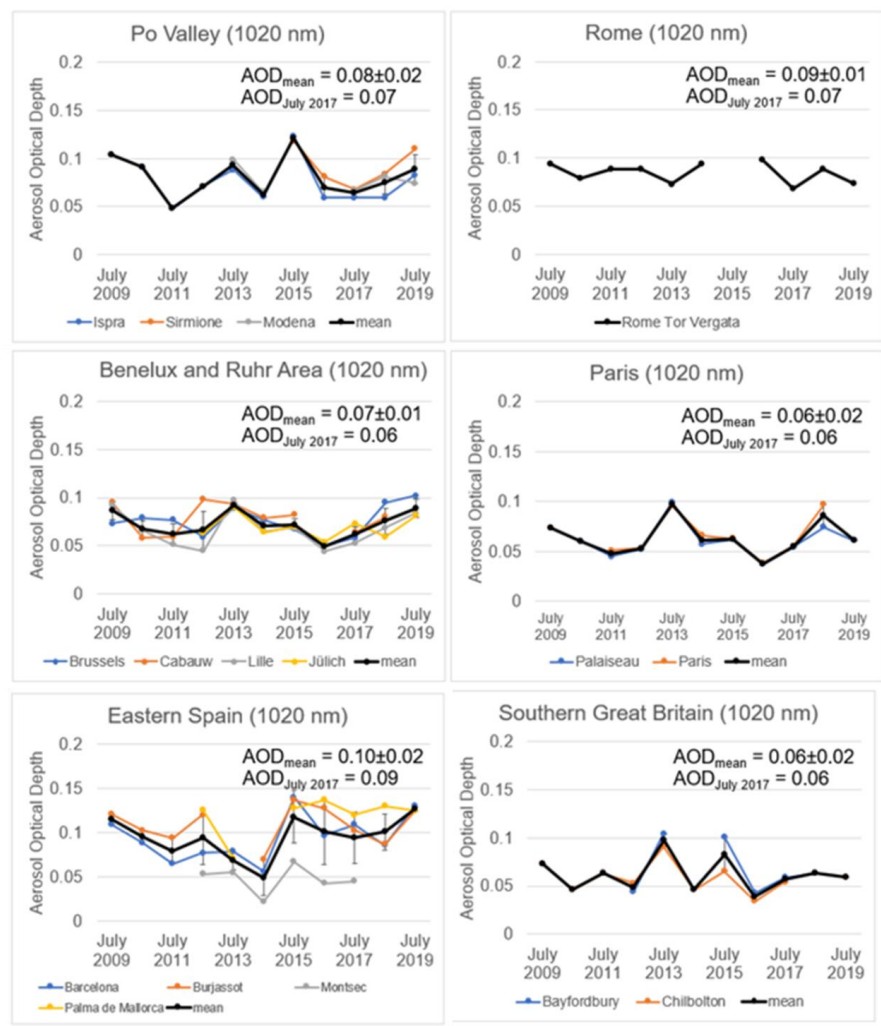

**Figure 3**: AODs derived at 1020 nm for AERONET stations in all six target regions of EMeRGe in Europe. Black lines show mean AOD values. The AODs derived for July 2017 and the 2009 to 2019 average are shown on each diagram. The AODs from July 2017 are representative of the average AODs from 2009 to 2019.

### 3.3 Flight routes and HALO flight tracks

The EMeRGe IOP in Europe comprised seven HALO flights from 11 July 2017 to 28 July 2017, for a total of 53 flight hours. As mentioned in Sect. 2.3, all HALO flights started from OP in Germany. The flight tracks are shown in Fig. 4 and Table 3 summarises the corresponding flight times and targets.

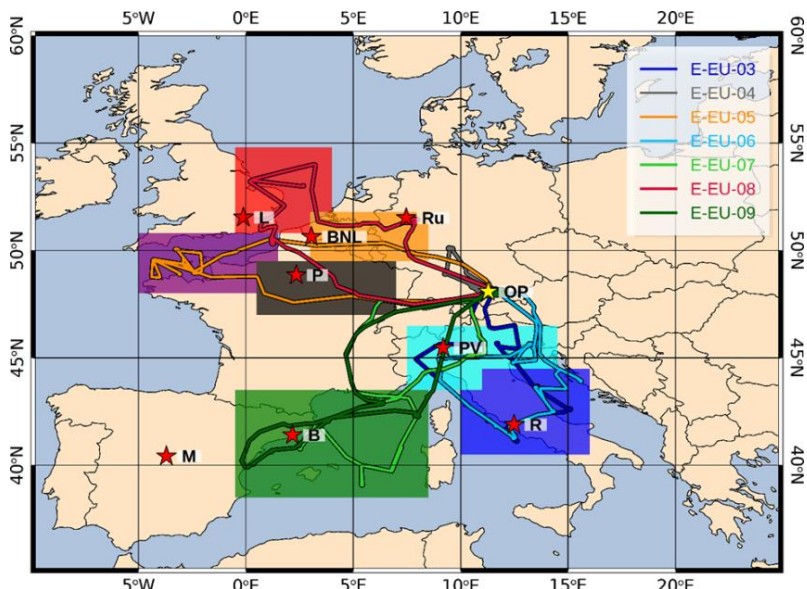

488

**Figure 4:** HALO flight tracks during the EMeRGe campaign in Europe on 11, 13, 17, 20, 24, 26 and 28 July 2017 (E-EU-03
to E-EU-09, respectively, colour coded). The specific flight times are presented in Table 3. MPC target areas are colour
coded by shading: English Channel (purple) North Sea (red) Benelux/Ruhr (orange), Paris (black), Po Valley (cyan), Central
Italy (blue), East Mediterranean (green).Distinctive locations/regions are marked with red stars, M: Madrid, B: Barcelona, P:
Paris, L: London; BNL: Benelux; Ru: Ruhr area; PV: Po Valley, R: Rome. The coordinates of the MPC areas can be found in
the supplement (S8). The position of the HALO base at DLR in Oberpfaffenhofen (OP) is also indicated by a yellow star for
reference.

Overall, 60% of the HALO measurements during EMeRGe in Europe were performed below 3000 m to probe

fresh and transported outflows of selected MPCs (see Fig. 5 for the distribution of HALO flight altitudes during

the EMeRGe IOP).

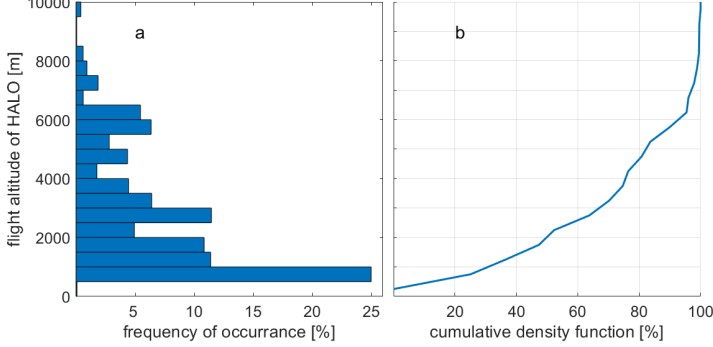


**Figure 5:** Frequency of occurrence of flight altitudes during EMeRGe in Europe in bins of 500 m, a) cumulated
frequencies of flight altitudes from the ground to 10000 m b) cumulative density function.



**Table 3:** Characteristics of the HALO flights carried out in Europe during EMeRGe. FR: flight route. Note that E-EU-01 and E-EU-02 were technical flights and are not considered in the present work.

| Flight number | Day/ Month | Start/ End time (UTC) | FR | MPC emission and transport targets | Other features |
|---|---|---|---|---|---|
| E-EU-03 | 11/07 | 10:00/16:30 | 1 | Rome, Po Valley; convection over Alps and Apennines | Mineral dust from Northern Africa; Fires in Southern Italy. Flights Sky Arrow over Rome |
| E-EU-04 | 13/07 | 10:40/15:00 | 2 | Central Europe; Intercontinental transport | HALO-FAAM blind comparison Canada fires |
| E-EU-05 | 17/07 | 10:30/18:30 | 2 | London, BNL/Ruhr, English Channel and Central Europe | FAAM flights over London PFC tracer release |
| E-EU-06 | 20/07 | 9:00/17:30 | 1 | Rome, Po Valley; Convection over Alps and Apennines | Mineral dust from Northern Africa; Fires in Southern Italy and Croatia |
| E-EU-07 | 24/07 | 9:45/18:15 | 3 | Po Valley, South France, Barcelona; West Mediterranean | Dust transport from Northern Africa, fires in Southern Europe |
| E-EU-08 | 26/07 | 7:45/15:20 | 2 | London, BNL/Ruhr, Paris; English Channel and Central Europe | PFC tracer releases London, Wuppertal |
| E-EU-09 | 28/07 | 10:00/18:30 | 3 | Po Valley, South France, Madrid, Barcelona; West Mediterranean | Fires in Southern France and Portugal |


Different flight routes were selected to optimise the identification and measurement of outflows of target MPCs
under the prevailing meteorological conditions. Taking the measurement objectives, the flight constraints and the
weather conditions into account, three flight routes were selected for the EMeRGe IOP:
a)  Flight route 1: Southern Europe - Italy
b)  Flight route 2: London and Central Europe
c)  Flight route 3: Southwestern Europe

**a)  Flight route 1: Southern Europe- Italy**
The flight route 1 was selected for the HALO flights E-EU-03 and E-EU-06 on the 11 and 20 July 2017,
respectively.
The synoptic situation in Europe during these days was characterised by a high-pressure system over the
Mediterranean region and a cut-off low over the British Islands associated with the rapid passage of low-pressure
systems over Great Britain and Scandinavia. As a result, a Southwest flow with a trough approaching from the
West and a short wave passage dominated. These conditions were suitable for the investigation of the MPC
targets in Italy (Po Valley and Rome) and of the transport of pollution over the Alps and Apennines.
Along the flight route, cloud formation in the Po Valley and thunderstorms in Southern Germany in the
afternoon after 15 UTC were observed on both days.
During these flights, BB emissions from forest and intentional fires in Southern Italy, particularly in the Naples
area and along the coast of Croatia were detected. In addition, the transport of mineral dust from Northern Africa
to the central Mediterranean and the Italian west coast was observed.
The E-EU-03 and E-EU-06 flights were carried out over approximately the same geographical area. Initially
HALO flew over the Alps, then along the Po Valley to the Mediterranean coast of Italy. During E-EU-06 the



vertical and horizontal distribution of pollutants was investigated in more detail by shuttles before entering the
Po Valley and flying at lower altitudes. The tracks followed the Tyrrhenian Sea heading to the South and
crossing the Italian Peninsula from West to East towards the Adriatic coast after a shuttle upwind of Rome.
Along the Adriatic coast, shuttles were made while flying to the North. Finally, the flights crossed over the Alps
back to OP. The E-EU-06 flight track details are summarised in Fig. 6.
During E-EU-03 the HALO airborne measurements were complemented by two circuits around Rome by the
Sky Arrow aircraft and its payload, starting at 8 UTC and at 12 UTC, respectively. Each circuit comprised three
vertical spirals from 200 m to 1800 m altitude approximately. In addition, ground-based measurements of trace
gases and aerosol particles are available at selected sites (see S6 in the supplement). The interpretation of these
airborne and ground-based observations is discussed in Barnaba et al. (2021, in preparation).
Whole air samples for VOCs and their carbon isotope ratios were collected at the ground in evacuated canisters
to determine a representative VOC fingerprint for Rome and Milan. To account for emission variations on the
ground during the day, air samples were taken around 9 to 10 and 14 h local time.

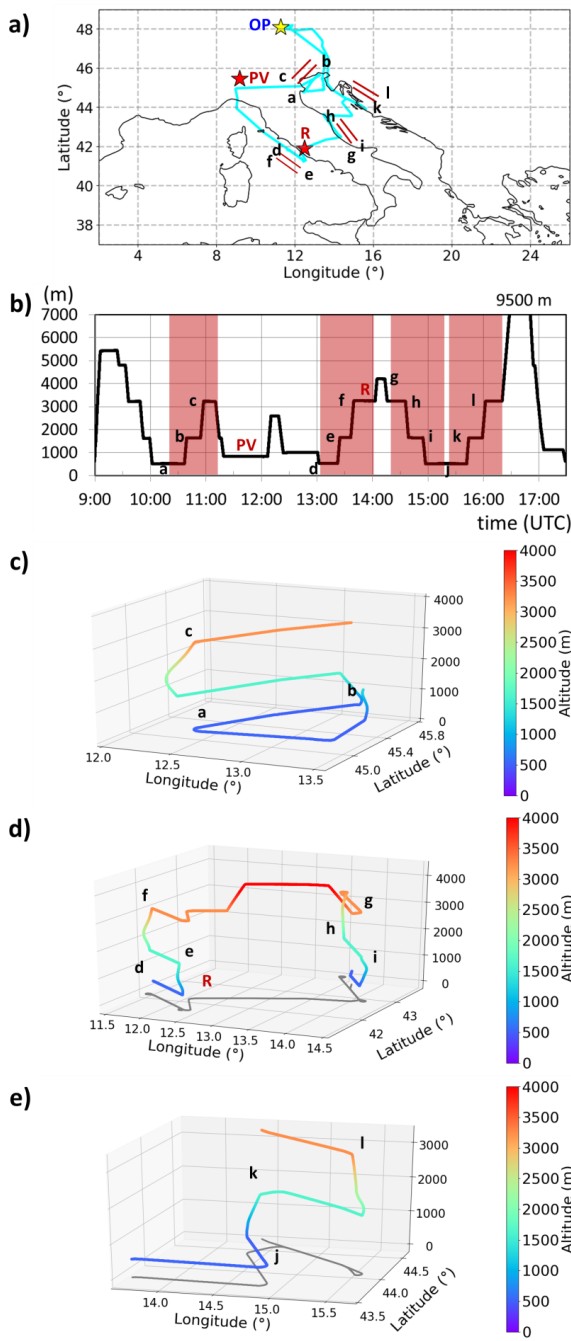

**Figure 6:** Details of the E-EU-06 flight on the 20 July 2017. Three shuttles took place downwind of the Po Valley (PV), upwind of Rome (R) and along the Adriatic coast and are marked with red lines on the map in a) as red shaded areas on the altitude diagram in b), and as a 3-D depiction in c), d) and e). The flight tracks during the shuttles d) and e) are shown in grey. The flight track in a) is coloured as in Fig. 4 and the EMeRGe MPC targets in red. Main changes in course and altitude are marked (a-l) on the graphs for reference. OP indicates the position of the HALO base.



**b) Flight route 2: London and Central Europe**
Flight route 2 was selected to study the London and BNL/Ruhr outflows with a scientific focus on their transport
and interaction over Central Europe. As mentioned in Sect. 3.1, July 2017 had an unsettled weather in the UK
and Central Europe with heavy, persistent rain at times and only brief hot spells. This made the selection of
optimal flight tracks for this investigation challenging. The precise flight route 2 was tailored for the
meteorological conditions prevailing during the E-EU-05, and E-EU-08 flights, which took place on 17 July and
26 July 2017 respectively, to optimally cover different aspects of the target outflows.
The flight E-EU-05 took advantage of a short high-pressure ridge that formed behind a trough over Scandinavia
on 17 July 2017. The outflow of the MPC London was predicted to travel to the English Channel and the
Northern coast of France. This area is regularly used by the UK and French air forces whose activities in the
SUAs constrained the original flight options and the flight track were optimised during the flight route. Over the
area of interest, HALO flew at different altitudes within the PBL. On the way back to OP, the outflow of Paris
was probed South of Orly. On that day, the FAAM platform carried out two complementary circuits around
London at 8:00 and 13:30 UTC.
On 26 July 2017, the synoptic situation changed slightly as a cut-off low moved eastwards over Germany while
a trough approached from the West. In the period after the cut-off low and before the passage of the warm front
over London, the route of E-EU-08 was chosen such that the outflow of London close to the East coast of
England and its mixing with the BNL/Ruhr outflow over the European continent were probed (see Fig. 7).
Cloudy conditions predominated throughout the day. This flight is studied in more detail in Sect. 4.2.
The identification of the London outflow was confirmed by the on-board measurement of a PFC tracer released
in the centre of London for both flights. During E-EU-08, a second tracer release was carried out in Wuppertal in
the afternoon to identify the BNL/Ruhr outflow. In addition, information on the isotopic fingerprints in VOCs
representative for London and Ruhr MPC air were obtained by collecting whole air samples at the tracer release
sites before, during and after the release, and in the afternoon (see Sect. 4.2.).
The E-EU-04 flight track on 13 July 2017 is a particular case that also covered Central Europe (see S9 in the
supplement). The first part of the flight was dedicated to the blind instrumental intercomparison between the
HALO and FAAM platforms described in 2.4.3 (see Schumann, 2020). A weak high-pressure ridge over
Germany dominated. The main objective for the rest of the flight was to probe intercontinental pollution
transport between 5000 and 7000 m altitude with signatures of fires originating in Canada.

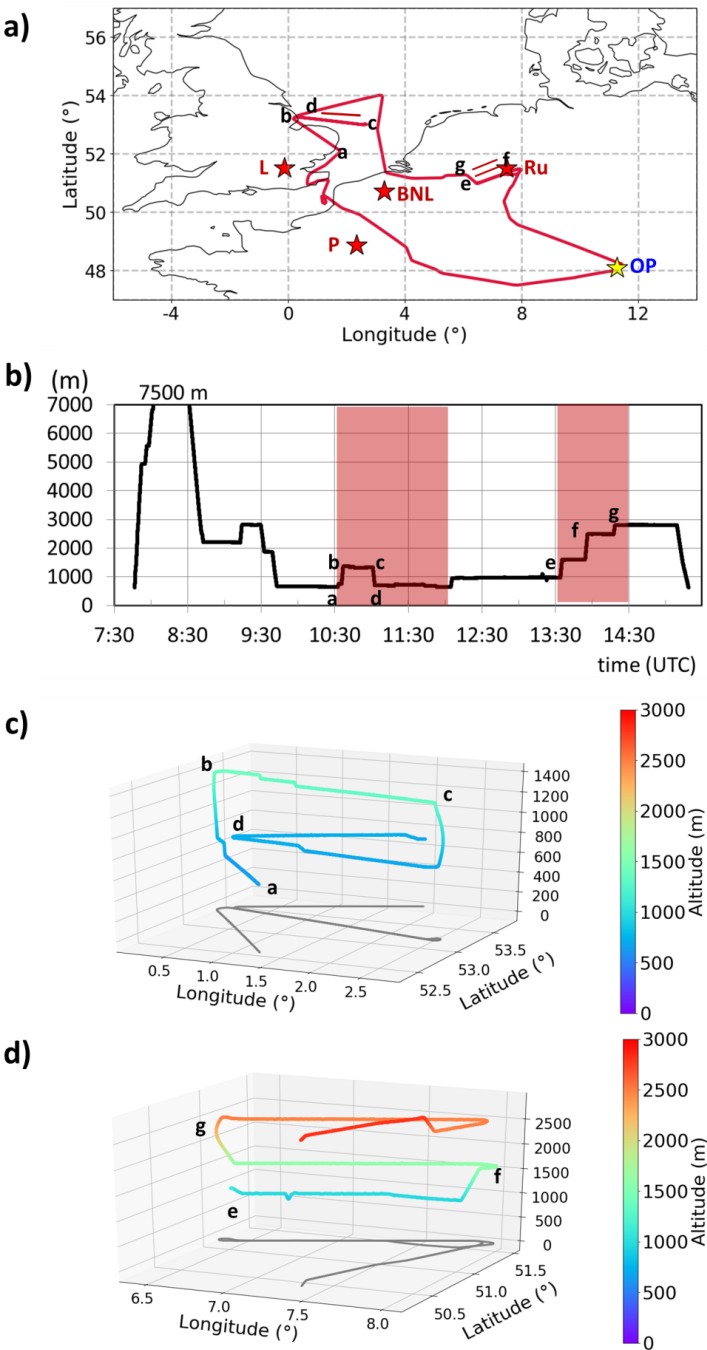

**Figure 7:** Details of the E-EU-08 flight on the 26 July 2017. The position of the shuttles downwind from London and the BNL/Ruhr area are indicated in red on the map in a), marked by the red shaded areas in b), and as a 3-D depiction in c) and d). The flight tracks during the shuttles are shown in c) and d) in grey. In a) the EMeRGe MPC targets are shown in red and the flight track coloured as in Fig. 4. Main changes in course and altitude are marked (a-g) on the graphs for reference. OP indicates the position of the HALO base.





**c)**      **Flight route 3: Southwestern Europe**
The objective of flight route 3 was to investigate the transport of Southern European MPC outflows into the
Western Mediterranean. This flight route was selected for the E-EU-07 and E-EU-09 flights on the 24 and 28
July 2017, respectively.
The meteorological situation on 24 July 2017 over Europe was characterised by the eastwards displacement of a
cut-off low leaving the British Islands. This was associated with a Southwest flow during the passage of a trough
over Spain and France. Dust transport from Northern Africa, thunderstorms in the Po Valley and fires in the
South Mediterranean coast of France and Corsica prevailed. The E-EU-07 flight track crossed the Po Valley and
focused on the measurement of the predicted outflow of pollution from Southern France and Barcelona into the
Mediterranean. Three shuttle flight patterns downwind from Marseille, Barcelona and close to the western coast
of Sardinia were carried out (see S9 in the supplement).
On 28 July 2017, a short wave trough with a weak cold front passed over France. This situation led to a
prevailing westerly flow and suitable conditions for the E-EU-09 flight over Southern Europe. Two shuttle flight
patterns were carried out downwind of Marseille and Barcelona. Features of interest during this flight were the
transport of the Madrid and Barcelona outflows in stratified layers into the Mediterranean and the transport of
forest fire emissions originating in Southern France and Portugal. This is described in more detail in 4.3.2.
Further details on all the flight tracks and shuttles are given in the supplement (S9).
**3.4 Model predicted pollution transport patterns**
CAMS global model data (see S3 for the model description) were used to evaluate characteristic pollution
transport patterns during the EMeRGe IOP over Europe. CAMS operational near-real time (NRT) simulations
with full emissions and chemistry were incorporated in the analysis. A stratospheric $O_3$ tracer as a proxy for
stratospheric-tropospheric transport was also used. In addition, passive CO tracers (i.e., no chemical loss or
production) provided through the CAMS field campaign support (https://atmosphere.copernicus.eu/scientific-
field-campaign-support) were used with either a) only emissions from EMeRGe target cities switched on in the
simulations (CO city tracer), or b) only BB emissions switched on in the simulations.
Figures 8, 10 and 11 show composite average maps of 12 h CAMS-global forecast for the EMeRGe flights to the
North (Flight route 2: E-EU-05 and E-EU-08) and to the South of Europe (Flight routes 1 and 3: E-EU-03, E-
EU-06, E-EU-07, and E-EU-09; see Fig.4 and Table 3 for description). The model was initialised at 00 UTC, for
the forecast at 12:00 UTC. The CO city tracer simulations at 500 and 925 hPa (see Fig. 8) indicate that the
anthropogenic MPC emissions remained close to the surface within the PBL. The emissions from the MPCs in
the North (e.g. London, Paris) are expected to be frequently transported eastwards due to the dominant west-
southwesterly winds. In contrast, emissions from MPCs South of the polar front, such as Madrid, spread in all
directions due to variable weak winds. In the highly polluted Po Valley, the emissions were transported to the
Northeast and lifted over the high mountains of the Alps.
Higher temperatures and dry conditions in Southern Europe during the EMeRGe IOP favoured $O_3$ production
and smog events. This was the case for flights to the South of Europe, as indicated by the simulations at 925 hPa
(see Fig. 8 and Fig. 11). These meteorological conditions supported the propagation of multiple and mostly
intentionally started fires in the Mediterranean area. Figure 9 shows average fire radiative power observed by
MODIS (MODerate resolution Imaging Spectroradiometer, http://modis-fire.umd.edu/) and assimilated within
CAMS-global over Europe in July 2017. In the target area, fire hot spots are visible around the Mediterranean





623 (e.g., Southern Italian Peninsula, Sicily, Sardinia, Croatia, France around Marseille, North Africa) and in

624 Portugal.

625 Further evaluation of the CAMS simulations shows that CO emitted by fires around the Mediterranean mainly

626 remained at altitudes below approximately 700 hPa. In contrast, CO resulting from the LRT of North American

627 fire emissions was observed around 500-700 hPa over Europe. The average fields show that CO from North

628 American fires was expected to be more pronounced during flights to the North (see Fig. 10), than to the South

629 (see Fig.11) with a maximum in the average fields over Great Britain.

630 The stratospheric $O_3$ tracer indicates that stratospheric intrusions over the flight domain during the campaign

631 concurred with the LRT of North American fire emissions initially lofted by warm conveyor belts or deep

632 convection. The LRT of fire emissions towards Europe is associated with mid-latitude cyclones crossing the

633 Atlantic. Dry air masses rich in $O_3$ were then transported downwards to comparably low altitudes. In the average

634 fields of stratospheric $O_3$ for flights towards the North (see Fig. 10, lower right panel), the stratospheric intrusion

635 over Europe stretches broadly from Southern Greece and Southern Italy to the Northeast. The latter is associated

636 with the cut-off low which developed on 20 July 2017 over UK and started to move eastwards on 26 July 2017.

637

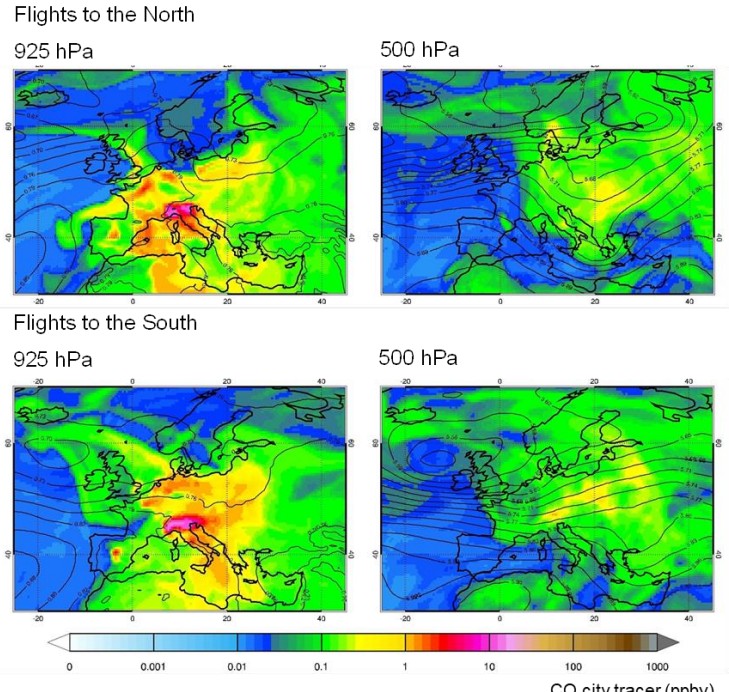

638

639 **Figure 8:** Coloured shadings of composite averages of CAMS-global city tracer forecasts of CO
640 (ppbv) at 12:00 UTC for days of flights to the North ((E-EU-05, E-EU-08, top) and South (E-EU-
641 03, E-EU-06, E-EU-07, E-EU-09, bottom) of Europe. Black contours show corresponding averages
642 of geopotential height (km) from the ECMWF-Integrated Forecasting System (IFS).

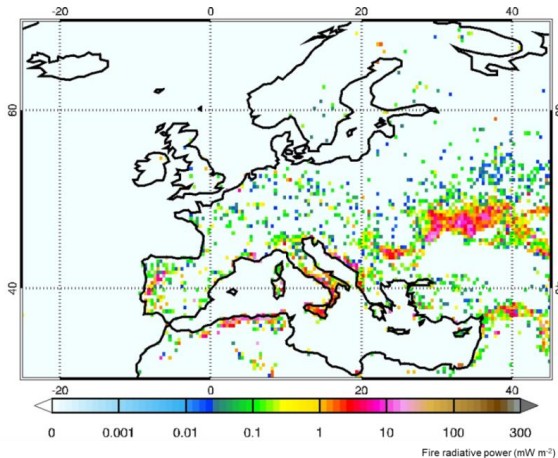

**Figure 9:** Average fire radiative power (mW m$^{-2}$) as observed by MODIS over Europe in July 2017. Data
from the CAMS Global fire assimilation system (GFAS). https://www.ecmwf.int/en/ forecasts
/dataset/global-fire-assimilation-system-gfas) fire emission database (Kaiser et al., 2012).

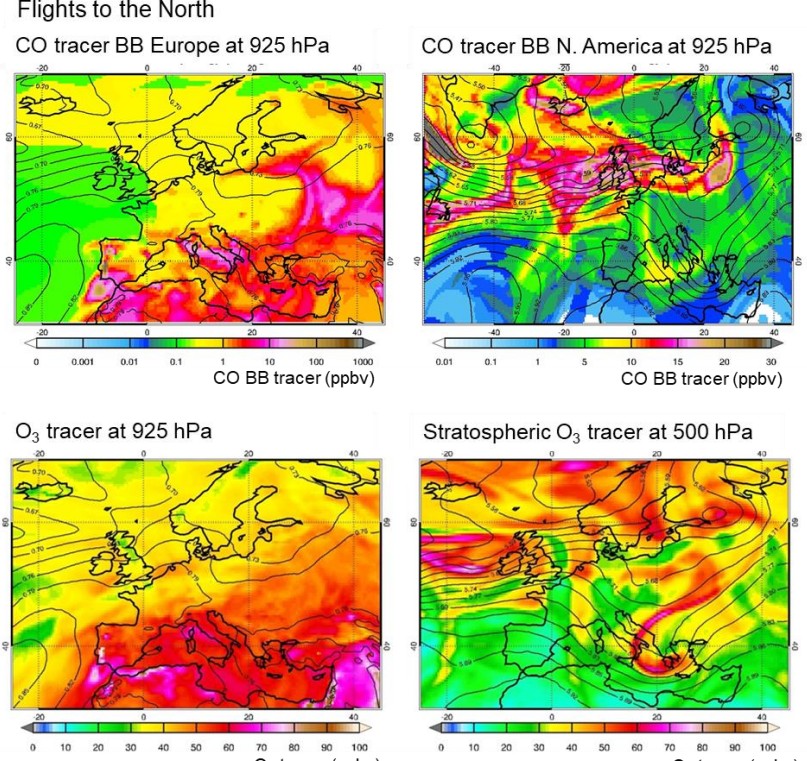


**Figure 10:** Coloured shadings of composite averages of CAMS-global forecasts at 12:00 UTC for
flights to the North (E-EU-05, E-EU-08): BB CO tracer (ppbv) from Europe (top left), and from
North America (top right) at 925 hPa; O$_3$ (ppbv) at 925 hPa (bottom left), and stratospheric ozone
tracer (ppbv) at 500 hPa (bottom right). Black contours show averages of geopotential height (km)
from ECMWF-IFS. Note the different scales. The BB tracer from North America is shown on a
larger map than the other CAMS forecasts in this image.

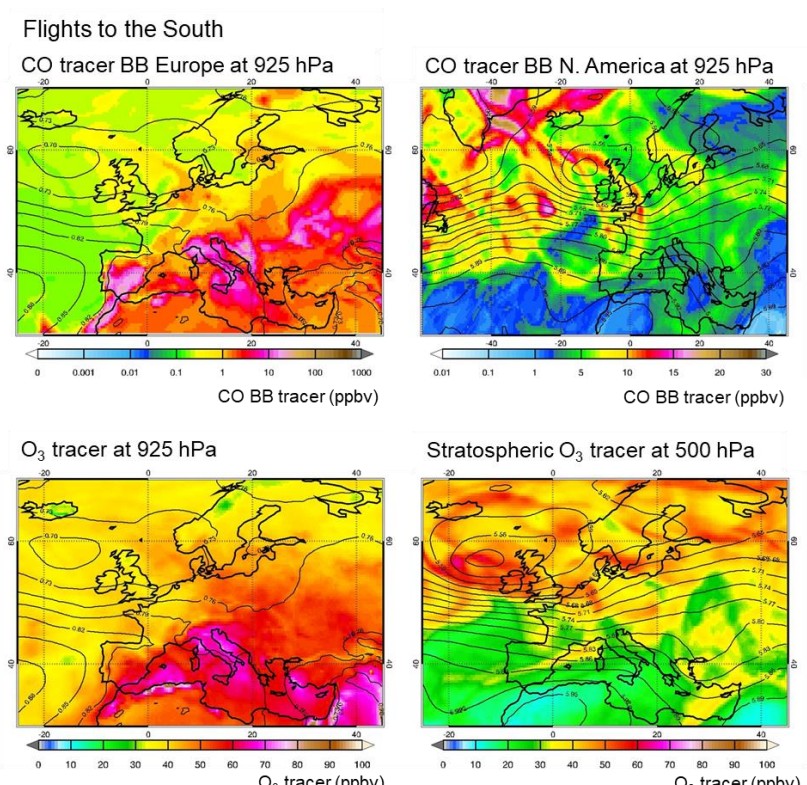

**Figure 11:** Coloured shadings of composite averages of CAMS-global forecasts as in Fig.10, for flights to the South (E-EU-03, E-EU-06, E-EU-07 and E-EU-09).

### 3.5 Measured amount and distribution of trace gases and aerosol particles

The chemical composition and the extent of photochemical activity of the air masses probed during the EMeRGe IOP were different for the different flight routes and tracks. This is to be expected as a result of the large geographical coverage of the flights, the different solar insolation conditions and the flight path of the air masses, the heterogeneous topography and the proximity of pollution sources of different types.

Table 4 shows the average, median and quartiles values of selected species measured during E-EU-08 and E-EU-06 as examples of flights in Northern and Southern Europe, respectively. The mean values and variability of most of the species are of the same order of magnitude in both flights and generally higher for E-EU-06 below 2000 m except for NO. Higher temperatures and insolation in the South are associated with higher $O_3$ and $RO_2^*$ as for example observed in E-EU-06 below 2000 m. The higher $SO_2$ and $CH_3CN$ mean values are associated to the plumes measured in the Po Valley and to the fires dominating in the South during the IOP, respectively. The average concentrations measured for the rest of the EMeRGe flights are included in the supplement (S10).

**Table 4:** Mean concentrations (mean), median (med) and quartiles ($25^{th}$ $75^{th}$) of selected measured trace gases and aerosol particles for E-EU-08 and E-EU-06 as examples of flights in Northern and Southern Europe. n.a. non-available $^x$HCHO: HCHO from PTRMS measurements; $^*$HCHO: HCHO from miniDOAS measurements; $N_{CN}$: $N_{D>250nm}$ particle with D> 10 nm, and D>250 nm, respectively (inlet cut-off 1.5 to 3 μm depending on height); BCm: black carbon mass concentration; BCn: black carbon number concentration; OA: Organic aerosol. Note that NCN, $N_D$, BCm, BCn, OA, $NO_3^-$, $SO_4^{2-}$, $NH_4^+$ and $Cl^-$ are given for standard temperature and pressure conditions.

| E-EU-08 | <2000 m | | | | 2000-4000 m | | | | >4000 m | | | | |
|---|---|---|---|---|---|---|---|---|---|---|---|---|---|
| species | mean | med | $25^{th}$ | $75^{th}$ | mean | med | $25^{th}$ | $75^{th}$ | mean | med | $25^{th}$ | $75^{th}$ | Unit |
| $O_3$ | 43 | 45 | 37 | 49 | 51 | 53 | 49 | 55 | 64 | 63 | 56 | 73 | ppbV |
| CO | 98 | 96 | 92 | 102 | 90 | 91 | 85 | 93 | 94 | 93 | 92 | 96 | ppbV |
| NO | 407 | 225 | 155 | 450 | 138 | 77 | 60 | 108 | 109 | 102 | 82 | 131 | pptV |
| $NO_y$ | 3734 | 3039 | 2075 | 4018 | 1991 | 1302 | 720 | 1777 | 4619 | 3765 | 2652 | 5761 | pptV |
| HONO | n.a. | n.a. | n.a. | n.a. | n.a. | n.a. | n.a. | n.a. | n.a. | n.a. | n.a. | n.a. | pptV |
| $NO_2$ | n.a. | n.a. | n.a. | n.a. | n.a. | n.a. | n.a. | n.a. | n.a. | n.a. | n.a. | n.a. | pptV |
| $^*$HCHO | n.a. | n.a. | n.a. | n.a. | n.a. | n.a. | n.a. | n.a. | n.a. | n.a. | n.a. | n.a. | pptV |
| $RO_2^*$ | 20 | 21 | 10 | 29 | 31 | 28 | 21 | 37 | 19 | 13 | 0 | 35 | pptV |
| $SO_2$ | 193 | 99 | 68 | 169 | 55 | 54 | 43 | 64 | 55 | 52 | 38 | 68 | pptV |
| $N_{CN}$ | 4514 | 3186 | 2066 | 4551 | 1041 | 790 | 582 | 1245 | 2900 | 1635 | 728 | 3935 | cm$^{-3}$ |
| $N_{D>250nm}$ | 119.2 | 111.5 | 61.1 | 161.1 | 18.2 | 12.3 | 6.2 | 21.8 | 7.7 | 4.4 | 2.3 | 9.2 | cm$^{-3}$ |
| BCm | 0.14 | 0.12 | 0.07 | 0.18 | 0.02 | 0.01 | 0.01 | 0.03 | 0.01 | 0.00 | 0.00 | 0.01 | μg m$^{-3}$ |
| BCn | 71 | 68 | 42 | 92 | 10 | 8 | 4 | 13 | 4 | 3 | 2 | 6 | cm$^{-3}$ |
| OA | 1.80 | 1.88 | 1.21 | 2.37 | 0.58 | 0.51 | 0.34 | 0.71 | 0.49 | 0.50 | 0.36 | 0.63 | μg m$^{-3}$ |
| $NO_3^-$ | 1.21 | 0.96 | 0.60 | 1.68 | 0.10 | 0.07 | 0.05 | 0.11 | 0.07 | 0.06 | 0.05 | 0.08 | μg m$^{-3}$ |
| $SO_4^{2-}$ | 0.85 | 0.73 | 0.56 | 0.97 | 0.20 | 0.18 | 0.13 | 0.23 | 0.09 | 0.09 | 0.07 | 0.11 | μg m$^{-3}$ |
| $NH_4^+$ | 0.80 | 0.65 | 0.46 | 1.08 | 0.16 | 0.13 | 0.10 | 0.19 | n.a. | n.a. | n.a. | n.a. | μg m$^{-3}$ |
| $Cl^-$ | 0.09 | 0.08 | 0.05 | 0.12 | 0.03 | 0.02 | 0.01 | 0.03 | 0.03 | 0.03 | 0.02 | 0.03 | μg m$^{-3}$ |
| $C_3H_6O$ | 1517 | 1543 | 1347 | 1705 | 1384 | 1404 | 1312 | 1495 | 1602 | 1614 | 1534 | 1707 | pptV |
| $CH_3CN$ | 94 | 95 | 80 | 106 | 130 | 126 | 113 | 140 | 130 | 131 | 116 | 147 | pptV |
| $C_5H_8$ | 80 | 68 | 56 | 89 | 61 | 57 | 50 | 65 | 69 | 65 | 56 | 71 | pptV |
| $C_6H_6$ | 64 | 63 | 47 | 78 | 33 | 29 | 25 | 36 | 30 | 27 | 24 | 38 | pptV |
| $C_7H_8$ | 45 | 35 | 25 | 55 | 29 | 24 | 18 | 33 | 22 | 19 | 17 | 24 | pptV |
| $^x$HCHO | 1234 | 1165 | 937 | 1461 | 642 | 637 | 538 | 733 | 411 | 407 | 290 | 496 | pptV |
| $C_2H_2O_2$ | n.a. | n.a. | n.a. | n.a. | n.a. | n.a. | n.a. | n.a. | n.a. | n.a. | n.a. | n.a. | pptV |
| $C_3H_4O_2$ | n.a. | n.a. | n.a. | n.a. | n.a. | n.a. | n.a. | n.a. | n.a. | n.a. | n.a. | n.a. | pptV |



| E-EU-06 | <2000 m | | | | 2000-4000 m | | | | >4000 m | | | | |
|---|---|---|---|---|---|---|---|---|---|---|---|---|---|
| species | mean | med | 25th | 75th | mean | med | 25th | 75th | mean | med | 25th | 75th | Unit |
| $O_3$ | 69 | 71 | 58 | 77 | 52 | 51 | 50 | 52 | 58 | 56 | 53 | 64 | ppbV |
| CO | 111 | 113 | 94 | 125 | 78 | 77 | 73 | 81 | 77 | 78 | 70 | 82 | ppbV |
| NO | 189 | 123 | 84 | 205 | 71 | 56 | 47 | 66 | 483 | 42 | 23 | 136 | pptV |
| $NO_y$ | 3321 | 2542 | 1701 | 4104 | 737 | 581 | 465 | 939 | 2006 | 366 | 283 | 490 | pptV |
| HONO | 15 | 13 | 0 | 27 | 3 | 0 | 0 | 9 | 0 | 0 | 0 | 0 | pptV |
| $NO_2$ | 454 | 378 | 238 | 531 | 169 | 174 | 115 | 199 | 191 | 172 | 43 | 303 | pptV |
| *HCHO | 1408 | 1219 | 996 | 1731 | 709 | 690 | 627 | 748 | 588 | 597 | 580 | 599 | pptV |
| $RO_2^*$ | 49 | 52 | 36 | 63 | 41 | 44 | 30 | 53 | 31 | 38 | 16 | 44 | pptV |
| $SO_2$ | 673 | 514 | 289 | 877 | 136 | 131 | 113 | 152 | 120 | 85 | 73 | 100 | pptV |
| $N_{CN}$ | 6136 | 2943 | 2052 | 4823 | 1493 | 1291 | 1147 | 1496 | 914 | 803 | 603 | 1185 | $cm^{-3}$ |
| $N_{D>250nm}$ | 174.2 | 150 | 85.8 | 224.3 | 49 | 48.5 | 41.1 | 54.9 | 22.2 | 16.3 | 7 | 30.7 | $cm^{-3}$ |
| BCm | 0.30 | 0.28 | 0.14 | 0.40 | 0.09 | 0.07 | 0.05 | 0.10 | 0.04 | 0.02 | 0.01 | 0.04 | $\mu g\ m^{-3}$ |
| BCn | 127 | 127 | 65 | 176 | 34 | 33 | 28 | 39 | 11 | 7 | 4 | 18 | $cm^{-3}$ |
| OA | 3.12 | 3.25 | 2.02 | 3.92 | 1.07 | 1.00 | 0.73 | 1.32 | 0.45 | 0.34 | 0.28 | 0.51 | $\mu g\ m^{-3}$ |
| $NO_3^-$ | 0.69 | 0.15 | 0.09 | 0.62 | 0.07 | 0.06 | 0.05 | 0.08 | 0.07 | 0.05 | 0.04 | 0.08 | $\mu g\ m^{-3}$ |
| $SO_4^{2-}$ | 1.64 | 1.49 | 0.98 | 1.93 | 0.59 | 0.61 | 0.55 | 0.68 | 0.27 | 0.20 | 0.11 | 0.44 | $\mu g\ m^{-3}$ |
| $NH_4^+$ | 0.82 | 0.67 | 0.46 | 1.04 | 0.28 | 0.29 | 0.24 | 0.32 | 0.17 | 0.17 | 0.09 | 0.22 | $\mu g\ m^{-3}$ |
| $Cl^-$ | 0.04 | 0.04 | 0.02 | 0.05 | 0.02 | 0.02 | 0.01 | 0.02 | 0.03 | 0.03 | 0.03 | 0.03 | $\mu g\ m^{-3}$ |
| $C_3H_6O$ | 2444 | 2434 | 1935 | 2937 | 1645 | 1656 | 1514 | 1799 | 1476 | 1452 | 1316 | 1605 | pptV |
| $CH_3CN$ | 140 | 131 | 115 | 152 | 129 | 131 | 118 | 138 | 135 | 132 | 123 | 145 | pptV |
| $C_5H_8$ | 98 | 78 | 59 | 112 | 62 | 57 | 50 | 64 | 73 | 67 | 55 | 83 | pptV |
| $C_6H_6$ | 109 | 94 | 56 | 152 | 36 | 34 | 25 | 41 | 32 | 30 | 22 | 37 | pptV |
| $C_7H_8$ | 57 | 42 | 25 | 77 | 35 | 25 | 22 | 51 | 32 | 30 | 26 | 37 | pptV |
| xHCHO | 1843 | 1651 | 1088 | 2374 | 891 | 875 | 748 | 993 | 641 | 616 | 491 | 782 | pptV |
| $C_2H_2O_2$ | 220 | 192 | 132 | 276 | 182 | 103 | 49 | 260 | 101 | 63 | 8 | 111 | pptV |
| $C_3H_4O_2$ | 1496 | 1275 | 1075 | 1577 | 1351 | 790 | 574 | 1622 | 817 | 571 | 296 | 756 | pptV |


The transport, transformation and radiative impact of pollutants depend on their vertical distribution. During the
EMeRGe IOP the maximum concentrations of trace gases and aerosol species were typically measured below
2000 m. Figure 12 shows the vertical distribution of CO, $O_3$, $NO_y$ and PAN mixing ratios for all HALO
observations made during the EMeRGe IOP, averaged over altitude bins of 500 m. CO, total reactive nitrogen
($NO_y$) and its most reactive forms NO and $NO_2$, are key species in the identification of anthropogenic pollution.
During daylight, NO and $NO_2$ are typically in or close to a photostationary state that is established in the order of
minutes. Further photochemical reactions convert NO and $NO_2$ into longer lived reservoirs such as PAN or
$HNO_3$. PAN has major implications for the global distributions of $O_3$ and OH as it can release $NO_2$ at higher
tropospheric temperatures far from the sources of pollution (e.g. Fischer et al., 2014). On average, changes of
CO with altitude were not pronounced except below 2000 m and above 8000 m. This is consistent with the
relatively long lifetime of CO and a well-mixed troposphere in summer. As the lifetime of $NO_y$ is much shorter
than that of CO, the distance from the source has a stronger influence on $NO_y$ than on CO observations. $NO_y$
shows a pronounced height dependence and variability which is reflected in the large standard deviations and the
differences between mean and median values (not shown). The PAN measurements made up to 3000 m altitude
have a similar behaviour. The high $NO_x/NO_y$ ratios occasionally observed at high altitudes are attributed to $NO_x$
production by lightning and more rapid transport.

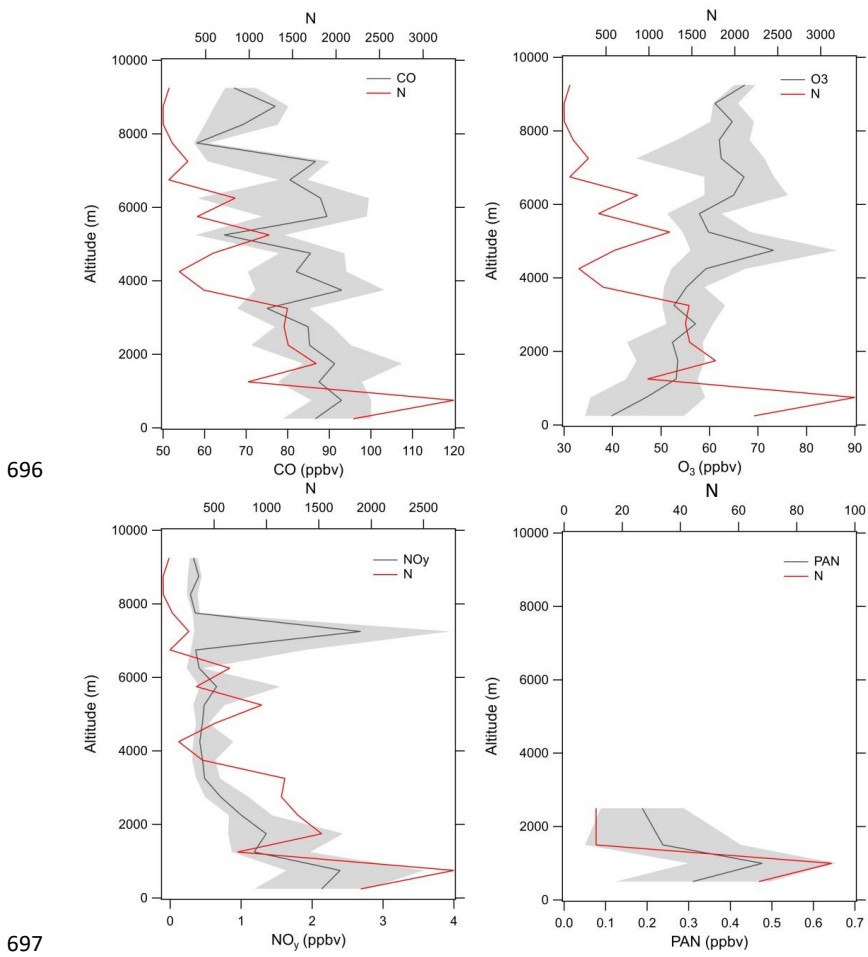



**Figure 12:** Variation of CO, $O_3$, $NO_y$ and PAN volume mixing ratios versus altitude during EMeRGe over Europe. Solid lines represent the medians averaged over altitude bins of 500 m and the shaded areas are the quartiles. The number of measuring points (N) is shown in red.

Figure 13 shows median vertical distributions of major primary and secondary VOCs observed during the EMeRGe IOP in Europe. Longer lived VOCs were well mixed in the troposphere and those with anthropogenic sources showed higher variability and highest mixing ratios below 2000 m. HCHO and acetaldehyde ($C_2H_4O$) have anthropogenic BB and significant biogenic sources. They are also generated downwind by the oxidation of transported VOCs. In contrast, benzene ($C_6H_6$) and toluene ($C_7H_8$) are primarily of anthropogenic origin. These species have a short lifetime as they are oxidised quickly in the lower layers of the troposphere. As a result, the concentrations observed above 2000 m were close to the instrumental limit of detection. The same is true for isoprene ($C_5H_8$) and xylene ($C_8H_{10}$) which have lifetimes in the order of some hours.

Acetonitrile ($CH_3CN$) and acetone ($CH_3COCH_3$) are typically well mixed in the troposphere due to their longer lifetimes, which are in the order of months. As a recognised tracer for BB, the increase of median $CH_3CN$ with altitude identifies the LRT of BB emissions from North America and the local transport of BB events in Europe.



The averaged vertical distribution of methanol ($CH_3OH$), having ~ 12 days lifetime, might result from the
convective mixing of a variety of ground sources which in the summer are largely of biogenic origin.
Known sources of glyoxal ($C_2H_2O_2$) and methylglyoxal ($C_3H_4O_2$) are the oxidation of $C_5H_8$ and BB. $C_2H_2O_2$ is
also an oxidation product of acetylene ($C_2H_2$) which is of anthropogenic origin. $C_3H_4O_2$ is produced in the
oxidation of $CH_3COCH_3$, which is thought to have a dominant biogenic source (Andreae, 2019; Wennberg et al.,
2018). Both gases are also formed during the oxidation of other VOCs, particularly alkenes, aromatics, and
monoterpenes (Myriokefalitakis et al., 2008; Fu et al., 2008; Taraborrelli et al., 2020) and are present both as
primary or secondary pollutants during BB events (e.g., Vrekoussis et al., 2009; Alvarado et al., 2020).

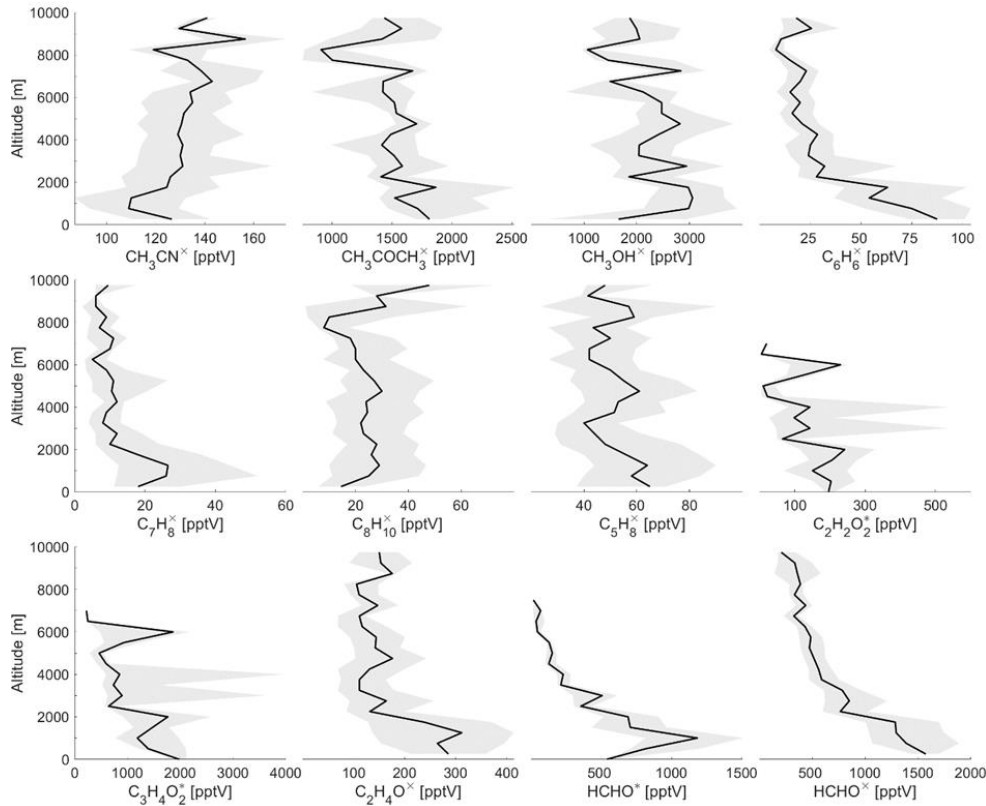


**Figure 13:** Variation of VOC versus altitude measured by the HKMS (labeled with [x]) and the miniDOAS (labeled with [*])
instruments during EMeRGe over Europe. Shaded areas are the quartiles, solid lines represent median concentrations.

The HCHO mixing ratios measured by the in-situ PTRMS (HKMS) and the remote sensing miniDOAS
instruments during the IOP in Europe are consistent with previous remote sensing observations over South East
Asia (Burrows et al., 1999) and North America in summer (Kluge et al., 2020; Chance et al., 2000; Dufour et al.,
2009; Boeke et al. 2011; De Smedt et al., 2015; Kaiser et al., 2015; Chan Miller et al., 2017, and references
therein). They are also in the same range as those measured in the Po Valley (Heckel et al., 2005).
The HCHO mixing ratios observed in the PBL and middle troposphere during EMeRGe are somewhat lower
than the North American mixing ratios (see Fig. 14). This might be related to the fact that several EMeRGe flight
tracks were carried out far from emission sources over the North and the Mediterranean Seas. In addition, the





emissions of HCHO and its VOC precursors have been reported in previous studies to be lower in Europe than in
North America (e.g. Dufour et al., 2009; De Smedt et al., 2015).

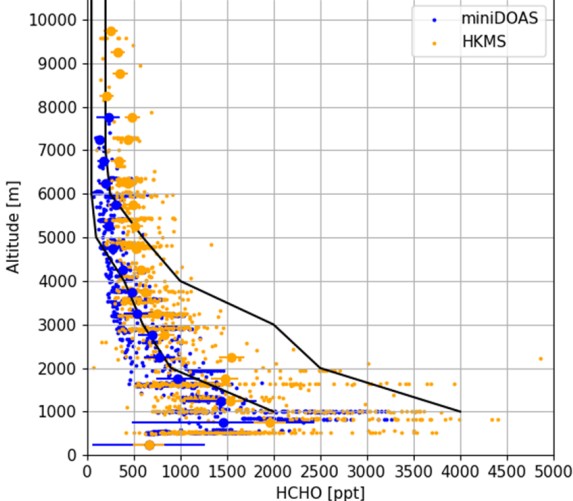


**Figure 14:** HCHO measurements by the HKMS (in orange) and the miniDOAS instruments (in blue). Mean values
(bigger dots) and the respective accuracies (horizontal bars) are also shown. The black lines indicate the range of
previous HCHO measurements over North America in summer (Kluge et al., 2020). Note that HKMS and miniDOAS
agree within their accuracies in spite of having different air sampling volumes, which did not perfectly overlap.

The vertical profiles shown in Fig. 14 are averages from the measurements taken along all flights at variable
distances from various source regions and under different meteorological conditions. In a next step, pollution
hotspots are identified by using the spatial distribution of trace gases and aerosol particles observed over the
flight tracks.
Figure 15 shows as an example the CO NO, $O_3$, $CH_3COCH_3$, $CH_4$, and the organic aerosol mass concentrations
measured during the EMeRGe flights in Europe.. A detailed analysis of the complexity of the air masses
measured and the variations encountered in individual flights is beyond the scope of the present work and will be
presented in dedicated publications.




**Figure 15:** Mixing ratios of CO NO, O₃, CH₃COCH₃, CH₄, and organic aerosol mass concentrations measured along all EMeRGe flights in Europe. To increase colour contrast, 50 ppbv has been set as lower limit for CO, and 0.5 ppbv and 80 ppbv as upper limit for NO and O₃ respectively. These limits are representative for more than 95% of all measurements. CH₄ mixing ratios are in 0.05 x 0.05° bins as in Klausner (2020). Organic aerosol mass concentrations are plotted for the original time resolution of 30 sec. Note that mixing ratios measured at different altitudes in the shuttle areas are not distinguishable in the figure.

During the EMeRGe IOP in Europe, the highest NO concentrations were found in the vicinity and downwind of

major pollution sources like London, the BNL/Ruhr region and the Po Valley. High NO concentrations are





indicative of recent or "fresh" anthropogenic emissions. The $NO_y$ lifetime of a few days enables a more reliable
identification of aged polluted air masses further out from the source regions. Maximum $NO_y$ values as large as
12 ppbv were measured. Elevated CO and $NO_y$ accompanied by low NO, as measured in the proximity of
Barcelona, indicate that there has been a significant amount of processing of the pollution plumes sampled.
Emission hot-spots can be hardly identified in the spatial distribution of $O_3$ as expected from its non-linear
secondary formation. Maximum $O_3$ mixing ratios were generally observed at a distance downwind of MPCs,
determined by $O_3$ production and loss in the plumes.
Organic aerosol has strong anthropogenic sources such as combustion (traffic, fossil fuel combustion, BB) and
industrial activity, and shows similar behaviour to CO and NO, in that larger mass concentrations are closer in
time and space to MPCs such as London, Po Valley, and BNL. The lifetime of aerosol particles in the PBL is a
few days, which explains the high variability observed. Additionally, aerosol particle concentrations have a
strong gradient above the PBL (see Sect. 4.1). As a result, the flight shuttles at different altitudes have large
variability in the horizontal distribution.
The highest and most distinctive $CH_4$ mixing ratios in the PBL were likewise encountered in the Po Valley (up to
2.4 ppm), downwind of London and across the BNL/Ruhr region (up to 2 ppm). Slightly lower mixing ratios
were detected downwind of Barcelona (up to 1.94 ppm). The mixing ratios were higher than the global mean
ground level mixing ratio of around 1.85 ppm for July 2017. The emission plume signatures were generally more
evident when shuttles were performed close to the respective MPC regions. At large downwind distances the
$CH_4$ emissions are diluted and/or mixed with pollution from surrounding sources. For the assignment of the
GHG enhancements to their source region, supporting model simulations and complementary measurements of
shorter-lived species with smaller background concentrations and thus better signal-to-background ratios are
needed (Klausner, 2020).
The distribution of highly reactive species such as peroxy radicals, during the flights is determined by the rates
of photochemical production and loss of $HO_2$ and organic peroxy radicals $RO_2$. The $RO_2^*$ measured is the sum of
$HO_2+ \sum RO_2$, R being an organic chain which produces $NO_2$ in its reaction with NO. Oxygenated VOC (OVOC)
result from the oxidation of VOC emissions (e.g. $CH_3COCH_3$ or HCHO) and are strong sources of $HO_2$ and
$CH_3O_2$. The $RO_2^*$ mixing ratios observed in EMeRGe are shown in Fig. 16. Mixing ratios up to 120 pptv $RO_2^*$,
3 ppbv of $CH_3COCH_3$ and 4 ppbv of HCHO were measured in the air masses probed. Provided insolation
conditions (i.e. actinic fluxes) and amount of precursors are similar, the production of peroxy radicals is
observed as long as plumes mix at any altitude. Generally, higher $RO_2^*$ were measured below 45°N. This is in
part due to the higher insolation during the flights over the Mediterranean area, which accelerates photooxidation
and the production of $RO_2^*$. The $O_3$ production rates calculated from the $RO_2^*$ measured on-board are consistent
with the values reported in urban pollution for NO<1 ppbv (e.g. Tan et al, 2017; Whalley et al, 2018, 2021). The
photochemical activity of the air masses has been studied using the $RO_2^*$, the trace constituents and photolysis
rates measured during the EMeRGe IOP (George et al., 2021, in preparation).

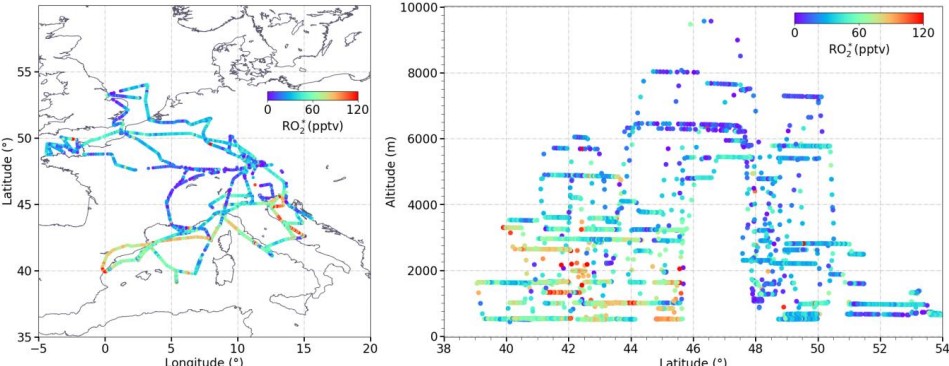


**Figure 16:** $RO_2^*$ spatial and vertical distribution measured along all EMeRGe flights in Europe.

The identification of MPC outflows and the investigation of the pollution events benefits from knowledge of the
mixing of anthropogenic, natural and biogenic sources during the EMeRGe flights. The curtain maps showing
the latitudinal and vertical distributions of selected species help to classify the air mass mixtures, especially in
the lower 2000 m of the troposphere. Differences observed North and South of the Alps are evident in Fig. 17,
showing a reasonable agreement between the vertical distributions of CCN and CO which has been documented
in earlier studies (e.g. Pöhlker et al. 2016, 2018).
The vertical and latitudinal distribution of the cloud condensation nuclei number concentration ($N_{CCN}$) shows a
strong vertical gradient. Generally, $N_{CCN}$ is highest in and above the PBL, up to ~2000 m a.s.l. The $N_{CCN}$ depend
strongly on the particular air mass, its photochemical history and the source of pollution as shown in Fig. 17b. In
Northern Europe, (50 to 55 °N), $N_{CCN}$ up to 1200 cm$^{-3}$ were measured in the London outflow over the North Sea
and over the BNL/Ruhr region. Below 46 °N, $N_{CCN}$ often exceeds 1500 cm$^{-3}$ above the MPC in the Po Valley,
Rome, Marseille and Barcelona, the highest concentrations being observed in the Po Valley.
An interesting observation was the distinct layer of BB smoke measured above the PBL between 2000 and 3500
m altitude, close to Marseille and Barcelona (40 to 42 °N). The high $N_{CCN}$ due to BB are episodic in nature,
whereas the CCN emissions from anthropogenic activity are produced daily with probably a weekend
modulation.. The vertical profile in Fig. 17b is a composite of all data but clearly shows that altitudes below
2000 m have the highest $N_{CCN}$. The peak between 2000 and 4000 m is associated with air masses, which either
come from BB events upwind and flow into the Mediterranean, or are Po Valley air being lifted up the Alps.

**a)**

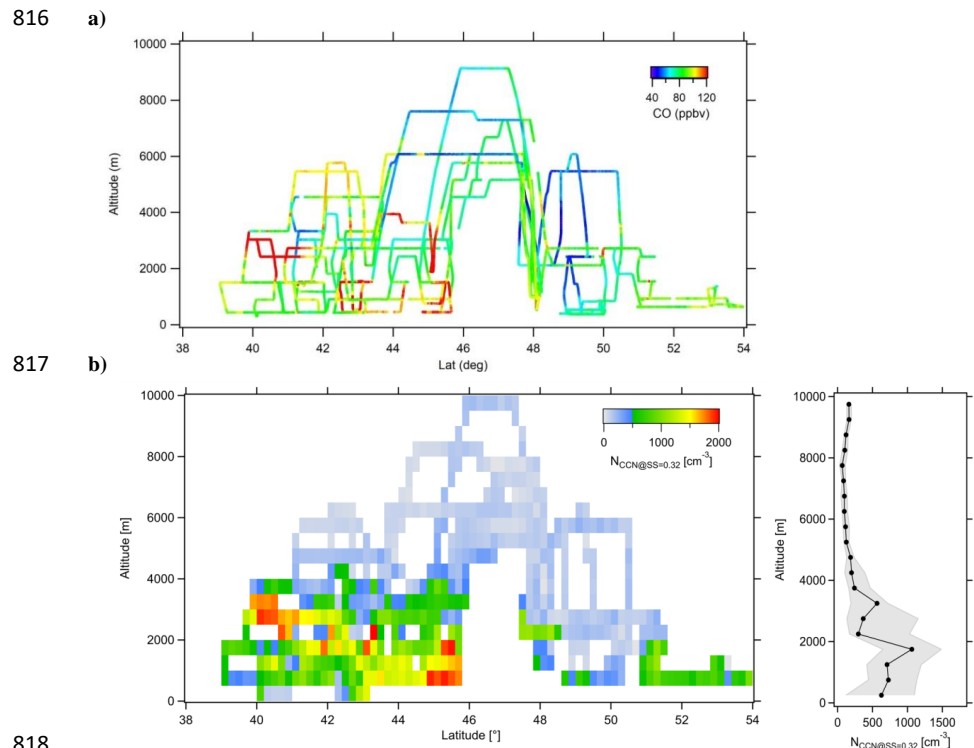

**b)**

**Figure 17:** Vertical and latitudinal distribution observed during the EMeRGe IOP of a) CO mixing ratios, and b) CCN number concentration at a supersaturation ($S$) of 0.32 % (except for E-EU-04, due to instrumental failure). The CCN curtain plot on the left is made with latitude- (0.2°) and altitude-binned (500 m) CCN number concentrations. On the right, the median vertical $N_{CCN}$ ($S$=0.32 %) profile is represented by a solid black line and the interquartile range by a grey shaded area. CCN data is STP corrected.

### 4 Identification of pollution outflows within the EMeRGe IOP in Europe

The investigation of transport and transformation of MPC outflows over Europe benefits from the unambiguous identification of individual MPC sources. With this objective, a series of complementary plume tagging or identification approaches were used in the EMeRGe IOP in Europe:

I) Enhancement in the concentration of selected atmospheric species

Periods in which large pollution plume events were measured on-board HALO were initially categorised into the following: a) anthropogenic pollution (AP), b) biomass burning (BB) and c) mixed plumes, by using the presence and enhancements of VOCs in these plumes, which are characteristic for different sources. For example, $CH_3CN$ is almost exclusively emitted from BB (de Gouw et al., 2003; Warneke et al., 2010) whereas $C_6H_6$ is emitted by traffic and petroleum- related industrial activities (Paz et al., 2015) as well as BB (Simpson et al., 2011; Andreae, 2019). Hence, $C_6H_6$ enhancements in the absence of $CH_3CN$ can be used to identify relatively "pure" anthropogenic pollution. Similarly, $CH_3CN$ enhanced plumes in the absence of $C_6H_6$ are identified as pure or aged BB events. Events with only $CH_3CN$ can originate from mixed sources, as $C_6H_6$ may have decayed while $CH_3CN$ remains, due to the different atmospheric lifetimes of these two tracers ($CH_3CN$ ~ 6 month, $C_6H_6$ ~



10 days). When both VOCs are enhanced, the plumes are considered to have air masses from either BB and AP
sources or only from recent BB. Additionally, enhanced $C_5H_8$ as short-lived biogenic tracer is used as an
indicator for recent contact with the PBL having biogenic sources (Förster et al., 2021, in preparation).
These large categorised pollution events were then further classified into single plumes by using altitude, water
content, wind direction and enhancements in the concentrations of pollution tracers such as CO and $NO_y$
measured on-board HALO. Fine structures or signatures in individual plumes were numbered relative to the
main plume event they belong to.
All plumes encountered are numbered using the notation E-EU-FN-S-PL similarly to the flight nomenclature
mentioned in Sect. 2.3, i.e., E stands for EMeRGe, EU for the campaign in Europe, FN are 2 digits for the flight
number, S is the letter assigned to the identified captured pollution event, and PL are two digits reserved for the
plume number within each pollution event.
II) Backward trajectories: last contact with PBL
The origin and history of the plumes probed at each point of the flight track are traced by using highly-resolved
backward trajectories calculated by the kinematic trajectory model FLEXTRA (Stohl et al., 1995, 1999).
Parameters calculated using FLEXTRA and meteorological fields are used to assign the origin of the observed
plumes to the EMeRGe targets in different parts of the flight tracks. Typically, the last contact to the PBL
(lcPBL), i.e., the time when the backward trajectory reaches the PBL the first time, and sensitivity trajectories
which provide the probability of contact of a particular air mass with the lower meters of the PBL before the
measurement are used. This information is cross-checked with the estimated age of air masses based on
HYSPLIT CO dispersion calculations in III). More details about trajectories and related parameters are given in
S11 in the supplement.
III) Forward trajectories: dispersion of MPC outflows
In a similar approach to that used in the forecast procedures (see Sect. 2.3 and S3 in the supplement), the
HYSPLIT dispersion model was used to calculate the dispersion of CO emissions using emission rates from the
EDGAR HTAP V2 emission inventory. They are expressed as CO enhancement caused by the selected MPC
outflow over the CO background. The performances of FLEXPART and HYSPLIT for the EMeRGe data are
compared for the case studies within EMeRGe.
IV) Detection of released PFC tracers
Sampling of PMCH from a tracer release in the centre of London during E-EU-05, and from a tracer release in
the centre of London and at the University of Wuppertal during E-EU-08, enabled the prediction of the
dispersion and the mixing of the targeted MPC outflows in these flights to be compared. Details on the tracer
experiments during the EMeRGe IOP over Europe are described in Schlager et al. (2021 in preparation).

**4.1 Characterisation of polluted air masses by using chemical tracers**

Initially, as described in I) in the previous section, in-situ measurements of $C_6H_6$ and $CH_3CN$ on-board HALO
(Förster et al., 2021, in preparation) were used to identify measurements of unpolluted background air (absence
of both tracers) and of anthropogenic polluted air masses (enhancement of $C_6H_6$ and absence of $CH_3CN$).





In Fig. 18, the HCHO measured by the miniDOAS and HKMS instruments on board is shown. In the air masses
classified as polluted the HCHO results from direct emission and oxidation of VOC precursors and is discernibly
higher than the lower boundary of the measurements. The HCHO in the less polluted or background air in
Europe is then attributed to be predominantly released from $CH_4$ oxidation.

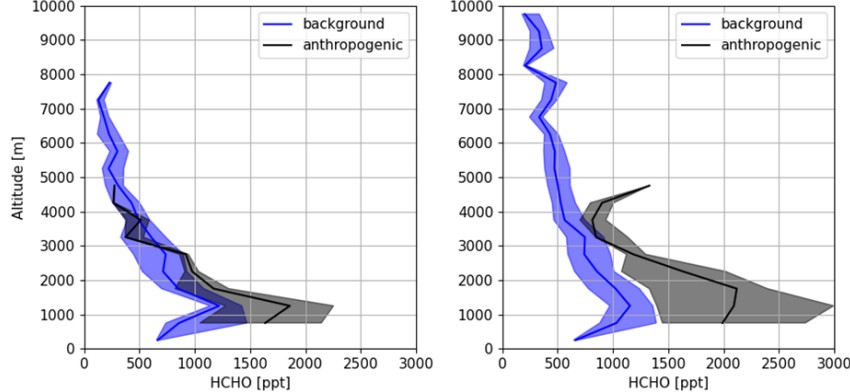


**Figure 18:** Vertical profiles of HCHO (miniDOAS left, HKMS right) for pure anthropogenic emissions ($C_6H_6$
enhancement in absence of $CH_3CN$) and background air (in the absence of $C_6H_6$ and $CH_3CN$). Shaded areas are the
quartiles, solid lines represent median concentrations.
In a similar manner, the aerosol particle concentration and composition have been tagged for anthropogenic and
background air masses (see Fig. 19).

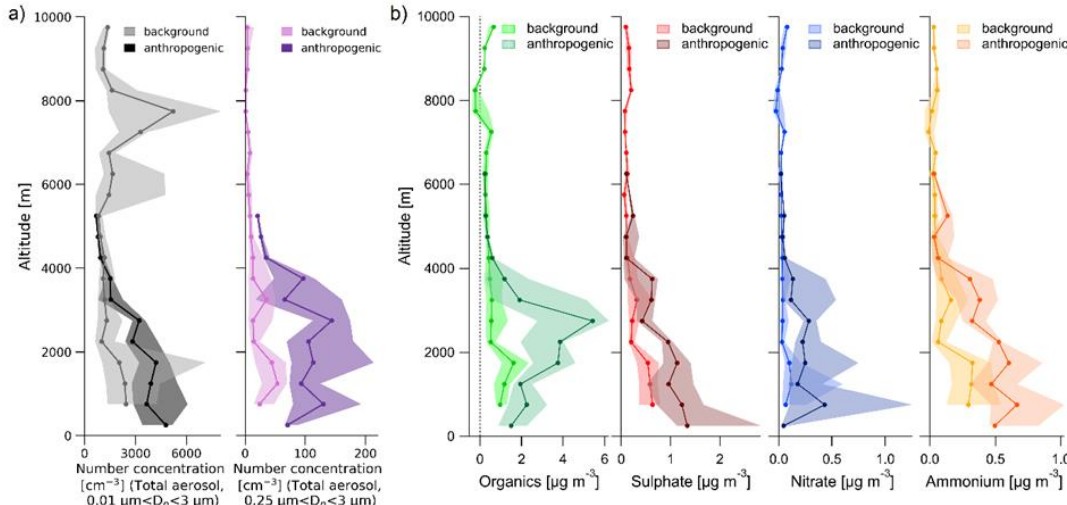


**Figure 19:** As in Fig. 18, for a) the total aerosol number concentrations for two different size ranges (0.01-3 µm and 0.25-3
µm) and b) organic, sulphate, nitrate and ammonium mass concentrations in the aerosol particles. The dots in the solid lines
represent the medians averaged over altitude bins of 500 m and the shaded areas are the quartiles.





893 In the vertical distribution of the total aerosol number concentrations (Fig. 19a), the difference between
894 anthropogenic and background air masses is more pronounced in the size range between 0.25 μm and 3 μm than
895 in the size range between 0.01 μm and 3 μm. At altitudes below 4000 m the averaged total aerosol number
896 concentrations show several maxima which are mainly caused by local pollution plumes. In contrast to all other
897 profiles, there are two additional maxima in the number concentration compared to background aerosol for the
898 size range 0.01 μm to 3 μm at around 6000 m and 7500 m. These maxima are not apparent in the profiles of
899 particle larger than 0.25 μm. This is consistent with the attribution of LRT of air masses from North America,
900 where they had contact with BB emissions. New particle formation events cannot be excluded but are considered
901 unlikely.

902 The vertical profiles of the chemically resolved aerosol mass concentrations in Fig. 19b clearly show the
903 enhanced concentrations in the anthropogenically influenced air masses compared to the background air masses.
904 Differences in the median vertical profiles of the inorganic and organic aerosol suggest that organic aerosol in
905 anthropogenic air masses is mainly formed by secondary processes. As a result of the time required by the
906 emitted precursor VOCs to be converted into secondary organic aerosol, the anthropogenic organic aerosol
907 concentration increases above 2000 m altitude. In contrast, the inorganic components of the aerosol, especially
908 ammonium and sulphate ions, show a steady decrease in the anthropogenically influenced air masses until up to
909 about 4000 m. Above that altitude, the difference between background and anthropogenic profiles becomes
910 small for both organic and inorganic aerosol components. This is a very interesting finding, implying that the
911 direct influence of anthropogenic emissions on the aerosol of the free troposphere over Europe is small.

912 Additional information is provided by the vertical distribution of carbon isotope ratios obtained from whole air
913 samples taken on HALO and at the ground sites in London, Wuppertal, Milan and Rome. The $\delta^{13}$C values in
914 pentanal ($C_5H_{10}O$) and $C_6H_6$ shown in Fig. 20 are colour coded according to the different areas sampled, as given
915 in the overview map in Fig. 4. In general, the $\delta^{13}$C values are in the expected range reported by previous studies
916 (e.g. Rudolph et al., 2000; Goldstein and Shaw, 2003).

917 The air samples taken during the EMeRGe IOP at ground stations exhibited different features in $\delta^{13}$C values for
918 the Southern and for the Northern European MPCs. In general, lower $\delta^{13}$C values for $C_5H_{10}O$ and $C_6H_6$,
919 indicative of fresh emissions, were observed below 2000 m altitude. On average, $C_5H_{10}O$ is less enriched in $^{13}$C
920 in the Rome and Milan (-32.6 ‰) than in the London and Wuppertal samples (-31.4 ‰), whereas it is the
921 opposite for $C_6H_6$, i.e., (-27.3 ‰) and (-29.0 ‰), respectively. Moreover, the $\delta^{13}$C ground values in Italy indicate
922 more constant sources in $C_5H_{10}O$ and $C_6H_6$ as in the Northern MPCs, as is apparent from the standard deviations
923 of 0.8 ‰ and 0.7 ‰ in contrast to 1.2 ‰ and 3.3 ‰, respectively.

924 The EMeRGe flights to the Southern MPCs in Europe covered a larger altitude range than the flights to the
925 Northern MPCs. The upwind and downwind shuttles at different flight altitudes of the Rome MPC illustrate a
926 general increase in $\delta^{13}$C in $C_5H_{10}O$ and $C_6H_6$ with increasing altitude. This implies that chemically processed air
927 was encountered during the transits over the Apennines. In comparison to $C_5H_{10}O$, the enrichment in $^{13}$C with
928 altitude in $C_6H_6$ is not very pronounced. This is consistent with the longer lifetime of $C_6H_6$ and a well-mixed
929 troposphere with a variety of ground sources mixed by convection in summer. Consequently, the values for $\delta^{13}$C
930 in $C_5H_{10}O$ represent local conditions, whereas those in $C_6H_6$ provide regional or LRT information. The isotopic
931 signatures reveal a second layer with rather fresh emissions in the altitude region between 2000 and 3000 m
932 which extends to 4000 m in the Southern MPCs (e.g. Rome and Po Valley). These observations are  consistent
933 with the trace gases and aerosol measurements.



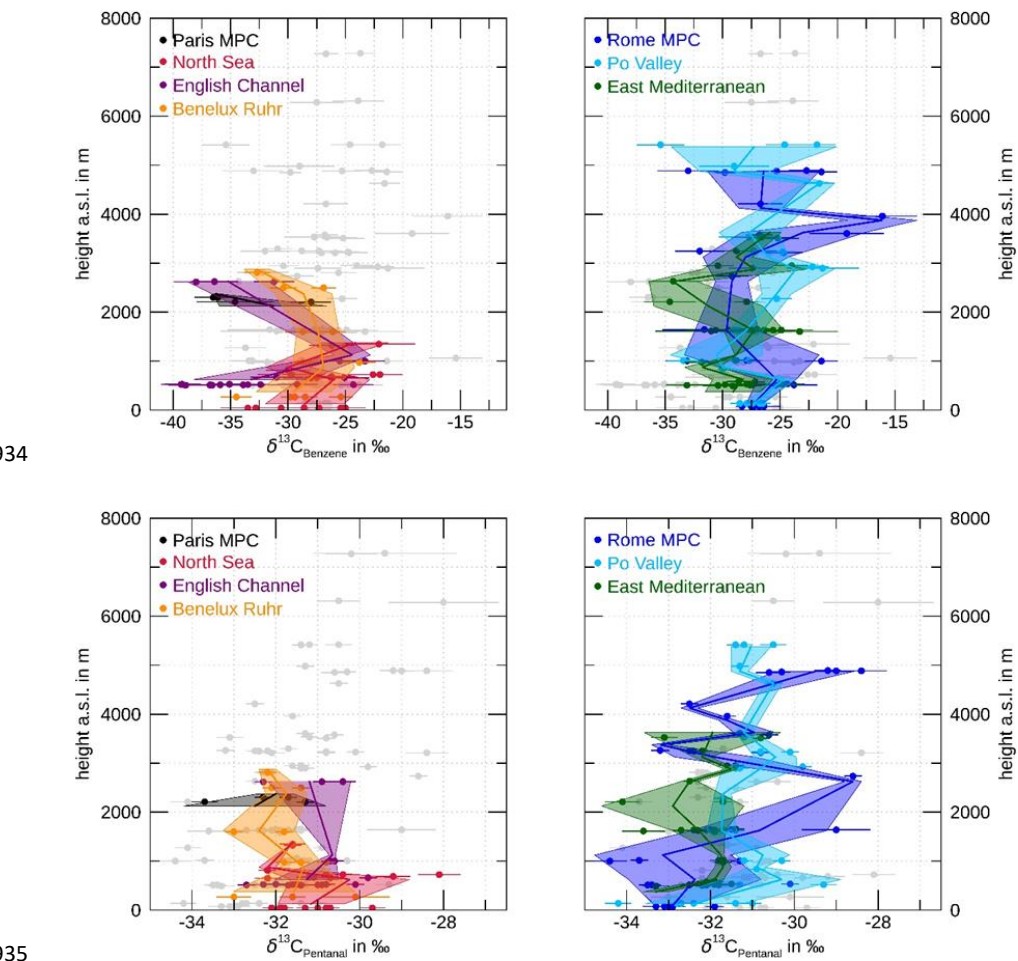

934

935

**Figure 20:** Vertical distribution of $\delta^{13}C$ values in $C_5H_{10}O$ (left) and $C_6H_6$ (right) in whole air samples taken on HALO and at the ground sites in London, Wuppertal, Milan and Rome. Data for northbound flights (left column) are colour coded for Paris MPC (black), North Sea (red), English Channel (violet), BNL/Ruhr (orange). Data for southbound flights (right column) are colour coded for Rome MPC (blue), Po Valley MPC (cyan) and East Mediterranean (green). The coloured shadings refer to the standard deviation of $\delta^{13}C$ values in altitude bins of 250 m. Mean $\delta^{13}C$ values of the respective altitude bins are represented as solid colour-coded lines. The $\delta^{13}C$ values at the lowest altitudes in each colour represent the results of air samples at the ground stations: London (red), Wuppertal (orange), Rome (blue) and Milan (cyan). Error bars in $\delta^{13}C$ are given for each sample value.

Typically, plumes of anthropogenic and biogenic origin were mixed in the air probed over Europe. The EMeRGe IOP was characterised by the contribution of fresh wildfires in the Mediterranean area, which add BB signatures to the probed air masses, and mixed with anthropogenic plumes as indicated by VOCs and in particular by the $CH_3CN$ observations. For particles emitted from BB, a frequently used tracer is levoglucosan which is identified using the m/z 60 ion ($C_2H_4O_2^+$) in aerosol mass spectrometry (Schneider et al., 2006; Alfarra et al., 2007). The photochemical degradation of levoglucosan is fast in summer (Hennigan et al., 2010, 2011; Lai et al., 2014), and the BB aerosol observed during the IOP in Europe flight tracks was generally processed too fast to be distinguished from other secondary aerosol.

A more robust indicator for particles from BB is BC. BC particles are formed in processes of incomplete
combustion, and therefore are an important component of both BB and urban aerosol particles (Bond et al.,
2013). The microphysical properties of BC give insights into the combustion sources and atmospheric ageing
time of the pollution plumes (Liu, 2014, Laborde, 2012, Holanda et al., in preparation 2021). Figure 21 shows
average BC mass size distributions for different plumes encountered during the E-EU-06 flight (anthropogenic,
BB, and mixture). The plumes were classified according to the VOC observations as described in I) in Sect. 4.
Larger BC cores were found in pure BB plumes and mixed BB and AP plumes, with mean modal diameter ($D_c$)
of 200 and 210 nm, respectively. Smaller BC cores, with mass size distribution peaking at $D_c = 170$ nm, were
found in urban pollution, as a result of the different fuel burnt and combustion conditions. These values obtained
during EMeRGe are consistent with previous aircraft observations for urban and BB plumes (Schwarz et al.,
2008; Laborde et al., 2013). During E-EU-06, the average total BC mass concentration was also substantially
higher in BB and mixed BB (0.61 ± 0.12 µg m$^{-3}$ and 0.81 ± 0.35 µg m$^{-3}$, respectively) than in urban pollution
(0.35 ± 0.15 µg m$^{-3}$).

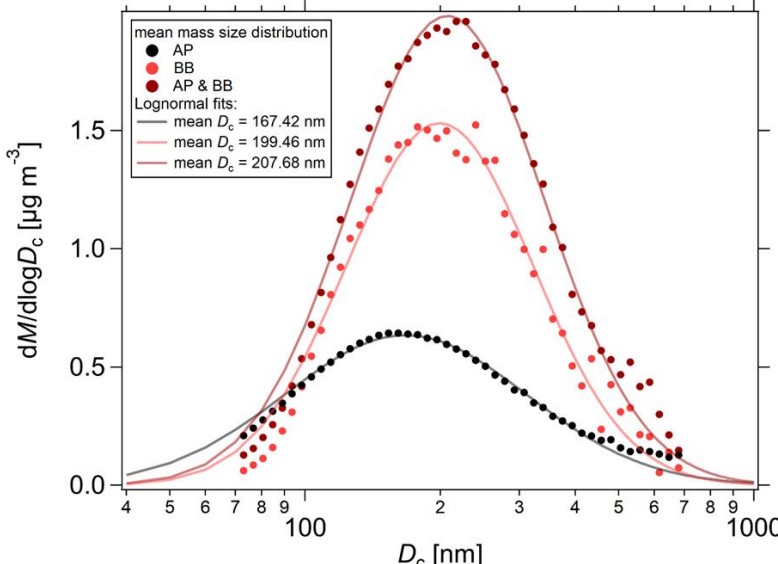


**Figure 21:** Mean mass size distribution of black carbon particles measured in anthropogenic pollution (AP, black),
BB (light red), pollution from anthropogenic/ BB mix (AP & BB, dark red) during E-EU-06 on 20 July 2017.
Lognormal fits were applied to the mean size distributions for $100 < Dc < 300$ nm.
**4.2 Identification and classification of MPC outflows: London**
The flight E-EU-08 on 26 July 2017 has been selected to illustrate the procedure for the identification and
classification of air mass origin and the different source contributions to the plumes. As briefly described in Sect.
3.3, the E-EU-08 investigated the London and BNL/Ruhr MPC outflows. HYSPLIT dispersion calculations of
the CO city plumes were used to define the location of the outflows, which were measured along the Eastern UK
coast between 10 and 12 UTC and over the European continent between 13:20 and 14:15 UTC approximately.
Cloudy and rainy conditions prevailed throughout the flight  reduced flight visibility and limited further tracing
of the BNL/Ruhr outflow over Germany in the afternoon. However, the PMCH was observed from the two
releases showing the success of this technique and the adequacy of the description of the transport in HYSPLIT.

### 4.2.1 Identification of pollution plumes


Figure 22 shows the time series of $C_6H_6$ and $CH_3CN$, their enhancements colour-coded on the altitude and the
identified plumes along the flight by using the time series of CO and $NO_y$, as described in I) in Sect. 4. Figure 23
summarises the result of applying the tagging tools II) and III) to the E-EU-08. Overall, the HYSPLIT dispersion
and FLEXTRA backward calculations agree reasonably in identifying fresh emitted London plumes such as B-
02 and B-04: the measured 22 and 19 ppbv CO increases over background are estimated by HYSPLIT as 25 and
22 ppbv (sum of all transport times). B-05 is a good example of significant mixing with aged plumes (12-24 h)
which seem to dominate in B-06 and B-08 (see detail in Fig. 23). Plume B-09 is a good example of mixing of
freshly emitted plumes from BNL/Ruhr (0-6 h) and aged emissions (>24 h) of London origin. The PFC tracer
measured on-board is also depicted in Fig. 23. For B-02, B-04 and B-05, enhanced PMCH volume mixing ratios
above the 8.5 ppqv atmospheric background in Europe were clearly detected.

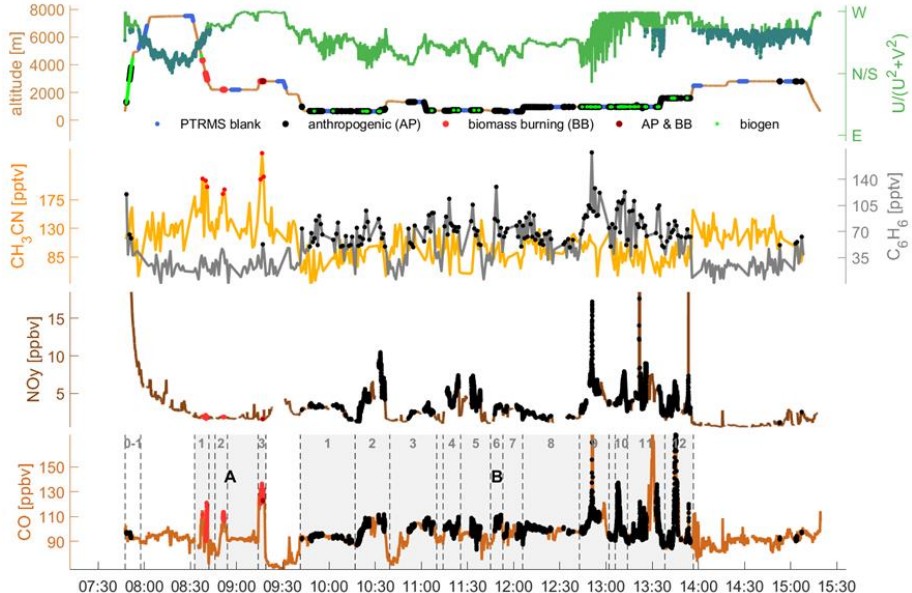


**Figure 22:** Time series for E-EU-08 on the 26 July 2017 used for the categorisation of plumes based on VOC measurements: altitude, wind direction, $CH_3CN$, $C_6H_6$ and $NO_y$ as refinement. The wind direction is given as $U/(U^2+V^2)$, -1 is east wind, +1 is west wind, values around zero have North or South components. South components are marked with dark green colour. Altitude is colour-coded in light-green during $C_5H_8$ enhancements, in light red during $CH_3CN$ enhancements, in black during $C_6H_6$ enhancements and in dark red during both, $CH_3CN$ and $C_6H_6$ enhancements. Additionally, blue colour-coded blank measurements of $CH_3CN$, $C_6H_6$ and $C_5H_8$ are given. Final numbering of structures and plumes according to concentration enhancements are shown for CO. Colour-coding indicates $CH_3CN$ enhancements (light red), $C_6H_6$ enhancements (black), and both, $CH_3CN$ and $C_6H_6$ enhancements (dark red).

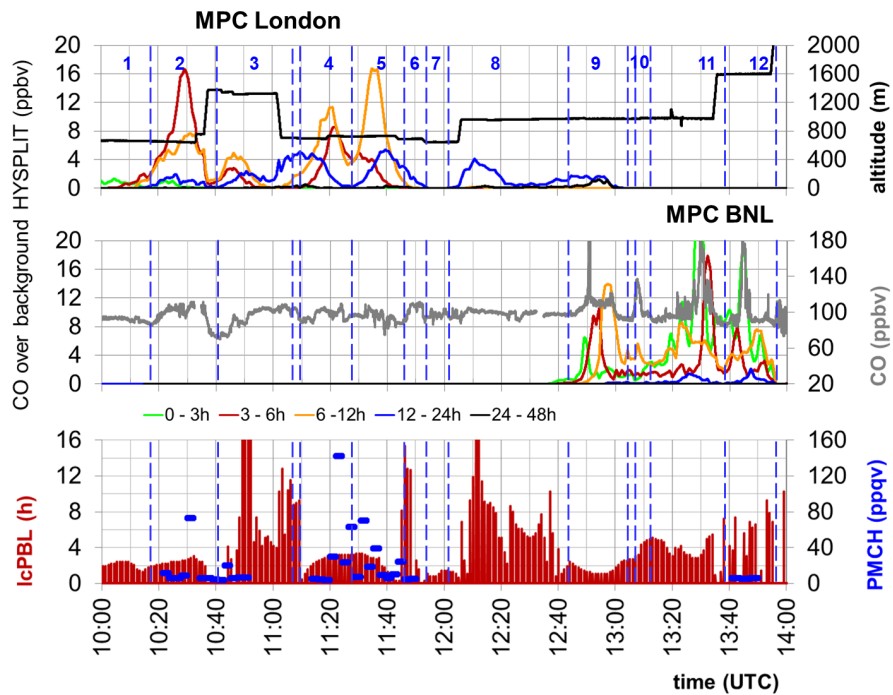

**1000**

**Figure 23:** Detail of the MPC outflow of London (B-01 to B-09) and BNL/Ruhr (B-09 to B-12) probed with HALO along
the E-EU-08 flight track. Numbering in blue corresponds with the classification in Fig. 22 (B is omitted for simplicity). The
position of the plumes is also indicated by the blue lines. Dispersion of CO emissions of target MPCs and the transport time
of the air mass calculated by HYSPLIT are depicted. The last contact with the PBL (lcPBL) calculated using FLEXTRA is
also shown. Elevated PMCH mixing ratios were measured for B-02, B-04 and B-05.

The plumes identified using I), i.e., enhanced concentrations of mixing ratios of selected atmospheric species,

and the MPC assigned outflow with the estimated air-mass transport times are summarised in Table 5. These

plumes show mixtures of anthropogenic pollution (AP), BB and biogenic emissions (BIO).





**Table 5:** Synopsis of identified structures (A and B) and plumes with anthropogenic (AP), biomass burning (BB) and biogenic signatures (BIO), MPC assignments and estimated transport times (Ttime) based on HYSPLIT and FLEXTRA for E-EU-08.

| Notation | begin [UTC] | end [UTC] | signature | MPC origin | Ttime [h] |
|---|---|---|---|---|---|
| E-EU-08-0-01 | 07:47:34 | 07:57:40 | BB, BIO | | |
| E-EU-08-A-00 | 08:32:45 | 09:19:00 | | | |
| E-EU-08-A-01 | 08:32:45 | 08:42:00 | BB | | |
| E-EU-08-A-02 | 08:46:00 | 08:54:00 | BB | | |
| E-EU-08-A-03 | 09:14:00 | 09:19:00 | AP, BB | | |
| E-EU-08-B-00 | 09:41:25 | 13:56:45 | | | |
| E-EU-08-B-01 | 09:41:25 | 10:17:00 | AP, BIO | London | 0-3 |
| E-EU-08-B-02 | 10:17:00 | 10:39:30 | AP, BIO | London | 0-3 |
| E-EU-08-B-03 | 10:39:30 | 11:10:00 | AP, BIO | London | 6-24 |
| E-EU-08-B-04 | 11:14:10 | 11:25:35 | AP, BIO | London | 3-6 |
| E-EU-08-B-05 | 11:25:35 | 11:45:00 | AP, BIO | London | 3-6 |
| E-EU-08-B-06 | 11:45:00 | 11:53:00 | AP | London | 12-24 |
| E-EU-08-B-07 | 11:53:00 | 12:05:50 | AP | | |
| E-EU-08-B-08 | 12:05:50 | 12:42:45 | AP | London | 12-24 |
| E-EU-08-B-09 | 12:42:45 | 13:02:00 | AP, BIO | London/BNL/Ruhr | 12-48/0-6 |
| E-EU-08-B-10 | 13:06:00 | 13:14:00 | AP, BIO | BNL/Ruhr | 0-12 |
| E-EU-08-B-11 | 13:14:00 | 13:38:15 | AP, BIO | BNL/Ruhr | 0-3 |
| E-EU-08-B-12 | 13:38:15 | 13:56:45 | AP, BIO | BNL/Ruhr | 0-3 |

### 4.2.2 Characterisation of the MPC London outflow

The vertical and horizontal extension of the observed outflows during EMeRGe is investigated by combining the information from transects and shuttles in selected areas. Figure 24 shows, as an example, the CO, $O_3$, $SO_2$, $RO_2^*$, $NO_y$, NO, $C_6H_6$ and BC observations made for the B-01 to B-12 plumes during the E-EU-08 flight. The E-EU-08 track included a flight transect (a-b-c-d-e) at approximately 600 m altitude and a shuttle (600-1400 m) between b-c and c-d in the outflow of London from 10 UTC to 12 UTC. A second shuttle (g-h-i) at 900, 1500 and 2400 m was made in the BNL outflow from 13:20 UTC approximately. Relevant changes in the HALO course and altitude are marked by coloured circles and letters in Fig. 24.

Backward trajectories indicate that the air measured at around 10:30 UTC at 600 m (blue circle), 11:00 UTC (point c at 1400 m and 600 m), 11:20 UTC (yellow circle) and 11:50 UTC at 600 m (pink circle) had passed over the MPC London a few hours before being probed at an altitude below 1000 m. Selected backward trajectories are shown in Fig. 24c. At these times, the measured enhancements in CO and $NO_y$ and the $NO/NO_y$ ratios are in reasonable agreement with the transport time predicted by HYSPLIT for the CO enhancement in the MPC London plumes in Fig. 23. For plume B-02, HYSPLIT predicts the London contribution to be a mixture of air masses transported in the previous 3 to 24 hours. The air probed had up to 10 ppb of $NO_y$ and approximately 2 ppbv NO. The latter suppresses $RO_2^*$. OH and RO are produced but also react with NO and $NO_2$. These measurements confirm the predicted mixing of relatively fresh emissions with aged and more photochemically processed air masses. The vertical distribution of CO in the plume during the shuttles is depicted in the 3D diagrams in Fig. 24b. The CO measured indicates that the plume B-03 is well mixed horizontally with the plume B-06 up to 1400 m altitude. According to the backward trajectories (not shown), the plume at 11:52 UTC is





transported from the Northeast coast of UK and had no recent contact with the outflow of London. This is
distinguishable by the significantly higher $SO_2$ mixing ratios measured.
The plumes B-08 and B-09 measured over the continent at 900 m are predicted to have been in contact with
emissions of the MPC London within the previous 24 hours (Fig.23 and Fig. 24c). From 12:50 UTC the air
probed is expected to mix with recent emissions of the MPC BNL as indicated by the observed higher NO levels
and enhancements in NOy, $SO_2$ and $C_6H_6$ in Fig. 24a.
The composition of the air measured during the shuttle between the way points g and h in Fig. 24a at 13:30 and
13:45 UTC and the backward trajectories indicate that the outflow from the MPC BNL was sampled in a plume
extending from 1000 m to 1500 m. This air mass was not detectable at 2500 m.

**a)**

**b)**

**c)**

**Figure 24:** a) CO, $O_3$, $SO_2$, $RO_2*$, $NO_y$, NO, $C_6H_6$ and BC measured in the outflow of London and BNL during E-EU-08 on 26 July 2017. The position and numbering of the plumes are indicated by blue lines and numbers as classified in Fig. 22 (B is omitted for clarity), b) 3D shuttles colour coded with the CO mixing ratios observed. Relevant changes in the HALO course and altitude are marked by colour circles and letters (a-i). c) Selected backward trajectories (24h). The red stars indicate the position of the MPCs of interest.

Further information about the characteristics of the plumes is obtained from the air samples gathered with

MIRAH on–board HALO and on the ground sites in London and Wuppertal during the flight E-EU-08. As stated



in 4.1, lower carbon isotope ratios indicate fresh emissions, whereas higher values indicate an enrichment of the
compound in $^{13}$C, which is linked to chemical ageing.
In Fig. 25, the measured $\delta^{13}$C values of $C_5H_{10}O$ and $C_6H_6$ are shown as examples. The identified London outflow
is also evident in the carbon isotope ratios obtained from HALO samples taken between 10 and 11 UTC. The
latter remain in the range of the representative source values from whole air samples collected at the ground
station in London. The higher $\delta^{13}$C values observed between 11:10 and 12:00 UTC indicate chemically-
processed London outflow air.
Later in the flight, the $\delta^{13}$C values measured over the BNL/Ruhr area are in the range of the source values in air
samples collected in Wuppertal. The range in $\delta^{13}$C values of ± 1.5 ‰ in $C_5H_{10}O$ (± 3.5 ‰ in $C_6H_6$) implies a
mixture of slightly aged air and rather fresh emissions from the Ruhr area.

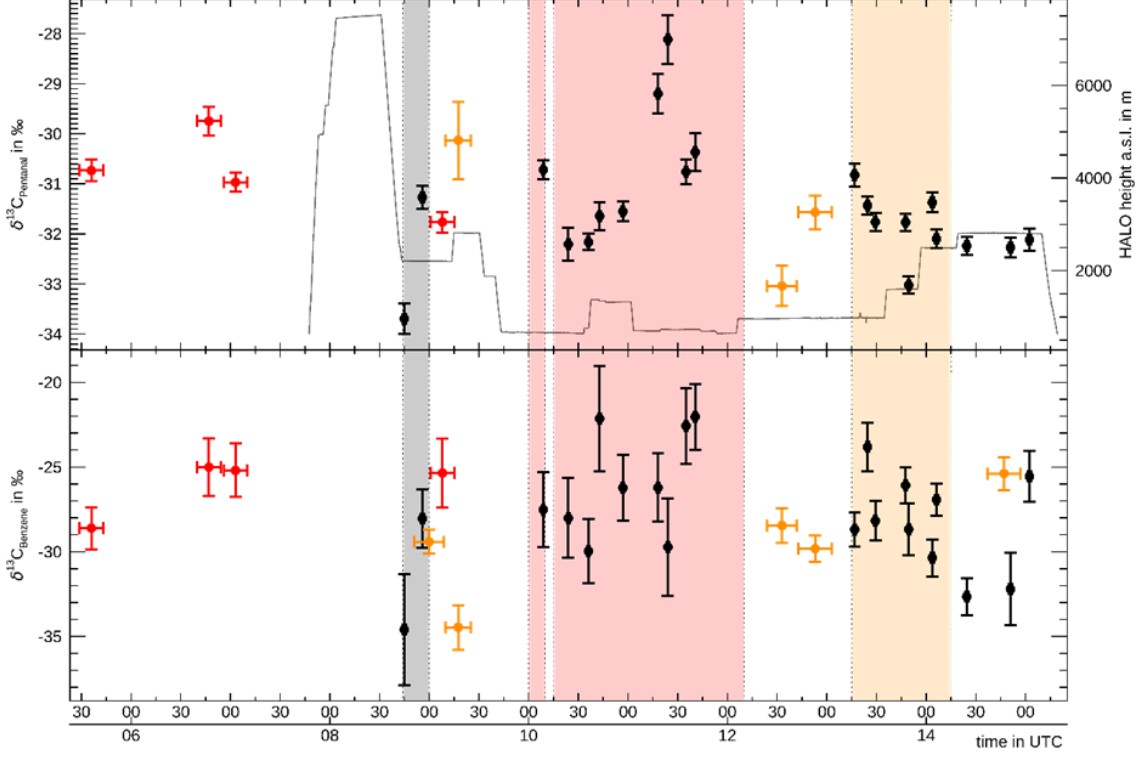


**Figure 25:** $\delta^{13}$C values in $C_5H_{10}O$ (top panel) and $C_6H_6$ (bottom panel) in whole air samples gathered with the whole air
sampler MIRAH on the HALO aircraft (black) during E-EU-08 as well as on the ground sites in London (red) and Wuppertal
(orange). The HALO flight altitude is given in grey on the top panel. Background shadings indicate different measurement
regions during the flight according to Fig. 4: Paris (grey), South of London and North Sea region (red), BNL/Ruhr (orange).
Pollution plumes of the London MPC outflow were also assigned during E-EU-05. These were measured after
transport over the English Channel and to the European continent. Similar to the study of Ashworth et al. (2020),
the processing of the plumes from the emissions probed by the FAAM aircraft in the circuits around London will
be addressed in separate publications. Observations of the released PFC tracer in London improved the definition
of the plume in the area of measurement.



**4.3 Specific case studies of MPC outflows**
In addition to the plume from London, other MPC outflows were identified and analysed during the EMeRGe
IOP in Europe by combining tagging and observational tools. Two representative case studies are briefly
presented in the following. The corresponding detailed analysis is subject of separate publications.
**4.3.1 MPC Po Valley and Rome**
Shuttles at different altitudes upwind of Rome in the Mediterranean and along the Adriatic coast during the
flights E-EU-03 and E-EU-06 provided information about the vertical distribution of trace gases at different
distances from the sources of the MPC Po Valley and MPC Rome.
As for the MPC London case in Sect. 4.2., backward and sensitivity trajectories support the identification of
plumes downwind from these MPCs. The density distribution for forward trajectories (FT) of MPC Rome
outflows in Fig. 26 highlights the typical transport pattern towards the Adriatic coast and the representativeness
of the HALO measurements. The flight tracks for E-EU-03 and E-EU-06 are colour-coded with the BC mass,
showing a good agreement between the four-year FT analysis and the actual in-situ measurements. These results
also strengthen the assumption of the HALO measurements being representative for the transport of air masses
from the MPC Rome. The FT density distribution was calculated as explained in Pöhlker et al., (2019). The FT
starts at 100 m above ground level for the month of July in a multi-year period (2017 until 2020) by using the
HYSPLIT package (version 4, Revision 664, October 2014) (Stein et al., 2015; Rolph et al., 2017).
For the Rome MPC, the airborne measurements at low altitudes made by the Sky Arrow research aircraft agree
reasonably well with the columnar amounts of gases observed by the PANDONIA global network for air quality
and atmospheric composition (https://www.pandonia-global-network.org/) and the remote sensing observations
on-board HALO. These data support the determination of the geographical extension and location of the Rome
outflow (see Barnaba et al., Campanelli et al., in preparation 2021).

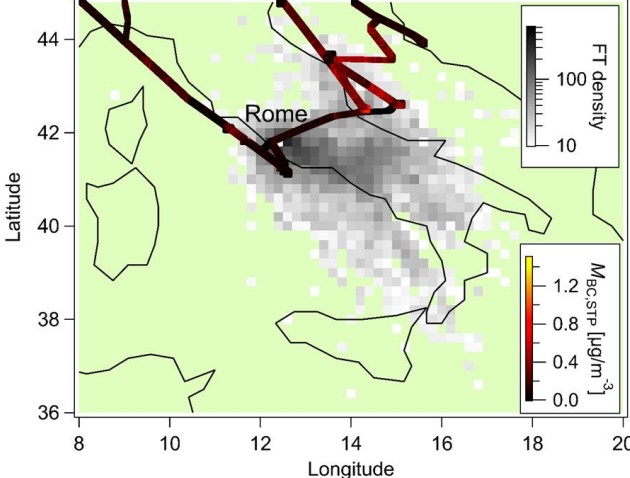


**Figure 26:** Forward trajectory (FT) density plot for air masses starting in Rome (100 m a.g.l.) in the month
of July from multiple years (2017 to 2020). The grey scale represents the counts of FT points in each grid
cell. The flight track of E-EU-03 and E-EU-06 is colour-coded with the BC mass concentration.





The MPC Po Valley has surface emissions from the urban agglomeration over a relatively large area. It is a good
example of a patchy and complex outflow that has largely been investigated as pollution hot spot in Europe.
Several studies show the importance of the pollution transport from this area to the surrounding regions (e.g.
Diémoz et al., 2019a, 2019b) and the complexity of chemical and dynamical processes within the Po Valley
mixing layer (e.g. Curci et al., 2015). The Alps and Apennines on the Italian Peninsula lead to the transport of
the Po Valley outflow southwards along the Italian Adriatic coast which is the geographic opening of the Po
Valley (Finardi et al., 2014). In a dedicated study, the in-situ and remote instruments at ground-based sites and
airborne measurements from two aircrafts are combined to examine in detail the transport of pollutants during
the EMeRGe IOP for the case Po Valley (Andrés Hernández et al., in preparation 2021).
When HALO flew over MPC outflows but did not sample them in-situ, the down-looking remote sensing
instruments on-board enabled the identification of plumes as illustrated in Fig. 27 by using HAIDI measurements
at 8 km of the Milan outflow during E-EU-09. The measurements of HAIDI were used to estimate emissions and
plume geometries, $NO_2$ being an important target species.
The HAIDI instrument has three scanning telescopes pointed at nadir, 45° forward and 45° backwards direction.
On the left side of Fig.27, the data from the nadir telescope scanner are shown at high spatial resolution. The
map shows a strong $NO_2$ plume Northeast of Milan. The plume substructures are also clearly visible. On the
right side of the figure, the data from all three telescope scanners are plotted as a function of time at a lower
spatial resolution. The time delay of about 80 s between the peak as seen in the forward and backward scanners
indicates that this plume is close to the ground. Wind data from the lowest layer from the ECMWF ERA-5
reanalysis product [Copernicus Climate Change Service, 2017] implied a wind angle of 23.8°, which is
consistent with this plume originating from the city of Milan. The estimated $NO_2$ emission rate of 607± 67
kg/day may have a higher uncertainty due to the low wind speed (0.6 m/s), the complex plume shape and the
small relative angle between the HALO flight track and the plume direction.

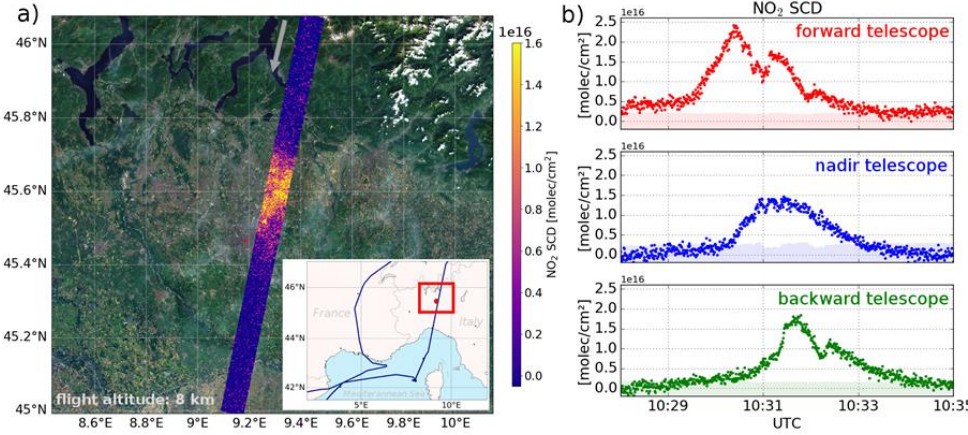


**Figure 27:** HAIDI measurement at 8 km altitude of the Milan outflow during the flight E-EU-09: a) pixel-resolved $NO_2$ slant
column densities observed by the nadir camera (marked by the red square on the map). An enhancement of up to $1.5\times10^{16}$
molec/cm² over the background is observed Northeast of Milan (red coloured circle), b) $NO_2$ slant column densities averaged
over the whole swath for all three telescopes: forward (top) nadir (middle) and backward (bottom). The height of the plume
centre is estimated from the time difference of the maxima. Sources of background imagery: ESRI, DigitalGlobe, GeoEye, i-
cubed, USDAFSA, USGS, AEX, Getmapping, Aerogrid, IGN, IGP, swisstopo, and the GIS User Community.





### 4.3.2 MPC Madrid and Barcelona

The vertical distribution of pollutants observed at the coast of Barcelona during E-EU-09 is a particular case of interest for the study of vertical layering of pollution. HYSPLIT CO dispersion simulations indicate that the Madrid outflow was transported over a long distance above the Iberian Peninsula to the North-Eastern coast at altitudes above 2000 m while in the lower layers the Barcelona outflow predominated, as illustrated in Fig. 28. In contrast with the air sampled at 500 m, the backward trajectories and HYSPLIT dispersion calculations indicate that the air probed from 15:15 to 15:25 UTC at 1600 m had passed over MPC Barcelona within 6-12 hour before sampling. There is no indication of fresh NO emissions, and $NO_y$, $C_6H_6$ and CO are significantly higher than at the lower altitude. The layering is attributed to be the result of the recirculation of emissions in the Barcelona outflow within the land-breeze regimes close to the coast. Later at this FL (green and red circles in Fig. 28), the backward trajectories and HYSPLIT estimations indicate sampling of regional emissions that had travelled along the coast from Valencia. This is consistent with the observed decreases in $C_6H_6$, $NO_y$ and BC. In the upper FL at 15:45 UTC, $NO_y$, $C_6H_6$ and CO significantly increase in air transported from Portugal (as in the 36 h backward trajectories) across the Iberian Peninsula at altitudes above 2000 m, after PBL contact with the MPC Madrid below 1000 m the evening before. According to the pollution control network of Madrid, the average CO surface concentration exceeded 350 ppb on the 27 July 2017, the zonal wind direction was WSW and the average wind speeds were greater than 16 km/h. The observed mixing ratio decreases when this feature at 3000 m disappears. Re-entering and stratification of plumes having different processing along the Spanish coast has also been documented in the past (e.g. Millán et al., 1997, 2000 and references therein).

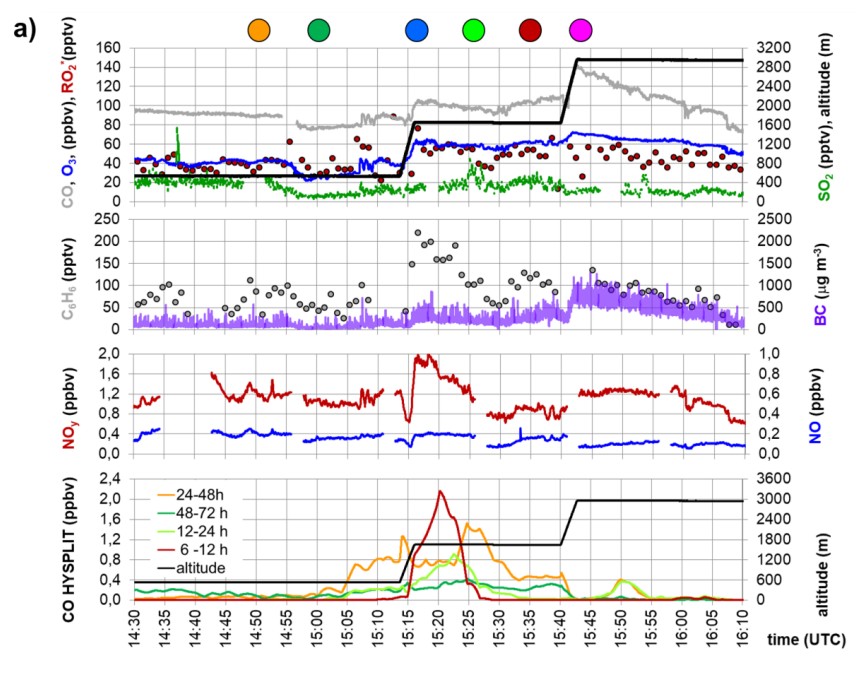

1152

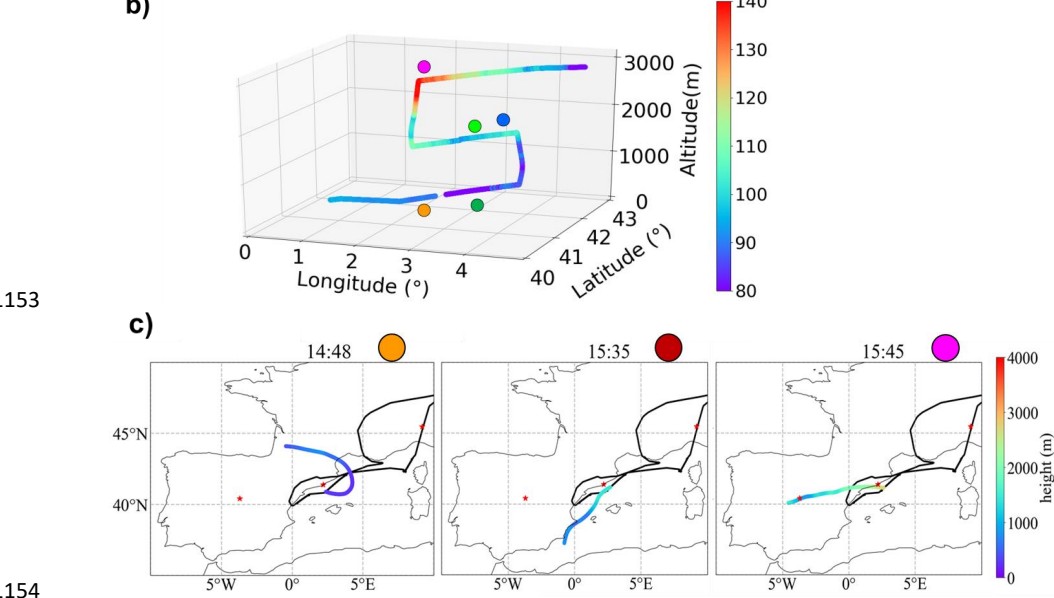

1153

1154

**Figure 28:** Stratified pollution layers along the Spanish coast during the E-EU-09 flight on the 28 July 2017, a) temporal variation of CO, O₃, RO₂*, NOᵧ, NO, SO₂, C₆H₆ and BC during the shuttle, b) 3D view of the shuttle colour coded with CO mixing ratios, c) selected backward trajectories (last 24h). Coloured circles marked the corresponding times. Red stars indicate the position of the MPCs of interest.

These HALO measurements are consistent with the long-term analysis of data from the closest four ground-based remote sensing stations available in the framework of EMeRGe international. These are data of a lidar in



Barcelona (BRC) and three ceilometers in Montseny (MSY), on top of the Serra del Montsec (MSA) (Titos et
al., 2019) and in Burjassot (VLC) near Valencia. Figure 29 shows the location of the stations with respect to the
HALO flight track. The stations MSY and MSA were approached at a flight altitude of 2600 m when HALO
entered the air space above the Iberian Peninsula. Subsequently, HALO shuttles were carried out Northeast of
Valencia at 500, 1000, 2000 and 2600 m as well as East of Barcelona at 500, 1600 and 3000 m, as presented in
Fig. 28.

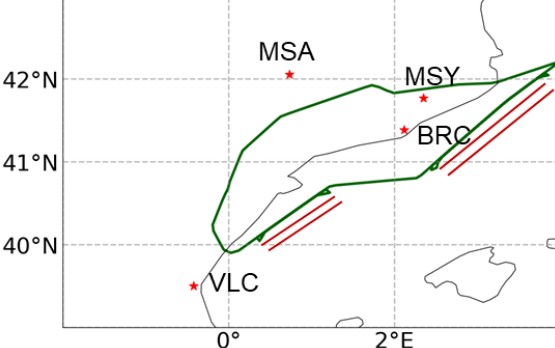


**Figure 29:** Detail of E-EU-F09 flight track (in green) and the ground-based stations with coordinated
remote sensing measurements in the vicinity: Montseny (MSY), Sierra del Montsec (MSA), Burjassot
(VLC) and Barcelona (BRC). Red lines indicate the position of the HALO shuttles.
A lofted aerosol layer from above the PBL up to 4000 m altitude was observed at all ground-based remote
sensing stations and also probed by HALO (see Fig. 30).

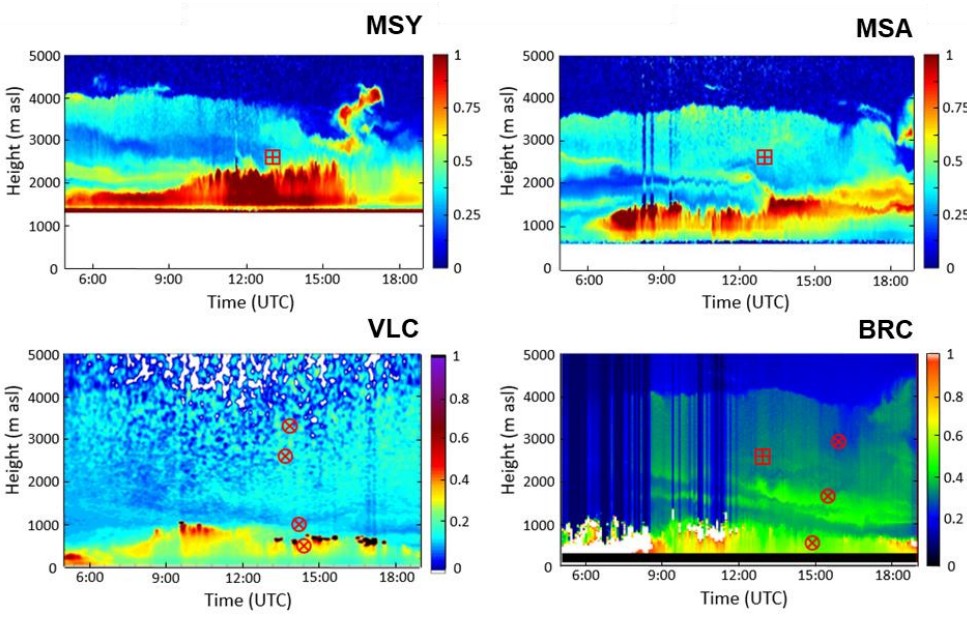


**Figure 30:** Time series of range-corrected lidar signals ground-based remote sensing measurements in MSY, MSA (both at a
wavelength of 1064 nm), VLC (910 nm) and BRC (532 nm) on the 28 July 2017. Signal strengths relative to the maximum
signal of the corresponding measurement are depicted. Red circles show time and altitude of the HALO overpasses used for
comparison of airborne with ground-based remote sensing measurements (see Fig. 31 and Fig. 32). Red squares show further
HALO overpasses.



The profiles of the backscatter coefficient derived at MSA, MSY, VLC and BRC on the 28 July 2017 are
displayed in Fig. 31 and Fig. 32. These measurements illustrate the lofted aerosol layer shown in Fig. 30 with
increased backscatter coefficients ranging from 0.4 to 1.9 (Mm·sr)$^{-1}$. The composition of PM1 particles (i.e.,
with diameter up to 1 micron) was retrieved from the HALO in-situ measurements at different altitudes during
the shuttles. The observed PM1 composition near Burjassot is shown in Fig. 32. Although the ceilometer
measurements refer to total aerosol and the in-situ data only to PM1, both reveal two distinct aerosol layers: a) a
PBL below 1000 m altitude with a backscatter coefficient between 2.0 and 2.7 (Mm·sr)$^{-1}$ and enhanced
concentrations of sulphate and ammonium, and b) a lofted aerosol layer between 1500 and 3500 m altitude with
higher organic, nitrate and BC mass fraction. The difference in composition is likely related to different aerosol
sources. While the boundary aerosol layer has a local origin, the lofted aerosol layer is influenced by the
transport of regional emissions. This is consistent with the transport of the MPC Madrid outflow as indicated in
Fig. 28.

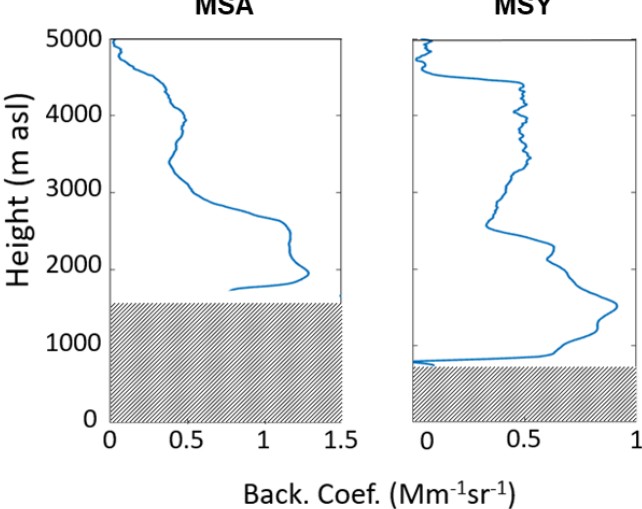


**Figure 31:** Profiles of the backscatter coefficient derived at 1064 nm in MSA and MSY for the 28 July
2017 from 12:50 to 13:20 UTC. The grey shadings indicate the height of the ceilometers.
Similarly, the lidar and in-situ measurements close to Barcelona reveal a different aerosol composition of the
PBL below 900 m and a lofted aerosol layer above 2000 m. In addition, a third aerosol layer evolved between
1000 and 1800 m altitude with a backscatter coefficient up to 1.5 (Mm·sr)$^{-1}$. The mass fractions of ammonium,
sulphate and organic aerosol are between the values of those of the PBL and of the lofted aerosol layer above.

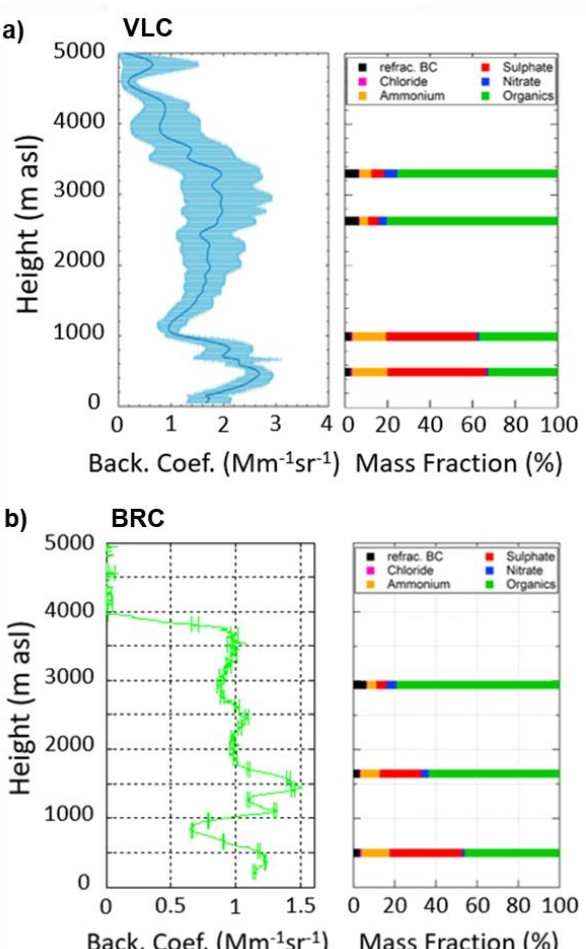


**Figure 32:** Distinct aerosol layers observed near Burjassot/Valencia and Barcelona. a) Profile of the backscatter coefficient derived at 910 nm for 13:30-14:30 UTC in VLC (left), and fractional composition of PM1 measured (SP2 and AMS) on-board HALO (right), b) the same derived in BRC at 532 nm for 14:45-15:45 UTC. The periods of comparison with the HALO data are 13:42-13:56 (9:30 min) at 3300 m; 13:34-13:40 (5:30 min) at 2630 m, 14:03-14:14(11:30 min) at 1000 m and 14:18-14:31 (23 min) at 500 m for VLC, and 15:43-16:00:(17:30 min) at 2940 m; 15:16-15:40 (24 min) at 1650 m, and 14:47-15:14 (27 min) at 500 m for BRC.

**4.4 Specific case studies of mixing of MPC outflows with air masses of biogenic origin: forest fires and dust**

Typically the composition of the measured pollution plumes indicated that  emission came from sources other than of the targeted MPCs. These influence the photochemical oxidation and chemical reactions of the probed air masses. Supporting satellite- and ground-based measurements of forest fire and dust signals enable the identification of these sources.





BB emission from fires was e.g. probed during the E-EU-07 flight downwind of Marseille. The plume transport eastwards from near Marseille is well-captured by SEVIRI with AOT values around 0.25 at 0.55 µm in the afternoon, as shown in Fig. 26. This plume was probed by HALO in-situ measurements at around 11:30 and 16:30 UTC. As an example of the agreement between remote sensing satellite retrievals and HALO observations, BC mass concentrations are also depicted in the figure. The highest BC was measured at roughly 2000 m and exceeded 7 µg m$^{-3}$. In the PBL, measured BC mass concentrations were as high as 1 µg m$^{-3}$. The stratification of pollution plumes above the PBL is a typical feature for BB emissions (Holanda et al., 2020).

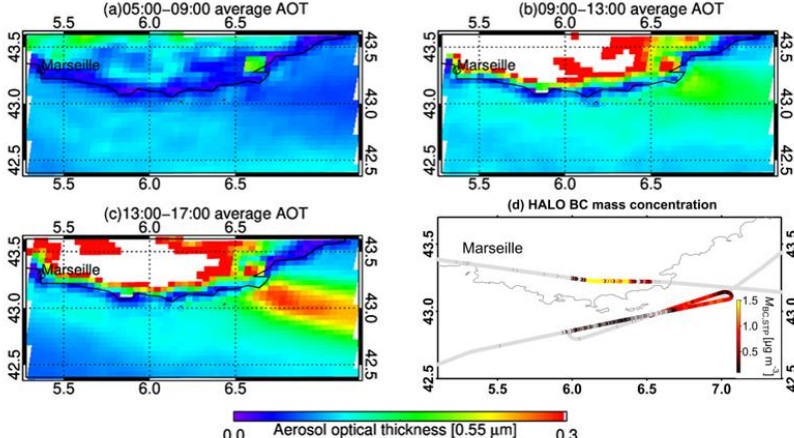

**Figure 33:** (a - c) Aerosol optical thickness at 0.55 µm as retrieved from SEVIRI from 05:00 to 17:00 UTC on 24 July 2017. (d) E-EU-07 flight track, colour-coded with BC mass concentration (M$_{BC}$). For a better contrast, the scale for M$_{BC}$ ranges from 0.1 to 1.5 µg m$^{-3}$. Grey colour on the flight track indicates values below 0.1 µg m$^{-3}$. The mass concentration reached values up to 7 µg m$^{-3}$ at the French coast.

Mixing ratios of CH$_4$ comparable to those in urban plumes were measured in this BB event during E-EU-07 (not shown). This distinct peak concentration strongly influences the local GHG distribution (Klausner, 2020), although the contribution of BB emissions to total global anthropogenic CH$_4$ is on the order of a few percent (Saunois et al., 2019).

Dust events were observed and contributed significantly to some of the plumes measured over Europe during the EMeRGe IOP. On 11 July 2017, there was a Saharan dust event affecting the air masses measured during E-EU-03, as indicated by both satellite- and ground-based observations. Figure 34 shows the MODIS satellite RGB image at 10:30 UTC and the corresponding elevated AOT at 0.55 µm as retrieved from SEVIRI from 09:00 to 13:00 UTC.



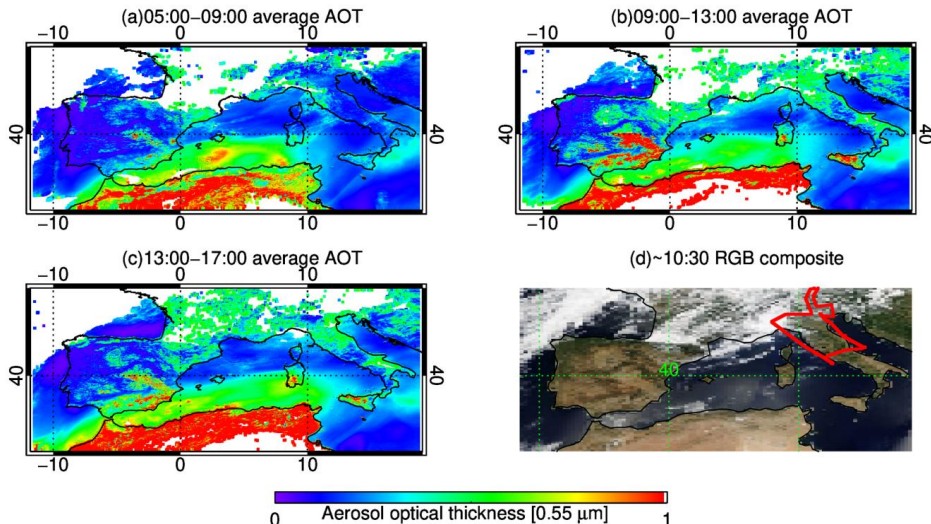

**Figure 34:** (a-c) Aerosol optical thickness at 0.55 μm as retrieved from SEVIRI from 05:00 to 17:00 UTC on 11 July 2017, (d) MODIS RGB composite figure showing corrected reflectance at 10:30 UTC (https://worldview.earthdata.nasa.gov/). The MODIS RGB composite is created combining red, green and blue bands into one picture. White areas are clouds. The E-EU-03 flight track (in red) is superimposed on (d).

The impact of dust on the aerosol size distributions observed on board HALO close to the western coast of Italy during E-EU-03 is illustrated in Fig. 35. The concentration of particles with a diameter below 250 nm was analysed by the Differential Mobility Analyzer (DMA) in 6 steps of 30 s duration, resulting in a period of 3 minutes for each integrated measurement. The evaluated DMA data points are then combined with the data from an Optical Particle Counter (OPC) for particles in the range from 250 nm to 3 μm. The first two sequences in Fig. 35 are taken at 2900 m and the third at 1300 m altitude. The third period and lowest in altitude had the smallest total number concentration with a clear enhancement of the particles above 600 nm. According to FLEXTRA, HALO flew approximately 800 m above the PBL at the time of sampling. The increase in the coarse mode particles above the PBL implies mineral dust rather than sea salt. According to backward trajectories, the air mass probed had recent contact at altitudes below 1000 m with the dust plumes over the Mediterranean near Sardinia shown by MODIS in Fig.34.

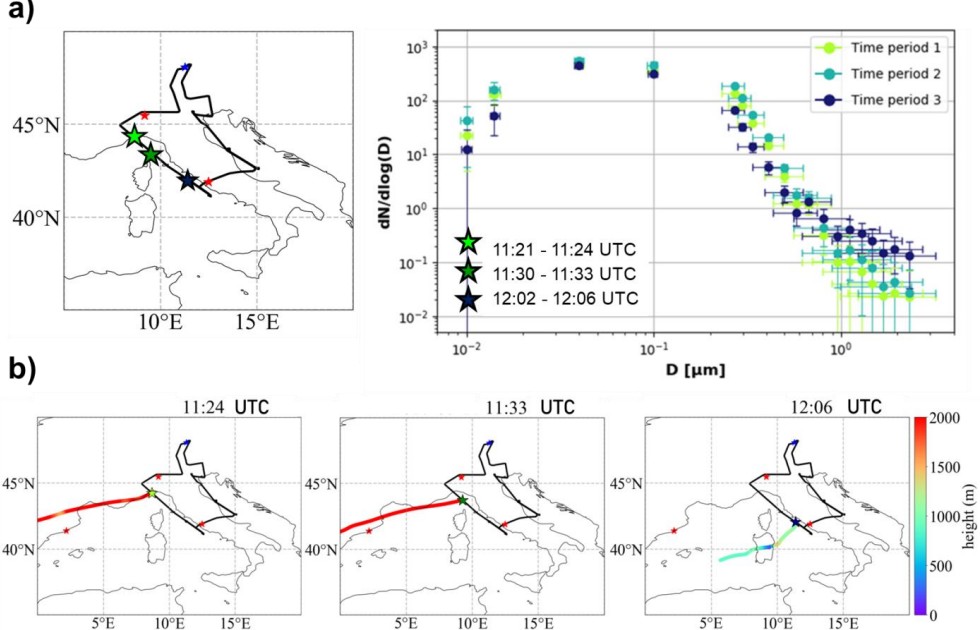

**Figure 35:** Example of the effect of dust plumes on the aerosol concentration during E-EU-03 on the 11 July 2017. a) Particle size distribution for 3 selected time periods (right) and position of the sample points in the flight track (left). The error bars on the y-axis are the standard deviations of the mean measured concentrations. The error bars in x-direction indicate the 16th and 84th percentile of the median diameters of the sensitivities of each size channel, b) 48h backward trajectories for the three periods selected. The red stars indicate the position of the MPCs of interest.

These observations agree with the measurement of the continuous automated lidar-ceilometer (ALC) in Rome on 11 July 2017, which include the overpass by HALO in the Rome area (see Fig. 36). A lofted aerosol layer with increased depolarization was detected at an altitude between 1000 and 2000 m from the morning and mixed with local particles lifted by PBL dynamics in the middle of the day, at the time of the DMA measurement. This indicates that HALO flew above a dust layer during the first two periods of the DMA measurement. Thus, HALO probed rather low concentrations of large particles. Subsequently, HALO dived into the dust layer and this explains the increase of particles larger than 600 nm.

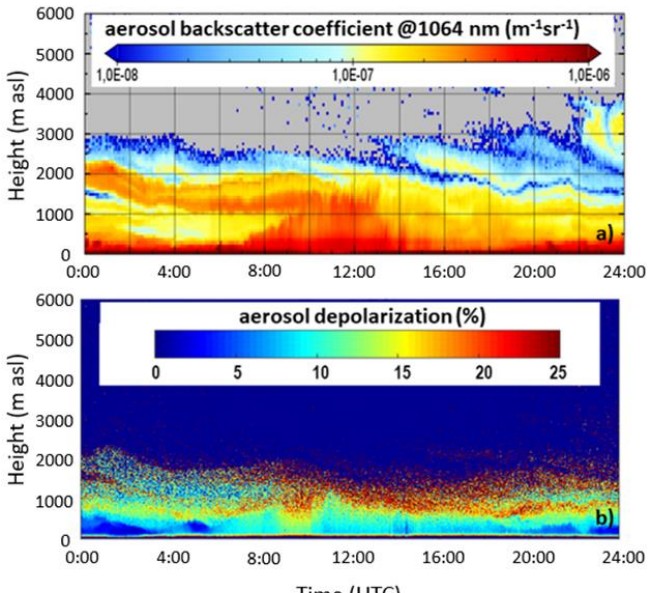


**Figure 36:** Aerosol profile measurements performed in Rome (Italy) on 11 July 2017 by the Automated Lidar-Ceilometer
network (ALICENET). Aerosol backscatter coefficient ($m^{-1}$ $sr^{-1}$) at 1064 nm (top), and aerosol depolarization in % (bottom).

The comparison of fine and coarse mode particles observed on board the Sky Arrow with aerosol properties at
the ground provides evidence for the important role of fine particle photo-nucleation in the MPC Rome, favoured
by high radiation and temperatures (Campanelli et al., 2021; Barnaba et al., 2021 in preparation).
The extent and effect of mixing of air masses of different nature observed during the EMeRGe IOP is
investigated in more detail elsewhere (Förster et al., 2021 in preparation; Holanda et al., in preparation 2021).
**5 Processing of polluted air masses during transport**
Chemical and physical processing of MPC emissions during transport has an important impact on the potential to
form $O_3$ and other secondary photochemical oxidants in the outflows. In addition, photochemical processing
changes the volatility and hygroscopicity of the aerosol particles and thereby their impact on cloud formation. In
this sense, the EMeRGe airborne observations of primary and secondary pollutants and the ratios between
species having different chemical lifetime were used as tracers of the degree of processing of the pollution
plumes probed.
The $NO/NO_y$ ratio provides information about the reactivity of the air mass but is not a reliable chemical clock
due to the complex and rapid chemistry involved in the air masses investigated. Depending on the chemical and
physical conditions, the lifetime of NO versus the formation of other reactive nitrogen compounds is of the order
of a few hours or less. Internal transformation processes within the family of total reactive nitrogen $NO_y$ do not
alter their integrated concentration. However, washout and aerosol formation are loss processes controlling the
lifetime of $NO_y$, which varies between hours and days.
A more robust chemical clock is the $NO_y$ to CO ratio which is generally used to study ageing of an air mass with
respect to ozone and nitrogen chemistry (e.g. Stohl et al., 2002). The CO lifetime varies between several weeks



and months (e.g. Emmons et al., 2010). Depending on the distance from the source as well as on the chemical
and physical properties of the air mass, the $NO_y/CO$ ratio declines to background values within a few days. As
expected within the EMeRGe IOP in Europe, the $NO_y/CO$ values were generally significantly higher for the
processed polluted plumes than for the background air masses. For instance, during E-EU-08 discussed in Sect.
4.2, the $NO_y$ to CO ratio was of the order of 0.01 to 0.02 in the air sampled outside the outflow of London and
increased up to 0.1 in the London outflow plumes, as the air mass was processed and mixed.
The ratio between VOCs with comparable emission sources but significantly different chemical lifetimes is often
used as a chemical clock to study emissions from point sources. This is the case for $C_7H_8$ and $C_6H_6$ emitted from
gasoline-powered engines used in traffic and industry (Gelencsér et al., 1997; Shaw et al., 2015; Warneke et al.,
2001). The atmospheric lifetime of these aromatic hydrocarbons, i.e., 1.9 and 9.4 days, respectively (Garzón et
al., 2015), is assumed to be controlled only by the reaction with OH radicals (Atkinson, 2000). Provided that the
emission rates are known, the $C_7H_8/C_6H_6$ ratio is expected to decrease with increasing distance to the pollution
source and can be used to estimate the photochemical age of the sampled air (Winkler et al., 2002; Warneke et
al., 2007). For EMeRGe, the ratio of $C_7H_8/C_6H_6$ is a good indicator for the presence of freshly or already
processed anthropogenic emissions in the probed air. However, since the emission ratios of distinct VOC sources
vary (Barletta et al., 2005), the active plume mixing before sampling as in EMeRGe, limits the use and
feasibility of this chemical clock for the determination of the transport time of a specific outflow.
Information about the ageing of the air mass is additionally derived from differences in the chemical
composition of aerosol particles. Aerosol mass spectrometer data using organic ions containing oxygen, e.g.
$CO_2^+$ (m/z 44) and $C_2H_3O^+$ (m/z 43), are used to assess photochemical oxidation. Observations from laboratory
and field studies indicate that during photochemical processing the ion signal of m/z 43 decreases while that of
m/z 44 increases (Ng et al., 2010; Lambe et al., 2011). This metric is used to infer the degree of photochemical
processing of organic aerosol in the atmosphere (e.g., Ng et al., 2011; Schroder et al., 2018; de Sa et al., 2018).
In that regard, photochemical processing of aerosol particles was evident during the transport of MPC plumes
during the EMeRGe IOP.
Since photo-oxidation of fresh plumes is fast and mixing of aged plumes with the background occurs, the use of
aerosol composition to asses photochemical processing requires complementary information from other
measurements to act as a reliable indicator. Figure 37 shows an example of photochemical processing of the gas
and the aerosol phases in ageing London plumes as measured by the C-ToF-AMS during E-EU-08. The data are
plotted in $f44$-$f43$ space, where $f$ denotes the ratio of the respective ion to the total organic ion signal. In these
metric, atmospheric processing moves the data points towards the upper left corner of the triangle indicated by
the dotted lines (Ng et al., 2010). The simultaneous measurements of CO are used to indicate dilution, while the
atmospheric processing is inferred from other gas-phase measurements ($C_7H_8/C_6H_6$ and $NO_y/CO$ colour codes).
Lower CO concentrations due to plume dilution along transport correspond to higher photochemical processing
in the upper part of the triangle. As $NO_y$ has a shorter lifetime than CO, the $NO_y/CO$ ratio indicates that the
processing is taking place in addition to dilution. Therefore, lower $NO_y/CO$ and $C_7H_8/C_6H_6$ ratios in the upper
part of the triangle indicate aged and processed air. In this case, the FLEXTRA backward trajectories revealed
that the air masses identified as "background" were transported above the PBL and had no recent contact to the
MPC London. The anthropogenically influenced air masses represent a mixture of recent emissions and
photochemically processed London outflow as mentioned in 4.2.2 (see Fig. 24 and Fig. 25).

a)

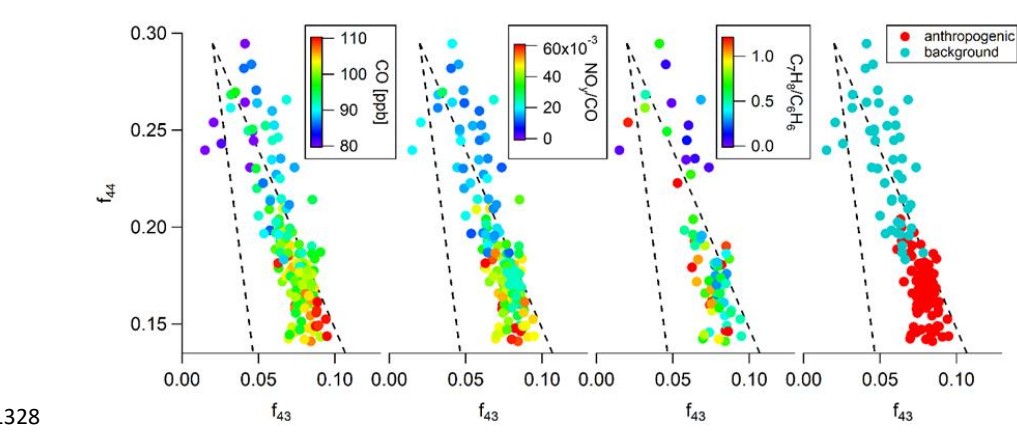

b)

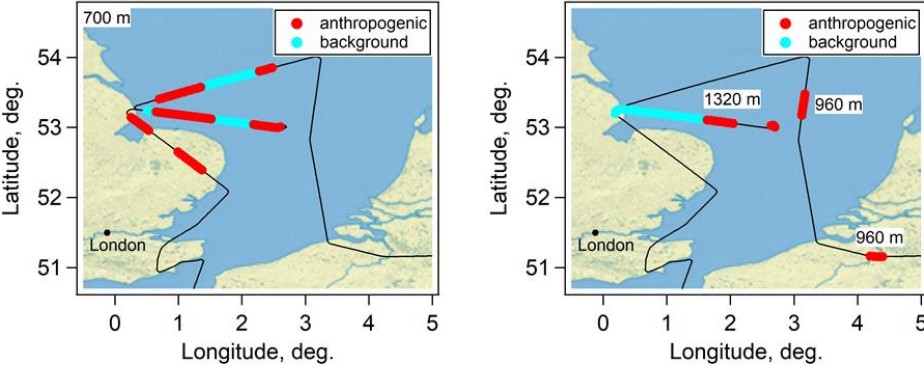

**Figure 37:** a) Scatter plots of C-ToF-AMS signal fractions at m/z 44 (f44) and m/z 43 (f43) of the London plume measured during the E-EU-08 on 26 July 2017 between 10:20 and 12:57 UTC. In this metric, the degree of photochemical processing increases to the upper left corner of the triangle which encompasses the range of typical atmospheric observations The colour code indicates dilution (CO) and processing of the gas phase ($NO_y$ to CO and $C_7H_8$ to $C_6H_6$ ratios). The right panel shows the assignment to unpolluted background air and air masses of anthropogenic polluted origin as introduced in Sect. 4.1. b) Spatial distribution of the background and anthropogenic polluted air masses identified in a). The flight altitudes are indicated in the graphs.

The results presented above confirm the complexity of the air masses as a result of the mixing of sources. Following the ageing of the outflow of a single MPC is challenging. However, the distinction between fresh and aged air is possible and gives a coherent picture for the applied methods and chemical clocks. At large distances from the source, the use of gas and aerosol trace species is insufficient for identifying MPC plumes. In this context, the relevance of PFC tracers and the support of adequate transport models becomes obvious.

The secondary formation of pollutants as a result of plume processing was further investigated with the support of HYSPLIT plume age simulations. An example is formic acid (HCOOH), the most abundant organic acid in the troposphere. Although HCOOH has primary sources, i.e., the emissions by fossil fuel combustion and biomass burning, the secondary formation from gas-phase and aqueous photochemistry has been suggested to be



1347 dominant in the troposphere (Paulot et al., 2011). During EMeRGe, HCOOH was measured by CI-ITMS by

1348 using $CO_3^-$ as reactant ion (Viidanoja et al., 1998). Significantly enhanced volume mixing ratios up to 25 ppb

1349 were observed in the pollution plumes of MPCs in Europe, and HCOOH was found to be more abundant in the

1350 plumes than sulphur and nitrogen precursor species of inorganic acids (Eirenschmalz et al., in preparation 2021).

1351 Figure 38 shows HCOOH enhancements above ambient background relative to CO enhancements in different

1352 MPC plumes as a function of plume age. Here, ΔHCOOH and ΔCO are determined from the measurements, and

1353 the plume age from HYSPLIT simulations considering CO emissions from EDGAR and the dispersion of the

1354 plumes during transport. CO is used as an indicator of the strength of emissions from combustion in the

1355 individual MPC plumes and as tracer for the dilution of the plumes for the actual meteorological conditions

1356 during the measurements. The ΔHCOOH to ΔCO ratios significantly increase with plume age indicating

1357 secondary formation of formic acid to be its main source in the MPC plumes, mainly due to oxidation of $C_5H_8$ in

1358 the plume.

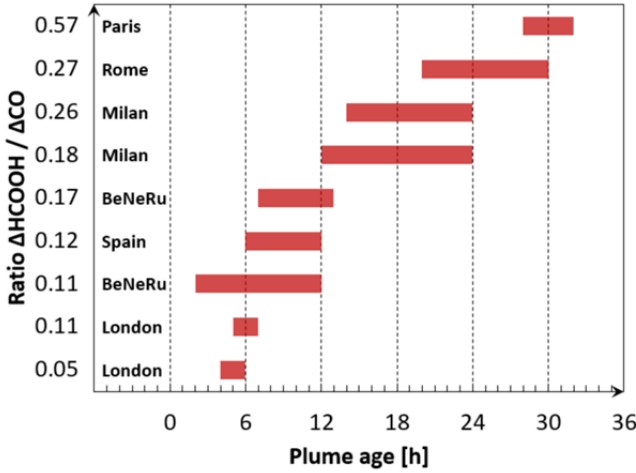

1359

1360 **Figure 38:** Observed enhancements of formic acid (ΔHCOOH) in MPC plumes relative to observed CO enhancements
1361 (ΔCO) as a function of plume age from HYSPLIT simulations. The corresponding city-plume is indicated next to the ratios.

1362 **6 Simulating the processing of European MPC emissions with the MECO(n) model**

1363 Atmospheric modelling is used to place the spatially and temporally limited number of observations during

1364 EMeRGe into a broader context, e.g. by analysing long term trends or temporal and spatial variability in the

1365 MPC emissions in Europe.

1366 The EMeRGe data set offers an opportunity to test whether the transport and transformation of MPC emissions

1367 are well captured by state-of-the-art atmospheric models. In this context, simulations with the MECO(n) model

1368 (Kerkweg & Jöckel 2012, Mertens et al., 2016) were performed. The model couples a global and a regional

1369 chemistry climate model. In the set-up applied here, Central Europe was resolved with up to 7 km horizontal

1370 resolution. The model data was sampled along the HALO flight paths with 60 s temporal resolution using the

1371 MESSy submodel S4D (Jöckel et al., 2010). These sampled model data are used for a one-by-one comparison

1372 with the measurements. The EDGAR 4.3.1 emission inventory for the year 2010 was used.

https://doi.org/10.5194/acp-2021-500
The tagging method by Grewe et al., (2017) was applied as additional model diagnostics. This method
decomposes the budget of ozone and ozone related precursors into the contributions of different emission sectors
(Mertens et al., 2020a). Out of the 12 applied emission categories, land transport (mainly road traffic) in Europe,
anthropogenic (other than traffic) in Europe, shipping, land transport outside Europe, anthropogenic (other than
traffic) outside Europe, lightning and biogenic emissions are the most important ones (see Fig. 39b). A detailed
description of the model and the source apportionment technique are provided in the supplement (see S12).

The model results show a positive bias in $O_3$ and a negative bias in CO with respect to the EMeRGe
measurements over Europe. This confirms previous comparisons with other observational data (see Mertens et
al., 2016, 2020b). Given the complexity of the air masses sampled during EMeRGe, the comparison with the
model results was extended by undertaking different sensitivity studies to investigate the impact of specific set-
up changes on the simulated mixing ratios.

An example is given for the E-EU-05 flight on 17 July 2017. The comparison between measured $NO_y$ mixing
ratios and MECO(n) results is shown in Fig. 39a, when the London plume was probed over the English Channel.
The enhancements of $NO_y$ between 12 and 16 UTC below 900 hPa are reasonably well simulated by the model
except for the measurements at around 15:30 UTC which are strongly overestimated by the model. To address
this issue, two plumes marked with '1' and '2' in Fig. 39a were investigated in more detail.

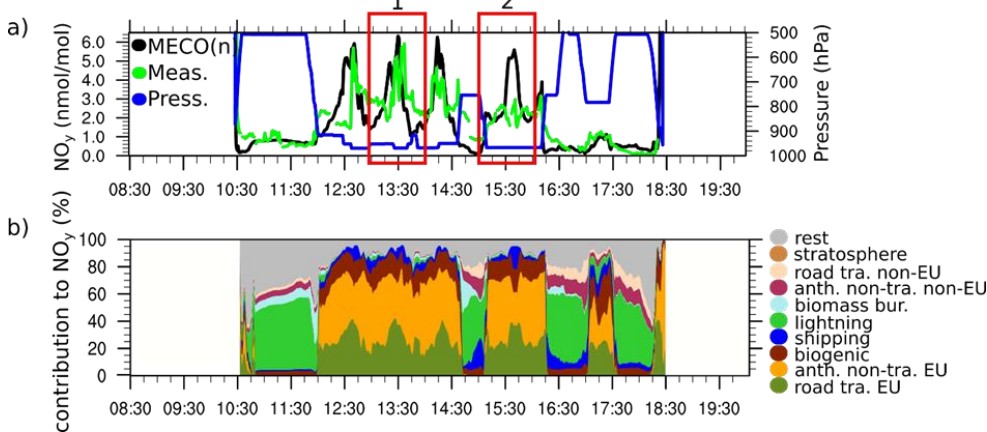

**Figure 39:** a) $NO_y$ mixing ratios measured (green) and simulated by the MECO(n) model (black) for E-EU-05 on 17 July
2017. The blue line denotes the pressure altitude of the aircraft (right axis). b) Relative contributions of different emission
sectors to the $NO_y$ mixing ratios simulated by MECO(n). Note that the NOy measurements were averaged to 60 s to fit the
MECO(n) temporal resolution.

The model results and the measurements on the plume marked '1' are shown at 980 hPa and 965 hPa in Fig. 40a.
980 hPa is the pressure of the model layer which is nearest to the HALO flight altitude at 13:30 UTC while 965
hPa is pressure of one model layer above. The model results show large horizontal and vertical inhomogeneities
in the $NO_y$ mixing ratios indicating different mixtures instead of a single London plume. The $NO_y$ enhancement
coincides with the London plume (marked with the turquoise square in Fig. 40a).

Similarly, Figure 40b shows the model results and measurements for the plume marked '2'. Here, the model
shows a large plume remanence in the western part (turquoise square in Fig. 40b) leading to the overestimation
of mixing ratios around 15:30 UTC. The simulated mixing ratios in a higher model layer are lower and agree





better with the observations. These results indicate that a vertical displacement of the plume remanence causes
the mismatch between measurements and model results around 15:30 UTC.

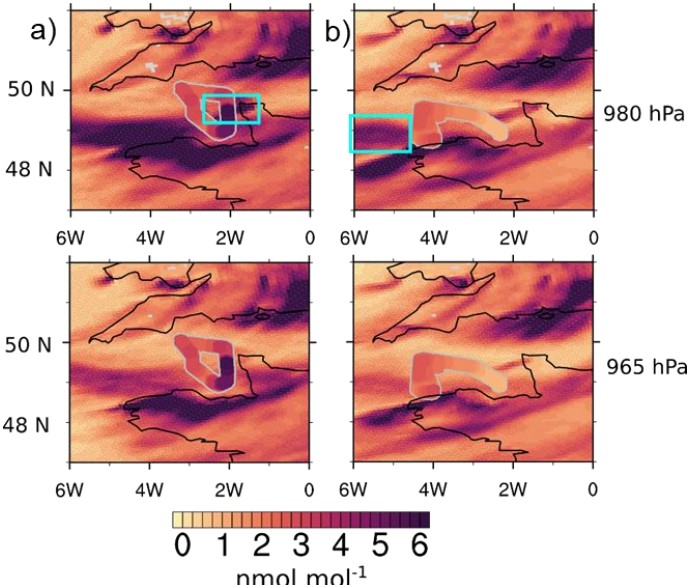


**Figure 40:** NO$_y$ mixing ratios as simulated by MECO(n) (background) and measured during E-EU-05. The model results at
980 hPa and 965 hPa are shown. Model results are averaged between a) 13 and 14 UTC, b) 15 and 16 UTC.  The measured
mixing ratios of NO$_y$ during 13-14 UTC and 15-16 UTC are colour-coded and highlighted by grey contours. Black lines
indicate coast lines. The turquoise rectangles highlight the regions discussed in the text.
The agreement between the measurements and model results shows that the emissions of NO$_x$ and/or their
further processing in the model (deposition, washout, chemical transformation) are reasonably well represented
by MECO(n). However, the simulation of complex plume structures would benefit from a higher model spatial
resolution.
The diagnostic capabilities of MECO(n), e.g. the tagging method, were applied to individual EMeRGe flight
tracks to provide a better understanding of the impact of emissions on the atmospheric chemistry in Europe.
Figure 39b shows the relative contribution of the different emission sectors to the measured NO$_y$ mixing ratios
during the E-EU-05 as a stacked graph. According to this, emissions from European road transport,
anthropogenic non-traffic and biogenic sectors dominate the NO$_y$ mixing ratios of the London plume with a
similar relative contribution in all four plume crossings. For the NO$_y$ measurements in the free troposphere (until
12 UTC approximately) a large relative contribution of lightning emissions is calculated in the model. In these
regions, however, the absolute mixing ratios are rather low. As the NO$_y$ lifetime is much longer in the upper
troposphere than in the PBL, LRT of NO$_y$ might be more likely than encounters of fresh lightning NO-plumes.
The MECO(n) model was further evaluated within EMeRGe by similar analysis with different measured
chemical species and emission inventories. The combination of the MECO(n) results with HYSPLIT backward
trajectories provides good insights into the uncertainty of the model-based estimates of the origin of the air
masses probed.



## 7 Summary

The present article provides an overview on some of the scientific achievements obtained within the EMeRGe IOP in Europe.

The EMeRGe campaign in Europe focused on the identification and measurement of the plumes of pollution from selected MPCs, i.e. their emissions, transport and transformation. EMeRGe achieved its measurement objectives by exploiting the unique capabilities of the HALO research platform to probe these plumes over a relatively large geographical coverage and by the use of forecasting models and tools.

The results obtained from EMeRGe provide new insights into the transport and transformation of pollution plumes over Europe during the IOP in July 2017:

- EMeRGe provides a unique set of in-situ and remote sensing airborne measurements of trace gases and aerosol particles along flight routes in the lower troposphere over Europe. The interpretation of the HALO measurement data is facilitated by the use of collocated ground-based and satellite measurements. In that respect, EMeRGe enhances previous pollution studies in Europe by adding an extensive experimental data set in the PBL.

- The selected MPCs are confirmed as pollution hot-spots by analysis using the aircraft measurements, backward and forward trajectories, dispersion models, CAMS tracer simulations and satellite observations.

- Distinct aerosol layering is observed over some of the investigated MPCs. Collocated ground-based remote sensing instruments improved vertical and temporal resolution as compared to HALO. The synergetic use of these data improves the understanding of the evolution of the airborne observed scenarios and the attribution of the vertical distribution of pollutants probed during the shuttles flights.

- Plumes originating from European MPC outflows are typically observed below the top of the BL at 2000 m and occasionally after being transported over long distances. The location and position of the city plumes are typically well forecasted by the CAMS-global, MECO(n) regional and by HYSPLIT dispersion simulations using urban city tracers.

- The composition of the pollution plumes measured along the flight tracks depend on the MPC emissions and the mixing with air from other emission sources. Enhancements in the concentration of selected species, such as CO, $NO_y$ and VOCs such as $C_6H_6$ and $CH_3CN$ measured on-board HALO, enable the identification of anthropogenic and BB signatures in the plumes.

- Isotope measurements in VOC samples collected at MPC ground sites and on-board HALO enable the determination of atmospheric residence times and the source apportionment. Different ranges of $\delta^{13}C$ values in VOCs are determined and attributed to MPC sources, e.g. for $C_6H_6$ in the Po Valley and Rome for the first time.

- Signatures of urban sources of long-lived greenhouse gases like $CH_4$ and $CO_2$ are identified in the airborne measurements in plumes close to the MPC regions in Europe. The identification of plumes of GHG and the quantification of the MPC contributions to the regional GHG budget are challenging. This results from the long lifetime of these gases which yields a well-mixed and large atmospheric background, and the distance from the MPC to the sampling.

- The aerosol inside the MPC plumes is typically dominated by smaller particles which are clearly visible in the total aerosol number concentration for the aerosol radius in the range 0.01 to 3 μm.


- Tagging of polluted air masses in the centre of MPCs by ground-based releases of PFC tracers provides a unique opportunity to identify successfully and unambiguously MPC outflows after transport times of between 5 and 26 hours. The tracer experiments during EMeRGe additionally test the ability of models (HYSPLIT, FLEXPART, FLEXPART-WRF, FALL3D) to simulate the transport and dispersion of the tracer for different meteorological conditions and topography around the release sites. While the simulated position of the PFC plumes agrees with the measurements, the tracer mixing ratios calculated by the dispersion models are by a factor 2 to 3 higher than detected. The degree of agreement between the tracer simulations and observations depends on the parametrisation of dispersion and the representation of the topography in the models, as well as the goodness of tracer sampling in the plume, e.g. matching the maximum PFC concentrations was not always possible due to restrictions by air traffic control and flight endurance. EMeRGe is one of the first airborne measurement campaigns to use this air mass tracer approach and has successfully demonstrated its value.

- Regional transport of several European MPC outflows is successfully identified and measured: a) London over the English Channel to Central Europe, b) Po Valley either North over the Alps or in a south-easterly direction towards the Adriatic, c) Rome over the Apennines into the Adriatic and d) Madrid and Barcelona into the Western Mediterranean.

- BB emissions mix frequently with anthropogenic pollution during the transport over Europe. BB signatures are encountered in a large fraction of the pollution plumes probed during the EMeRGe IOP.

- BB also contributes significantly to the concentration of pollutants above the PBL and represents an important particle source over Europe, in addition to urban, industrial emissions and mineral dust. BB observed during EMeRGe at altitudes above 5000 m is attributed to be in older masses, which had originated in North American fires, in agreement with models.

- Mineral dust is identified in the aerosol size distribution and the optical properties of some of the air masses probed in Southern Europe above the PBL, in agreement with space and ground-based observations.

- The photochemical activity as indicated by the presence of free radicals varies widely in the plumes. The largest peroxy radical, $RO_2^*$, mixing ratios are observed below 3000 m in Southern Europe. This is expected and results from higher insolation and temperatures, which accelerate the photochemical processing. The $O_3$ production rates calculated from the $RO_2^*$ measured on-board are in the same order of magnitude as those reported in urban pollution for mixing ratios of $NO < 1$ ppbv.

- HONO mixing ratios detected in the PBL and lower part of the free troposphere often exceed mixing ratios expected from known gas-phase reactions as indicated by comparisons with model simulations. Potential mechanisms for the heterogeneous HONO formation are explored using theoretical studies in combination with the gas-phase, aerosol composition and radiation observations

- The photochemical processing of the MPC outflows during transport is inferred from the airborne measurements. Ratios of species such as $NO/NO_y$, NO/VOC and $C_7H_8/C_6H_6$ and observations of oxidation proxies such as peroxy radical concentrations and organic aerosol composition indicate with reasonable agreement that chemical processing of the MPC emissions identified during EMeRGe was substantial. Measurements of $\delta^{13}C$ isotopes survey the chemical processing of MPC London plumes and of the MPC Rome outflow during the transit over the Apennines.

- The analysis of the aerosol composition during EMeRGe indicates that aerosol photochemical processing is fast under European summer conditions. Chemical processing modifies both the chemical properties and





the partitioning between gas and particle phase in the air masses over Europe. Simultaneous measurements
of organic ions, CO and $C_7H_8/C_6H_6$ and $NO_y/CO$ ratios on-board enable dilution and processing in the
plumes to be discriminated.
•   PFC tracers and adequate transport models are shown to be of indispensable value to quantify the
processing of MPC plumes at large distances from the sources. Mixing of plumes from the release to the
observation limits the application of VOC clocks, such as the ratio of $C_7H_8$ to $C_6H_6$, for the investigation of
the transformation of MPC outflows on large scales.
•   The precise knowledge of the transport times between the source regions and the HALO sampling sites in
the plumes obtained from the PFC experiments and dispersion models enables the analysis of chemical
transformations during transport, e.g. oxidation of $SO_2$ and formation of HCOOH. The photochemical
formation of HCOOH is shown to be the main source of HCOOH during the EMeRGe IOP in Europe.
HCOOH is found to be more abundant in the plumes than the precursor species of inorganic acids, $NO_2$ and
$SO_2$.
•   Secondary organic aerosol prevails in the polluted air masses probed in Europe above 2000 m. In the free
troposphere above 4000 m the direct effect of anthropogenic emissions on the organic and inorganic
aerosol components is observed to be small.
First efforts to simulate observations of the EMeRGe flight tracks were made with the global/regional
chemistry-climate model MECO(n). Further investigation of small-scale effects by complementary model
activities with validated data includes the development of a box model to account for fast chemical
transformation of pollution in air masses along the flight tracks. The EMeRGe set of airborne data supports
photochemical transport models to assess:
•   the relative contribution of biogenic, BB and anthropogenic sources to the VOC burden over Europe,
•   the net ozone production in the investigated MPC outflows in relation to the transport time and mixing of
the pollution plumes,
•   the contribution of VOC species such as glyoxal and/or methylglyoxal to secondary aerosol formation in
aged pollution plumes,
•   the adequacy of Angstrom coefficients, aerosol fine mode fraction products and the geostationary satellite
derived AOT to identify aerosol sources and transport features of mixing events of anthropogenic particles
and mineral dust, and
•   the significance and representativeness of the transport and concentration patterns obtained during
EMeRGe in summer 2017, which was a period with anomalous meteorological conditions in Central
Europe.
The collected data during EMeRGe help to improve the current understanding of the complex spatial distribution
of trace gases and aerosol particles resulting from mixing, transport and transformation of pollution plumes over
Europe. The wide range of observations presented here is the basis for further work being addressed within
dedicated studies. More detailed analyses of individual data sets are provided elsewhere. Prospective
deployments of similar characteristics are desirable to consolidate and contextualise the EMeRGe results in
Europe.
The analysis of the EMeRGe data obtained in the second IOP in Asia will be presented in separate publications.





**Acknowledgements**


The authors thank the following teams and individuals, without whom the EMeRGe in Europe IOP would not have been
possible:
• HALO flight organisation, permissions and related
the DLR-FX and the HALO EMeRGe team. Special thanks to Lisa Kaser, Frank Probst, Michael Großrubatscher, Stefan
Grillenbeck, Marc Puskeiler, for flight coordination and planning, to Alexander Wolf, and Thomas Leder, the flight
engineers and to the BAHAMAS team. The authors also thank enviscope GmbH in particular of Nicole Brehm and Rolf
Maser for the support during the integration and preparation phase of the IOP in Europe.
• Meteorological and chemical composition forecasting
Michael Gauss and Álvaro Valdebenito (MetNo) for provision of EMEP forecasts for the campaign and
CAMS/ECMWF, in particular Johannes Flemming and Luke Jones for providing the weather and trace constituent
forecasts for the field campaign support. The CAMS-regional modelling team are also acknowledged for providing
regional model forecast data for Europe.
• LIDAR Observations
EARLINET for providing aerosol LIDAR measurements and DWD, ALICE-net and RMI for ceilometer measurements.
The support from AERONET, Service National d'Observation PHOTONS/ AERONET-EARLINET part of the
ACTRIS-France research infrastructure and GOA-CF, part of ACTRIS-Spain, for their continuous efforts in providing
high-quality measurements and products, and in particular of all PIs and Co-PIs of the AERONET sites contributing to
EMeRGe for maintaining their instruments and providing their data to the community is greatly appreciated.
• Luca Ferrero (GEMMA and POLARIS Research Centers, Department of Earth and Environmental Sciences, University
of Milano-Bicocca) for the air samples collected at the ground in Milan (Italy) during the HALO flights,
• Tracer releases
Jonathan E. Murray and Helen Graven and the Imperial College team for releasing the PFC tracer in London.
KK and JohS would like to thank Christiane Schulz and Philipp Schuhmann for support during the integration phase. BAH,
OOK, CP, DW, UP and MLP would like to thank Thomas Klimach, Björn Nilius, Jorge Saturno, Oliver Lauer and Meinradt
Andreae for support during the EMeRGe campaign in Europe and during the data analysis.
MDAH, MG, YL and JPB thank Wilke Thomssen for support during the preparation and integration phases of EMeRGe and
Heiko Schellhorn for continuous technical support and retrieval of model data during the campaigns.

**Funding information**


The HALO deployment during EMeRGe was funded by a consortium comprising the German Research
Foundation (DFG) Priority Program HALO-SPP 1294, the Institute of Atmospheric Physics of DLR, the Max
Plank Gesellschaft (MPG) and the Helmholtz-Gemeinschaft.
FK, BS, and KP acknowledge the support given by the DFG through the projects PF 384-16, PF 384-17 and PG
385-19. KB acknowledges additional funding from the Heidelberg Graduate School for Physics. JohS, KK, and
SB acknowledge funding through the DFG, project No. 316589531. LE and HS acknowledge support by DFG
through project MEPOLL (SCHL1857/4-1). AH would like to thank DAAD and DLR for a Research
Fellowship. HS acknowledge financial support by the DLR TraK (Transport and Climate) project.   MS
acknowledges support from the EU (GA no. 654109, 778349, 871115 and 101008004) and the Spanish
Government (ref. CGL2017-90884-REDT, PID2019-103886RB-I00, RTI2018-096548-B-I00 and MDM-2016-
1586 0600).

MG, YL, MDAH and JPB acknowledge financial support from the University of Bremen. FLEXPART
simulations were performed on the HPC cluster Aether at the University of Bremen, financed by DFG within the
scope of the Excellence Initiative.  A.-M. Blechschmidt was partly funded through the CAMS-84 project.
JW acknowledges support from the German Federal Ministry for Economic Affairs and Energy – BMWi (project
Digitally optimized Engineering for Services – DoEfS; contract no. 20X1701B)
TK thanks DLR VO-R for funding the young investigator research group "Greenhouse Gases".
MM, PJ, MK acknowledge resources of the Deutsches Klimarechenzentrum (DKRZ) granted by the WLA
project ID bd0617 for the MECO(n) simulations and the financial support from the DLR projects TraK
(Transport und Klima) and the Initiative and Networking Fund of the Helmholtz Association through the project
"Advanced Earth System Modelling Capacity" (ESM).
BAH acknowledges the funding from Brazilian CNPq (process 200723/2015-4).



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
