# Peer review of "Overview: On the transport and transformation of pollutants 1"

_Atmospheric Chemistry and Physics, 2021_

## Author Comment (AC1)

**Comment on acp-2021-500**

**Anonymous Referee #1**

*I will start by apologizing for the delay in providing this review but will also note that this paper is VERY long. Much too long in my estimation. Overview papers are different than typical journal articles in that they are not expected to provide great scientific depth. Rather, they are intended to convey details that summarize a field study and can serve as a reference for the many related manuscripts, absolving them of repeating the many details summarized in the overview. As an overarching comment, I would suggest that the salient details of EMeRGe could be encapsulated in a much more condensed manuscript, but I will not attempt to specify how this might be done. Rather, I will comment on the material presented in its present form while also noting that the length of the paper will frustrate many readers and likely reduce its impact. I will also say that the length does not translate into scientific depth as myriad topics are touched upon but never satisfactorily developed to be useful. I would prefer to wait for the papers.*

**Answer**: It has now been made a considerable effort to present the manuscript in a more condensed form addressing the comments and suggestions of RC1 and RC2. Sections have been reduced in its length or moved to the electronic supplement. The text has been thoroughly restructured, in particular from section 4 onwards, to emphasise major findings. Overall, the length of the manuscript has been reduced in 15 pages.

*The most important omission in the current manuscript is the lack of any statement or information on the availability of the EMeRGe data. I apologize if I have missed it somehow, but a check of the EMeRGe website at http://www.iup.unibremen.de/ emerge/home/home.html is also devoid of any instructions or indication of how the data might be obtained. I would emphasize that this is a data set that is expected to have value to broader community beyond the EMeRGe team, and should be available at this point, ideally through a public portal, but minimally through a request that can be submitted via the website. With the campaign completed four years ago and manuscripts being written, the time is long past for any reasons to withhold data for quality control or proprietary purposes.*

**Answer**: The authors thank RC1 for the reminder. The section "data availability" has now been included in the revised version at the end of the paper as indicated by ACP.

**"Data availability**

The EMeRGe data are available in the HALO data base (https://halo-db.pa.op.dlr.de/) and can be accessed upon registration. Further data can be made available upon request to the corresponding author. "

**Specific Comments:**

*Lines 112-133 "secondary pollutants such as ozone or secondary organic aerosols (SOA)"It is not only happening in this paper, but sometimes it feels like the community has forgotten that secondary aerosol production includes more than just organics. Why not say "secondary aerosol (organic and inorganic)"?*

**Answer**: On line 131 the sentence has been changed as: "Primary MPC emissions are transported and transformed into secondary pollutants such as $O_3$ or secondary aerosols (organic and inorganic) and lead to smog episodes downwind of the source."

*Lines 124-132 "important unresolved issues" "inaccurate modelling", "insufficient subgrid parameterization", Inadequate characterization", Inaccurate prediction" The phrasing of*

*these bullets emphasizes the negative, leaving the impression that we are a community without capability. I think they could be stated more positively in terms of advancing capability. In that sense, EMeRGe contributes to the long history of providing observations to challenge models and investigate processes that contribute to continuous and incremental progress in the capability to forecast and simulate atmospheric composition and chemistry. In that sense, EMeRGe is unlikely to fully "resolve" any issues but still provides a clear step forward.*

**Answer**: The text has been modified by rather emphasising achievements in order not to give the wrong impression of a "community without capability"

*Lines 139-140 "The predicted changes in these patterns indicate that future air quality in MPCs will generally be less influenced by local emission sources than by the mixing of anthropogenic and natural emissions outside the MPC (Butler et al., 2012)." I both disagree strongly with this statement and think that it also misrepresents the referenced paper. The paper refers to MPC impacts on the global atmosphere, not local conditions. Even the abstract states that under one of the scenarios "the local influence of megacities in increased". That said, I think the real problem is that localities around theworld continue to point a finger upwind when local emissions will always dominate the local problem in an MPC. I would remove this sentence.*

**Answer**: The sentence has been removed.

*Lines 143-154 "Medium and long-term effects of anthropogenic emissions and their interaction with natural and biogenic emissions in the local and regional surroundings of individual MPCs are poorly understood and imprecisely quantified." Again, I think this language is too strong. While I agree that quantification is a problem, I do not agree that it is poorly understood. We actually know quite a bit about how this all works, but properly capturing the combined influence of emissions, chemistry, and dynamics for specific locations is still a big challenge.*

**Answer**: The section "Introduction" has been accordingly modified.

*Lines 149-150 "The current knowledge on all these aspects is still insufficient." While this statement is not necessarily incorrect, I think it would be more constructive to say that even as we make progress in understanding MPC emissions and their local and downwind impacts, the landscape of emissions and conditions in each location continues to evolve and requires ongoing attention.*

**Answer**: The whole section "Introduction" has been modified. In line with the comments by RC1, following text can be now read on line 152:

"However, capturing the combined influence of emissions, chemistry and dynamics for specific locations is still a big challenge. In particular, inconsistencies in local and regional MPC inventories (e.g. Denier van der Gon et al.; 2011, Mayer et al., 2000; Butler and Lawrence, 2009), and limitations in the prediction capabilities of pollution transport patterns and cumulative pollution events in downwind regions of MPCs (Zhang et al., 2007; Kunkel et al., 2012) require ongoing attention. Furthermore, controlling policies, changes in land cover and climate continue to evolve and might substantially modify the relation between anthropogenic emissions and both natural aerosol and trace gases, as predicted by e.g., Butler et al., (2012), and recently reported for East Asia (Fu et al., 2016; Silver et al., 2018 and references herein; Leung et al., 2018)."

*Line 176 "might" should be changed to "is expected to"*

**Answer**: The corresponding sentence has been modified.

*Lines 286-287 and Figure 1* **"The differences observed are most likely related to the special weather situation in 2017, as described in 286 Sect. 3.1." While I appreciate the detailed offered by the four panels, there are no striking or noteworthy differences. All the same hotspots of NO2 are there in each image. There is also no further discussion of these images in the text. Why not cut this down to a single satellite image with the hotspots for the MPC selections highlighted. After all, "Selection of MPC targets" is the purpose of this short section.**

**Answer**: Figure 1 has now only two panels for 2017 observations and the sentence has been removed.

*Table 1a* **FLEXPART is listed in the table, but never discussed. Does this reflect that it was not used as much?**

**Answer**: FLEXPART trajectories were an important part of the plume identification and tagging and are described in the supplementary information (S12).

*Table 1b* **This list seems quite limited. Were products from any other satellites (e.g., IASI) used?**

**Answer**: No, there were no other satellite products used. The authors agree that many more satellite data are available which could in principle be interesting for analysis of the EMeRGe data. Unfortunately, neither TROPOMI, nor GEMS were launched at the time of the campaign. IASI data would be an interesting option but so far, they have not yet been used in combination with EMeRGe data.

*Line 332* **The word "shuttle" is used here in a nonconventional manner and seems to have been coined to describe a specific pattern of stacked flight legs. Since the word is used throughout, I think it would help to formally define it. It would help to say "...incorporated vertical shuttles. Shuttles are defined here as..."**

**Answer**: The text has been accordingly extended in line 302:

"Consequently, the flights over Europe made use of the HALO long-endurance capabilities to fly in the PBL and incorporated vertical shuttles. Shuttles are defined here as a descent or climb pattern between holding altitudes. Typically, three flight levels (FL), upwind or downwind of the target MPCs were part of the shuttle."

*Section 3.1* **This meteorological discussion is rather cursory and does little to help place later discussions in context.**

**Answer**: The authors think that this short description of the overall meteorological situation in July 017 is necessary to place the campaign into a meteorological context. The text in this subsection has now been modified. References to upcoming sections are given, where the present meteorological information serves as a base for the understanding of the presented results. The meteorological situation of the individual flights is discussed in section 3.6.

*Section 3.2* **The discussion here is again overly simplistic. The statistics in the figure and text are rounded to one significant figure, which leads to strange numbers with all standard deviations being either 0.01 or 0.02. When I look at Figure 3, Rome has the least interannual variability of all, so I would consider it to be most representative of long-term conditions, but it is held out as the anomaly.**

**Answer**: The statistics have been recalculated by referring to the AERONET level 2.0 climatological tables. Now the AOD values at 500 nm are shown. This is for the higher aerosol signal in the visible with respect to the 1020 nm previously used and for better comparability to satellite observations data (AOD generally provided in the green). For those AERONET instruments not having the 500 nm filter, the values are interpolated using the Angstrom coefficient between the two closer wavelengths. Also note that the climatology has been extended further back in time (previously limited to 2009-2019) to gain a more meaningful statistics.

*Lines 496-498* **"Overall, 60% of the HALO measurements during EMeRGe in Europe were performed below 3000 m to probe fresh and transported outflows of selected MPCs (see Fig. 5 for the distribution of HALO flight altitudes during the EMeRGe IOP)."It would also be interesting to know how much of the data falls into the PBL, which varies with altitude. Also, is the altitude in figure 5 a pressure altitude?**

**Answer**: There is only punctual direct information about the height of the PBL along some of the flights. According to PBL thickness values obtained from ERA5 reanalysis, 20% of the data falls into the PBL. The altitude in Figure 5 is the WGS84 (World Geodetic System 1984) Elliptical Height.

*Table 3:* **The use of Flight Route (FR) could be confusing to the reader as it implies a specific flight path, but all of the paths are unique. You could keep "FR" but changing the term to "Flight Region" seems more appropriate.**

**Answer**: Flight route has been replaced by flight region in the text and tables.

*Line 599:* **Further details on all the flight tracks and shuttles are given in the supplement (S9).The tables in S9 read like shorthand for the science team and are not easily deciphered. If the authors feel it is necessary to describe every flight in detail, it should be easier to follow.**

**Answer**: The tables have been modified to be understandable.

*Figure 8* **These composite images of CO are somewhat overwhelming as they contain details that are not specific to any particular flight day. Is this much detail necessary? How is it helping the reader?**

**Answer**: Analysis of the CAMS global model output during and after the campaign has shown that there are many similarities between the pollution transport patterns of the individual flights towards the North and the same is true for flights towards the South. The composite averages are therefore representative of the main pollution transport patterns of the individual flights towards the North and the South. The following sentence has been added in section 3.4 (now 3.7) for clarification: "A division into southwards and northwards flights is meaningful, as pollution transport patterns during individual flights in the two subgroups mainly resemble each other."

*Figures 10 and 11* **In the captions for each figure, readers should be alerted that the domain of the upper right panel is different, reaching much further to the east. The upper right panel also appears to be mislabelled as 925 hPa in both figures. Isn't it showing 500 hPa?**

**Answer**: The wrong labels have been corrected. There was already a note on the domain in the caption of Figure 10 (now Fig. 9) which has been slightly changed and the figure caption in Figure 11 (now Fig.10) has also been extended to point this out: "Note that

the BB tracer from North America (top right) is shown on a larger map than the other CAMS forecasts in this image."

*Line 660-661* **"The chemical composition and the extent of photochemical activity of the air masses probed during the EMeRGe IOP were different for the different flight routes and tracks." This sentence doesn't say anything useful. I would remove it.**

**Answer**: The sentence have been removed

**Table 4 I don't find the values in this table or the supplement to be terribly useful. I would much rather see contrasts between MPC values and differences in pollutant mixtures between MPCs and the surrounding atmosphere. If you keep the table, it is important to specify what was measured in situ versus remotely sensed. For instance, it is important to acknowledge that if NO2 and HONO were remotely sensed, those values are not entirely local to the aircraft.**

**Answer**: The tables have been moved to the supplement (S11) and the Table caption has been extended by: "Note that HCHO, $NO_2$, HONO, $CH_2O$, $C_2H_2O_2$ and $C_3H_4O_2$ were remotely measured by the mini DOAS instrument; for the averaging volume, see Kluge et al., (2020)."

In addition, following information has been included:

"The averaging volume of the remotely measured $NO_2$, HONO, $CH_2O$, $C_2H_2O_2$, and $C_3H_4O_2$ concentrations are given in the vertical by the field of view of the telescope (0.380) (see table 2 in Hüneke et al., (2017)) and at clear skies in the horizontal perpendicular to the aircraft's flight they range between 10 km/25 km near the ground and up to 75/100 km at 14 km altitude at 343.7 nm/477.6 nm, respectively, with considerable shorter (photon path) lengths occurring in the aerosol loaded and cloudy atmosphere (for details see section 2 in Kluge et al., (2020)). In flight direction, the skylight spectra are coadded within less than 30s, which averages over 6 km for a typical aircraft speed of 200 m/s. Detailed radiative transfer simulations however indicated, that the averaging kernel in horizontal perpendicular to the aircraft flight direction maximizes for clear skies at about 20 % of the averages (i.e. for 2/5 km near ground and 15 km/20 km at 14 km for 343.7 nm/477.6 nm, respectively) (see figure 6.8 in Raecke, 2013), which even supported to resolve filaments due to mixing of trace gases in the polar jet (see figure 17 in Oelhaf et al., 2019)."

Refs:

Oelhaf, Hermann, Björn-Martin Sinnhuber, Wolfgang Woiwode, Harald Bönisch, Heiko Bozem, Andreas Engel, Andreas Fix, Felix Friedl-Vallon, Jens-Uwe Grooß, Peter Hoor, Sören Johansson, Tina Jurkat-Witschas, Stefan Kaufmann, Martina Krämer, Jens Krause, Erik Kretschmer, Dominique Lörks, Andreas Marsing, Johannes Orphal, Klaus Pfeilsticker, Michael Pitts, Lamont Poole, Peter Preusse, Markus Rapp, Martin Riese, Christian Rolf, Jörn Ungermann, Christiane Voigt, C. Michael Volk, Martin Wirth, Andreas Zahn , and Helmut Zierseis, POLSTRACC: Airborne experiment for studying the Polar Stratosphere in a Changing Climate with the high-altitude long-range research aircraft HALO, POLSTRACC: Airborne experiment for studying the Polar Stratosphere in a Changing Climate with the high-altitude long-range research aircraft HALO, Bull. Amer. Meteor. Soc., doi:10.1175/BAMS-D-18-0181.1, 2019.

Raecke, J. R., Atmospheric Spectroscopy of Trace Gases and Water Vapor in the Tropical Tropopause Layerfrom the NASA Global Hawk, Master thesis at the University of

Heidelberg, Heidelberg, Germany, 2013 https://www.iup.uni-heidelberg.de/de/research/stratosphere/files/theses/masters/MScThesis_Rasmus_Raecke.pdf

*Figure 12 (and Figure 13) As expressed in other comments, I don't find these campaign average statistics to be terribly useful or insightful. There is absolutely nothing surprising in them. This is somewhat consistent with the very cursory discussion they receive in the text. Why is PAN only shown below 3000 m? I would be much more interested in seeingcontrasts between MPC plumes against the rest of the atmosphere.*

**Answer**: Figure 12 has been removed and the discussion in text of Figure 13 (now Fig.14) has been re-written. An instrumental limitation prevented PAN to be measured at pressures above 3000 m during the IOP in Europe. This was solved for the IOP in Asia.

*Lines 729-731 "The HCHO mixing ratios observed in the PBL and middle troposphere during EMeRGe are somewhat lower than the North American mixing ratios (see Fig. 14). This might be related to the fact that several EMeRGe flight tracks were carried out far from emission sources over the North and the Mediterranean Seas." This is precisely the problem with aggregating all campaign data into a single figure without any discriminating information. It skews the ability to compare with HCHO over North America. That said, I think the North American numbers look a little on the low end. What was the source of information? The reference to Kluge et al. 2020 is a paper about Amazonia. This deserves some scrutiny.*

**Answer**: In Kluge et al. (2020) (Figure 10), the HCHO measurements done over Amazonia are put into the context of previous HCHO measurements performed over North (NA) and South America, Brazil, and the West Pacific. Taking together all summer HCHO measurements previously performed over contiguous NA, it appears that HCHO concentrations over Europe were found to be overall smaller, which after all might be a matter of a larger marine influence to the air masses analysed over Europe, rather of than less of specific anthropogenic emissions of HCHO and its precursors over Europe than NA.

*Lines 745-747 "A detailed analysis of the complexity of the air masses measured and the variations encountered in individual flights is beyond the scope of the present work and will be presented in dedicated publications." I agree with this statement, but on the opposite end of the spectrum, lumping all of the campaign data together for presentation purposes does not solve the problem.*

**Answer**: The revised version of the manuscript intends to find a compromise between the alternatives mentioned by RC1. Some figures have been removed and the text has been thoroughly restructured, in particular from section 4 onwards, to emphasise major findings.

*Figure 15 The caption states "Note that mixing ratios measured at different altitudes in the shuttle areas are not distinguishable in the figure." The lack of ability to distinguish altitude is only one of the problems with figures like this. These large-scale views are just confusing, and you aren't pointing to any particularly interesting features in the text. Am I supposed to study them and see things for myself?*

**Answer**: The aim of Figure 15 (now Fig.11) is to give a general impression about the geographical distribution of selected species respect to the target MPCs. It is not intended to provide information about the vertical distribution of pollutants, which in the case of shuttles can be retrieved from curtain maps as in Figure 17 (now Fig.12). The

original description of the figure has been modified in the new section 4.1 to focus the view of the reader on the interesting features observed.

*Lines 784-786 **"Oxygenated VOC (OVOC) result from the oxidation of VOC emissions (e.g. CH3COCH3 or HCHO) and are strong sources of HO2 and CH3O2."While acetone is an OVOC, it is most certainly not a particularly strong source of HO2 and CH3O2 in the lower atmosphere. HCHO is dominant and there are a slew of other reactive VOCs that are making much larger contributions than acetone, which is not worth mention in this sentence.***

**Answer**: This is true and has been deleted. The corrected and modified text is now in section 4.5.

***Figure 17 The whole discussion of this figure is rather wandering. I am not sure what is trying to be said and why CO and CCN are being used together. Is it just that that there are north-south gradients? Specific layers are mentioned, but I don't know what I am to take away from this figure. When doesn't CCN ever show a strong vertical gradient?***

**Answer**: CO is a good tracer for relatively fresh combustion (e.g. Andreae and Merlet 2001; Andreae, 2019). The CO emitted by open biomass burning exceeds concentrations from anthropogenic sources. Thus, the good agreement for CCN and CO in the lower troposphere for the peak concentrations (color-coded in red in figure 17; now figure 12) implies biomass burning as source of these CCN. This relationship between CCN particles and CO is shown to be nearly linear in previous studies (e.g. Pöhlker et al. 2016, 2018). Furthermore, the relatively low CCN in the altitudes above 4000 m, where CO partly still shows increased concentrations indicates cloud processing and removal of the CCN.

The text has been modified as follows: (lines 600-606; section 4.1):

"Differences observed North and South of the Alps are e.g. evident in Fig. 12, showing a reasonable agreement in its geospatial distribution of the cloud condensation nuclei (CCN) and CO which has been documented as a nearly linear relationship within the PBL by Pöhlker et al. (2016, 2018). CO is a good tracer for relatively fresh combustion (e.g. Andreae 2019). The CO emitted by open BB exceeds concentrations from anthropogenic sources. Thus, the good agreement for CCN and CO in the lower troposphere for the peak concentrations (color-coded in red in Fig. 12) implies BB to be the source of CCN. Elevated CO observations not related to increases in CCN indicate aerosol removal by cloud processing."

***Lines 835-836 "Hence, C6H6 enhancements in the absence of CH3CN can be used to identify relatively "pure" anthropogenic pollution." When first introducing the idea of using enhancements to filter the data, it is important to be more quantitative in your language. Enhancement is too vague. What constitutes an enhanced value?***

**Answer**: The identification of large pollution plume events results from concentration enhancements of $CH_3CN$, $C_6H_6$ and $C_5H_8$ when being measured significantly (three times the instrumental noise) over their individual atmospheric background values or their LODs, resulting in thresholds of 184, 49 and 85 ppt, respectively.

The text in the new section 4.2 has been accordingly modified (lines 645-653):

"Anthropogenic and biogenic signatures were identified in the pollution plumes by using enhancements in the concentration of selected species, such as CO, $NO_y$ and VOCs measured on-board HALO. Measured large pollution plume events were initially categorised into a) anthropogenic pollution (AP), b) biomass burning (BB), c) mixed and

d) biogenic plumes, by using enhancements of $CH_3CN$, $C_6H_6$ and $C_5H_8$ over 184 ppt, 49 ppt and 85 ppt thresholds, respectively. These thresholds take into consideration three times the instrumental noise over the limit of detection (LOD) or the individual atmospheric background values. Anthropogenic polluted air masses were e.g. identified by the enhancements of $C_6H_6$ and absence of $CH_3CN$ in contrast with the unpolluted background air in the absence of both chemical tracers. Similarly, $CH_3CN$ enhanced plumes in the absence of $C_6H_6$ were identified as pure or aged BB events (see S12 for details)."

The information in S12 in the supplement has also been extended.

*Lines 898-901* **"These maxima are not apparent in the profiles of particle larger than 0.25 μm. This is consistent with the attribution of LRT of air masses from North America, where they had contact with BB emissions. New particle formation events cannot be excluded but are considered unlikely."** **Why would aged air only show a signal in the smaller particles? There is no expectation of gravitational settling for particles in these size ranges. I do not follow the logic in these sentences.**

**Answer**: The authors agree with the reviewer that an enhanced concentration of Aitken mode particles without a signal in the accumulation mode is not indicative of LRT of BB emissions. The source of the enhanced fine mode in those cases is more likely associated with the outflow of convective systems that were passed at some distance during some of the flights. All of the higher altitude legs were transfer legs, primarily across the Alps, and as such not in a specifically targeted measurement area. Therefore the sentence (new section 4.2, lines 728-730) has been changed as follows:

"These maxima are not apparent in the profiles of particle larger than 0.25 μm. The corresponding sequences can be associated with air masses from convective outflows giving rise to enhanced particle concentrations in the sub-100 nm size range."

*Lines 905-907* **"As a result of the time required by the emitted precursor VOCs to be converted into secondary organic aerosol, the anthropogenic organic aerosol concentration increases above 2000 m altitude."** **The reasoning for this statement is insufficient. PBL mixing would easily disrupt such a gradient. Isn't it also possible that there is a temperature effect on the volatility of SOA?**

**Answer**: The authors admit that this statement is very speculative. There is certainly an effect of temperature, but it is not obvious that this effect applies only to anthropogenic secondary organic aerosol and not to background OA. The maximum in the organic mass concentration is mainly dominated by one event from flight E-EU-09 with high values above Spain (up to 8.5 µg m$^{-3}$ in both anthropogenic and biomass burning influenced air). For this figure the data was filtered for anthropogenic emissions only. In connection to chapter 4.3 (now line 932 onwards), the trajectories show uplifted and transported air masses from Madrid and Portugal. So, the feature is explained by a single event with strong convection above Spain together with an elevated boundary layer or transport out of the boundary layer, followed by secondary organic aerosol formation in this altitude range.

The lines 904-907 (now 739 onwards) have been accordingly rephrased to:

"Differences in the median vertical profiles of the inorganic and organic aerosol (OA) suggest that organic aerosol in anthropogenic air masses is mainly formed by secondary processes. The OA maximum between 2000 and 4000 m observed in the anthropogenically influenced air masses can be explained by one particular measurement period above Spain during flight E-EU-09. The trajectory analysis shows an uplift and

transport of anthropogenic influenced air masses from Madrid to the measurement location. Further possible reasons might also be lower temperature leading to enhanced SOA formation in this altitude range, but also a longer conversion time of VOCs to SOA in comparison to the conversion time for inorganic aerosol precursor gases. In contrast, the inorganic components of the aerosol…"

*Figure 20* **Based on the figure and text, I frankly don't know what it is you want me to see in this very busy data.**

**Answer**: The description in the figure caption of Figure 20 (now Fig.19) was unfortunately wrong. In order to gain in clarity, the caption has been corrected and the plots for the same species have been placed one above the other. In such a way the comparison of values between North and South regions is optically easier.

The correct figure caption of Figure 19 is now:

"Vertical distribution of $\delta^{13}C$ values in $C_5H_{10}O$ (left column) and $C_6H_6$ (right column) in whole air samples taken on HALO and at the ground sites in London, Wuppertal, Milan and Rome. Data for northbound flights (top row) are colour coded for Paris MPC (black), North Sea (red), English Channel (violet), BNL/Ruhr (orange). Data for southbound flights (bottom row) are colour coded for Rome MPC (blue), Po Valley MPC (cyan) and East Mediterranean (green). The coloured shadings refer to the standard deviation of $\delta^{13}C$ values in altitude bins of 250 m. Mean $\delta^{13}C$ values of the respective altitude bins are represented as solid colour-coded lines. The $\delta^{13}C$ values at the lowest altitudes in each colour represent the results of air samples at the ground stations: London (red), Wuppertal (orange), Rome (blue) and Milan (cyan). Error bars in $\delta^{13}C$ are given for each sample value. Remaining data are shown accordingly in grey."

**Figure 21 Why is the peak in the size distribution for AP&BB greater than either of them separately?**

**Answer**: During the measurement flight E-EU-06 it was observed a complex mixing of different open biomass burning sources. It was measured lightly aged biomass burning smoke from fires in Croatia, where grassland and savannas were the dominant combustion fuel, while the contribution of the fires in Italy to the BC burden over the Adriatic Sea were mostly from mixed forests and savannas. For the separation of the plumes were used the VOC flags as described in I), section 4 (now in S12 in the supplement). Thus, the contribution of different smoke sources, which strongly affects average BC mass size distributions, was disregarded in this analysis. However, it was shown the significantly smaller BC mass size distribution for urban haze, marked as pure anthropogenic. The resulting sizes agree with literature values (e.g. Schwarz et al., 2008; Liu et al., 2020, Holanda et al. 2020).

The text (Section 4.4, lines 987-1001) has been modified as follows: "A more robust indicator for particles from BB is BC. BC particles are formed in processes of incomplete combustion, and therefore are an important component of both BB and urban aerosol particles (Bond et al., 2013). The microphysical properties of BC give insights into the combustion sources and atmospheric ageing time of the pollution plumes (Liu, 2014, Laborde, 2012, Holanda et al., in preparation 2021). Figure 28 shows an example of average BC mass size distributions encountered during the E-EU-06 flight. A complex mixing of different open BB sources with lightly aged BB smoke from fires in Croatia was observed. Grassland and fires in Italy mostly from mixed forests and savannahs were the dominant combustion fuel. The plumes were classified according to the VOC observations

(see S12). This complex mix of biomass burning (BB, core diameter (Dc) = 200 nm) BC sources, got occasionally mixed with anthropogenic emissions (BB+AP, Dc = 210 nm). Rather pure anthropogenic urban haze (AP) with significantly smaller mean modal diameter (Dc = 170 nm) was additionally measured. The resulting sizes agree with literature values for urban haze and BB smoke (e.g. Schwarz et al., 2008; Laborde et al., 2013; Liu et al., 2020; Holanda et al. 2020). During E-EU-06, the average total BC mass concentration was also substantially higher in relatively pure BB smoke and in the mixed conditions with urban haze (BB, 0.61 ± 0.12 µg m$^{-3}$ and BB+AP, 0.81 ± 0.35 µg m$^{-3}$, respectively) than in urban pollution (AP, 0.35 ± 0.15 µg m$^{-3}$)."

**Line 1213** "Fig. 26" should be "Fig 33"

**Answer**: This is correct. The numbering of the figures has however changed with the restructuring of the text.

*Figure 37 From the figure, I have a hard time convincing myself that the various aging metrics provide any nuanced information beyond a simple distinction between background versus polluted conditions.*

**Answer**: This is one of the focus points that need further investigation. The simple distinction between anthropogenic and background conditions in the rightmost plot is based on a benzene and acetonitrile threshold and does not give any transition information. Therefore, we study other metrics in order to find out more on the transition of organic aerosol from fresh emissions to aged emissions. Thus, these metrics are mainly used to explain the processing. The results combine particle phase with gas phase measurements and it is remarkable that the metrics for both phases are consistent. Furthermore, even in the regime tagged as "background", an evolution of the processing is visible for the particle phase.

No changes in manuscript.

*Line 1400 "remanence" do you mean "remnant"?*

**Answer**: Yes. This has been corrected in the text.

*Lines 1434-1435 "The results obtained from EMeRGe provide new insights into the transport and transformation of pollution 1434 plumes over Europe during the IOP in July 2017."This paper does not cover results, nor does it provide any new insights. That will come from the other publications. I would suggest changing this sentence to say, "The ongoing analysis and publication of EMeRGe results is expected to provide new insights into the transport and transformation of pollution plumes over Europe during the IOP in July 2017."*

**Answer**: The complete Section 7 (now section 5) has been re-written. The sentence suggested by the RC1 is now used in the introduction of section 4 (lines 544-545).

*Lines 1441-1442 "The selected MPCs are confirmed as pollution hot-spots by analysis using the aircraft measurements, backward and forward trajectories, dispersion models, CAMS tracer simulations and satellite observations. "The word "confirmed" seems rather strange. Did the concept of cities as major pollution sources need confirmation? Why not say "The downwind impact of pollution from MPCs was identified and explored using..."*

**Answer**: The complete Section 7 has been re-written. The sentence suggested by the RC1 is now used in section 4.3 (line 845).

**Lines 1543-1544** *"Prospective deployments of similar characteristics are desirable to consolidate and contextualise the EMeRGe results in Europe." I am going to disagree with this statement. While it is not my job as a reviewer to judge the approach taken in EMeRGe, I must confess that I am not convinced that this was the best approach to the problem. Thus, suggesting more flights of the same type is something I cannot fully endorse. Instead, I would prefer to hear something like the following. "Continued scrutiny of the EMeRGe observations and the development of lessons learned will be needed to build upon and improve airborne measurement strategies for future deployments focusing on pollution in Europe."*

**Answer**: The Section 7 (now Section 5, line 1245 onwards) has been completely re-written in line with this comment and the suggestions of RC2.

---

## Author Comment (AC2)

**Comment on acp-2021-500**

**Anonymous Referee #2**

*The EMeRGe project and accompanying field campaigns have been a tremendous overall contribution to the field of atmospheric chemistry, providing extensive data and insights into how atmospheric pollutants are transported and transformed in the outflow of European and Asian megacities and other major population centers (MPCs). This overview paper brings together four distinct threads: 1) a review of the historical background and framing of the campaign; 2) the campaign setup, objectives and actual operations, along with the modeling and satellite observation contributions; 3) a presentation of the main observations; and 4) key scientific insights that can be gained from these observations. The quality of each of these individual components is very high. However, putting all of these together into one paper leads to an extremely long manuscript, which is nearly all on the same "level", with only a few very technical details being in the electronic supplement. There is nothing principally wrong with providing this extent of information to the community. However, having it all as one long, linear (as opposed to multi-level or hyperlinked) paper makes it quite a hindrance for readers to find the time to go through it adequately (especially not spreading it across various sittings over a week or more). This has contributed to the long delay in submitting this review; like the other referee, I sincerely apologize for this delay. Since this hindrance is likely to apply to other busy colleagues as well, leading to much less uptake than possible by the community of potentially interested readers, I provide suggestions for how to potentially improve this significantly. This manuscript should definitely be published, due to its excellent content. But prior to publication, it should be restructured in order to be appealing and directly useful to a much larger readership. Here I outline a suggested restructuring. This is not the only possible approach, and it is fine if the authors pursue a different strategy for restructuring, as long as it fulfills the same purposes of better distinguishing the four elements noted above, and bringing forth the highlights and putting the more mundane (albeit important) aspects on a "lower" level (e.g., the electronic supplement, an appendix, etc.).*

*The abstract is not going to be very appealing to many readers, starting off directly with the project/campaign name and setup/components, with only a short paragraph on the key scientific insights. I would suggest starting off with something like "Megacities and major population centers (MPCs) worldwide are major sources of air pollution, both locally as well as downwind. Characterizing this outflow has been XYZ... Here we provide an overview of the highlights of a major new contribution to the understanding of this issue, based on the data and analysis of the EMeRGe campaign...". Then follow the order of the four points noted above, moving the current last paragraph earlier and partly merging it, and expanding the current second-to-last paragraph to be a more appealing highlight description of the most interesting and important findings.*

**Answer**: First of all, the authors are thankful for the encouraging suggestions of RC2 to improve the structure of the manuscript. It has now been made a considerable effort to present the manuscript in a more condensed form addressing the comments and suggestions of RC1 and RC2. Overall, the length of the manuscript has been reduced in 15 pages.

The abstract has been modified and re-structured according to the suggestions of RC2.

**For the main text, I suggest the following: Introduction: Keep this short. Move the historical overview (lines 89-150) into section 2, and here just focus on the framing and introducing the EMeRGe campaign (without making that a subsection - it's a bit odd to have only section 1.1 and no 1.2).**

**Answer**: This has been done

*Section 2: make this a "Historical Review" section, combining the material from section 1 (lines 89-150) with Section 2.1. The present historical overviews in these two sections are very fitting and seemingly accurate, to my knowledge. Combining these sections will help consolidate the topic within the paper. Check for and reduce any redundancy. In this section, it would be good to be clearer about what are seen as the most important knowledge gaps (presently only a list of "Some examples are…" is given). These should then be connected back to in describing the most important findings, which should help to fill some of these gaps.*

**Answer**: This has been done.

*One minor point: MEGAPOLI was a project with largely modelling analysis at various scales, plus an embedded field campaign in Paris, so line 203 should read "The MEGAPOLI field campaign was conducted in Paris…".*

**Answer**: This has been corrected.

*Then sections 2.2-2.4 become the new section 3. To facilitate reading, I'd put the questions in 2.2 into a table, possibly with an extra column indicating where in the campaign or the overview paper each question is addressed (optional, but that would help the reader to orient and quickly track down the parts that are most interesting or relevant to them).*

**Answer**: These sections have been combined to form section 3.

*Most or even all of Section 2.4 could go into either the electronic supplement, or a "Methods" appendix. Only keep here what is really needed for understanding the campaign, which will interest most readers. (The specialists who are interested in the instrument details will know where to find that information in the supplement or appendix.)*

**Answer**: Most of the section 2.4 (now 3.3) has been moved to the supplement.

*Section 3 would become section 4. That's good for the first part, but the flight route descriptions (lines 514-599) can easily go into the electronic supplement for reference by those who are interested, and in the main text keep only what is really needed (if anything) to understand the main results and new findings in the later sections.*

**Answer**: The authors think that the description of the flight routes is necessary for understanding the campaign as it provides a general overview of the coverage of the flights and the meteorological conditions. The flight routes are now included as section 3.6.

*Section 3.4 is good as it stands.*

**Answer**: The section 3.4. is now included as section 3.7.

*From then on comes a larger restructuring: Sections 3.5 through Section 6 are all written in the form "Here's what we observed" –then described in great detail – followed by "...and this is what that means or tells us about the atmospheric outflow from the MPCs". While that's OK for a short (e.g., 5-10 page) paper, it honestly gets rather boring for a paper with 65 pages of text. The really exciting part is then bundled into bullet points in the "summary" in section 7.I would strongly recommend turning this all around: after the old section 3.4 (now 4.4.), start new sections 5 and 6 (and perhaps 7) which are structured along the lines of the key findings noted in Section 7, and only provide the observational evidence from the campaign that is needed to support those findings (then possibly putting further general description of observations which do not connect to the main findings into the conclusions).*

**Answer**: The text has been completely re-structured according to the suggestions of the RC2. Now there is a new section focusing on the bullet points of the previous summary in section 7. The new section 4 (Transport and transformations of pollution plumes during the EMeRGe IOP in Europe) highlights the main findings of EMeRGe, which are sorted in six subsections: 4.1 Observations; 4.2 Identification, classification and characteristics of pollution plumes; 4.3 Identification of MPC outflows; 4.4 Mixing of MPC outflows with air masses of biogenic and natural origin: forest fires and dust; 4.5 Photochemical processing of polluted air masses during transport; and 4.6 Model simulation of EMeRGe observations.

*Finally, instead of a summary at the end, provide a good outlook, describing how the knowledge gaps that have been filled contribute to our overall understanding, what is important for the community to focus on next, and suggestions of how to go about that. If it can be done without spoiling things, then providing a "teaser" of a highlight or two of what EMeRGe found to be different in Asia, garnering interest for the next paper, would fit well in the outlook section; otherwise the Asian campaign does not need to be noted here, since it was already discussed in the opening sections. Following those suggestions would make this a much more valuable, accessible and appealing result from a tremendous and successful effort of the large team involved.*

**Answer**: A new section "5 Outlook" has been included at the end.

*One final minor comment: Some of the references in the list are inconsistently formatted, with the date before the doi, whereas most have the date after the doi (when one is given).*

**Answer**: This has been corrected.

---

## Author Response (AR2)

**Comment from RC2**

*The authors have done a commendable job of rearranging and condensing the manuscript, and in its present form it now does justice to the extensive and high-quality observations and analysis of the EMERGE campaign, and I think will be much more useful than the submitted version would have been for the community working on and interested in megacity air pollution.*

*I recommend that it be published as is, with only one final technical/editorial suggestion. Throughout the manuscript, one approach to condensing the text has been to use bullet points through out many of the sections. While this can sometimes be appropriate, in many of the sections the text after each bullet point consists of a complete paragraph with multiple sentences (and in some cases even multiple paragraphs and lettered lists under a single bullet point). I think it would improve the readability to remove these many bullet points. In many sections, that can simply be done, without needing any further changes. In a couple sections, in which the bullet points offset very asymmetric text (some with only one sentence, others with multiple paragraphs), some minor editing might be needed for readability, and in a few cases, where the bullet points are meant to offset various topics, introduction an additional level of sub-section headings might help. This should be at the authors' discretion to implement, but I think it would help further improve the readability of the final manuscript.*

**Comments from Editor**

*I am happy to accept your manuscript for publication in ACP after some remaining technical corrections. I agree with the reviewer that some bullet points you have used to structure your manuscript can be removed without loosing a lot of structure, and I recommend to do so. This is particularly the case all over section 4. In other sections the bullet points may be appropriate, but I would like you to check your manuscript again with respect to this issue.*

**Answer to RC2 and Editor:**

The text has been rearranged as suggested by the second referee by removing the original bullet points in most of the sections. In addition, typing errors have been corrected and the format of the figures has been improved to follow the manuscript preparation guidelines. The corrections are highlighted in the submitted document.